# Prediction via Shapley Value Regression

**Amr Alkhatib** [1]   **Roman Bresson** [2]   **Henrik Boström** [2]   **Michalis Vazirgiannis** [2 3]

## Abstract

Shapley values have several desirable, theoretically well-supported, properties for explaining black-box model predictions. Traditionally, Shapley values are computed post-hoc, leading to additional computational cost at inference time. To overcome this, a novel method, called ViaSHAP, is proposed, that learns a function to compute Shapley values, from which the predictions can be derived directly by summation. Two approaches to implement the proposed method are explored; one based on the universal approximation theorem and the other on the Kolmogorov-Arnold representation theorem. Results from a large-scale empirical investigation are presented, showing that ViaSHAP using Kolmogorov-Arnold Networks performs on par with state-of-the-art algorithms for tabular data. It is also shown that the explanations of ViaSHAP are significantly more accurate than the popular approximator FastSHAP on both tabular data and images.

## 1. Introduction

The application of machine learning algorithms in some domains requires communicating the reasons behind predictions with the aim of building trust in the predictive models and, more importantly, addressing legal and ethical considerations (Lakkaraju et al., 2017; Goodman & Flaxman, 2017). Nevertheless, many state-of-the-art machine learning algorithms result in black-box models, precluding the user's ability to follow the reasoning behind the predictions. Consequently, explainable machine learning methods have gained notable attention as a means to acquire needed explainability without sacrificing performance.

[1]Örebro University, School of Science and Technology, Sweden [2]KTH Royal Institute of Technology, School of Electrical Engineering and Computer Science, Sweden [3]École Polytechnique, IP Paris, France. Correspondence to: Amr Alkhatib <amr.alkhatib@oru.se>.

*Proceedings of the 42nd International Conference on Machine Learning*, Vancouver, Canada. PMLR 267, 2025. Copyright 2025 by the author(s).

Machine learning explanation methods employ a variety of strategies to produce explanations, e.g., the use of local interpretable surrogate models (Ribeiro et al., 2016), generation of counterfactual examples (Karimi et al., 2020; Dandl et al., 2020; Mothilal et al., 2020; Van Looveren & Klaise, 2021; Guo et al., 2021; Guyomard et al., 2022), selection of important features (Chen et al., 2018; Yoon et al., 2019; Jethani et al., 2021), and approximation of Shapley values (Lundberg & Lee, 2017; Lundberg et al., 2020; Frye et al., 2021; Covert & Lee, 2021; Jethani et al., 2022). Methods that generate explanations based on Shapley values are prominent since they offer a unique solution that meets a set of theoretically established, desirable properties. The computation of Shapley values can, however, be computationally expensive. Recent work has therefore focused on reducing the running time (Lundberg & Lee, 2017; Lundberg et al., 2020; Jethani et al., 2022) and enhancing the accuracy of approximations (Frye et al., 2021; Aas et al., 2021; Covert & Lee, 2021; Mitchell et al., 2022; Kolpaczki et al., 2024). However, the Shapley values are computed post-hoc, and hence entail a computational overhead, even when approximated, e.g., as in the case of FastSHAP (Jethani et al., 2022). Generating instance-based explanations or learning a pre-trained explainer always demands further data, time, and resources. Nevertheless, to the best of our knowledge, computing Shapley values as a means to form predictions has not yet been considered.

The main contributions of this study are:

- a novel machine learning method, ViaSHAP, that trains a model to simultaneously provide accurate predictions and Shapley values

- multiple implementations of the proposed method are explored, using both the universal approximation theorem and the Kolmogorov-Arnold representation theorem, which are evaluated through a large-scale empirical investigation

In the following section, we cover fundamental concepts about the Shapley value and, along the way, introduce our notation. Section 3 describes the proposed method. In Section 4, results from a large-scale empirical investigation are presented and discussed. Section 5 provides a brief overview of the related work. Finally, in the concluding

remarks, we summarize the main conclusions and outline directions for future work.

## 2. Preliminaries

### 2.1. The Shapley Value

In game theory, a game in coalitional form is a formal model for a scenario in which players form coalitions, and the game's payoff is shared between the coalition members. A coalitional game focuses on the behavior of the players and typically involves a finite set of players $N = \{1, 2, \ldots, n\}$ (Manea, 2016). A coalitional game also involves a characteristic set function $v : 2^N \to \mathbb{R}$ that assigns a payoff, a real number, to a coalition $S \subseteq N$ such that: $v(\emptyset) = 0$ (Owen, 1995.). Different concepts can be employed to distribute the payoff among the players of a coalitional game to achieve a fair and stable allocation. Such solution concepts include the Core, the Nucleolus, and the Shapley Value (Manea, 2016; Ferguson, 2018).

The Shapley Value is a solution concept that allocates payoffs to the players according to their marginal contributions across possible coalitions. The Shapley value $\phi_i(v)$ of player $i$ in game $v$ is given by (Shapley, 1953):

$$\phi_i(v) = \sum_{S \subseteq N \setminus \{i\}} \frac{|S|!(n - |S| - 1)!}{n!} (v(S \cup \{i\}) - v(S)).$$

The term $\left( \frac{|S|!(n-|S|-1)!}{n!} \right)$ is a combinatorial weighting factor for the different coalitions that can be formed for game $v$. The difference term $(v(S \cup \{i\}) - v(S))$ represents the additional value that player $i$ contributes to the coalition $S$, i.e., the marginal contribution of player $i$.

Given a game $v$, an additive explanation model $\mu$ is an interpretable approximation of $v$ which can be written as (Lundberg & Lee, 2017; Covert & Lee, 2021):

$$\mu(S) = \delta_0(v) + \sum_{i \in S} \delta_i(v),$$

with $\delta_0(v)$ a constant and $\delta_i(v)$ the payoff of player $i$.

$\mu$ is a linear model whose weights are the payoffs of each player. Using the Shapley values as the payoffs is the only solution in the class of additive feature attribution methods that satisfies the following properties (Young, 1985):

*Property* 1. (**Local Accuracy**): the solution matches the prediction of the underlying model:

$$\mu(N) = \sum_{i \in N} \phi_i(v) = v(N)$$

*Property* 2. (**Missingness**): Players without impact on the prediction attributed a value of zero. Let $i \in N$:

$$\forall S \subseteq N \setminus \{i\}, \; v(S) = v(S \cup \{i\}) \Rightarrow \phi_i(v) = 0$$

*Property* 3. (**Consistency**): The Shapley value grows or remains the same if a player's contribution grows or stays the same. Let $v$ and $v'$ two games over $N$, let $i \in N$:

$$\forall S \subseteq N \setminus \{i\}, v(S \cup \{i\}) - v(S)$$
$$\geq v'(S \cup \{i\}) - v'(S)$$
$$\Rightarrow \phi_i(v) \geq \phi_i(v')$$

### 2.2. SHAP

In the context of explainable machine learning, the Shapley value is commonly computed post-hoc to explain the predictions of trained machine learning models. Let $f$ be a trained model whose inputs are defined on $n$ features and whose output $y \in Y \subseteq \mathbb{R}$. We also define a *baseline* or *neutral* instance, noted $\mathbf{0} \in X$. For a given instance $x$, the Shapley value is computed over each feature to explain the difference in output $x \in X$ and the baseline. The baseline may be determined depending on the context, but common examples include the average of all examples in the training set, or one that is commonly used as a threshold (Izzo et al., 2021).

In this context, a coalitional game for $S$ can be derived from the model, where the players are the features, and the value function $v$ represents how the prediction changes as different coalitions of features are masked out. In this game, a player $i$ getting picked for coalition $S$ means that its corresponding feature's value is $x_i$, otherwise it remains at its baseline value $\mathbf{0}_i$.

The Shapley values for this game can then be obtained as the solution of an optimization problem. The objective is to determine a set of values that accurately represent the marginal contributions of each feature while verifying properties 1 through 3. In the literature, they were obtained by minimizing the following weighted least squares loss function (Marichal & Mathonet, 2011; Lundberg & Lee, 2017; Patel et al., 2021):

$$\mathcal{L}(v_x, \mu_x) = \sum_{S \subseteq N} \omega(S) \Big( v_x(S) - \mu_x(S) \Big)^2, \quad (1)$$

where $\omega$ is a weighting kernel, the choice of the kernel can result in a solution equivalent to the Shapley value (Covert & Lee, 2021; Covert et al., 2021). Therefore, (Lundberg & Lee, 2017) proposed the Shapley kernel:

$$\omega_{Shap}(S) = \frac{(n - 1)}{\binom{n}{|S|} \cdot |S| \cdot (n - |S|)}. \quad (2)$$

Note that, for a $d$-dimensional output with $d > 1$, each output is considered as a different unidimensional model. That is, each of the $d$ dimensions will define a different game, and thus a different set of $n$ Shapley values. The explanation of the output is thus an $n \times d$ matrix of Shapley values, providing the contribution of each input feature to each output game. This can trivially be obtained through the same optimization process by stacking $d$ loss functions such as in (1). Thus, we will consider in the following that $y$ be unidimensional unless otherwise specified.

## 2.3. KernelSHAP

Computing the exact Shapley values is a demanding process as it requires evaluating all possible coalitions of feature values. There are $2^n - 1$ possible coalitions for a model with $n$ features, each of which has to be evaluated to determine the features' marginal contributions, which renders the exact computation of Shapley values infeasible for models with a relatively large number of features. Consequently, (Lundberg & Lee, 2017) proposed KernelSHAP as a more feasible method to approximate the Shapley values. KernelSHAP samples a subset of coalitions instead of evaluating all possible coalitions. The explanation model is learned by solving the following optimization problem (Covert & Lee, 2021; Jethani et al., 2022):

$$\phi(v_x) = \underset{\phi_x \in \mathbb{R}^n}{\arg\min} \ \underset{p(S)}{\mathbb{E}} \left[ \left( v_x(S) - v_x(\mathbf{0}) - \mathbf{1}_S^\top \phi_x \right)^2 \right] \quad (3)$$

$$\text{s.t. } \mathbf{1}^\top \phi_x = v_x(S) - v_x(\mathbf{0}) \quad (4)$$

where $\mathbf{1}_S$ is the mask corresponding to $S$, i.e., which takes value 1 for features in $S$ and 0 otherwise, and the distribution $p(S)$ is proportional to the Shapley kernel (2) (Covert & Lee, 2021; Jethani et al., 2022). Equation (4) is referred to as the *efficiency constraint*. Different value functions ($v$) can be applied to marginalize features out, such as:

1. Baseline Removal (Sundararajan & Najmi, 2020): $v_x(S) = f\left(x^S, \mathbb{E}\left[X^{N \setminus S}\right]; \theta\right)$

2. Interventional/Marginal Expectations (Chen et al., 2020): $v_x(S) = \underset{x^S}{\mathbb{E}}\left[f\left(x^S, X^{N \setminus S}; \theta\right)\right]$

3. Observational/Conditional Expectations: $v_x(S) = \underset{x^S}{\mathbb{E}}\left[f\left(X^S; \theta\right) | X^S = x^S\right]$

## 2.4. FastSHAP

Although KernelSHAP provides a practical solution for the Shapley value estimation, the optimization problem 3 must

be solved separately for every prediction. Additionally, KernelSHAP requires many samples to converge to accurate estimations for the Shapley values, and this problem is exacerbated with high dimensional data (Covert & Lee, 2021). Consequently, FastSHAP (Jethani et al., 2022) has been proposed to efficiently learn a parametric Shapley value function and eliminate the need to solve a separate optimization problem for each prediction. The model $\phi_{\text{fast}} : X \to \mathbb{R}^n$, parameterized by $\theta$ is then trained to produce the Shapley value for an input by minimizing the following loss function:

$$\mathcal{L}(\theta) = \underset{q(x)}{\mathbb{E}} \ \underset{p(S)}{\mathbb{E}} \left[ \left( v_x(S) - v_x(\mathbf{0}) - \mathbf{1}_S^\top \phi_{\text{fast}}(x; \theta) \right)^2 \right] \quad (5)$$

where $q(x)$ is the distribution of the input data, and $p(S)$ is proportional to the Shapley kernel defined in (2). In the case of a multidimensional output, a uniform sampling is done over the possible output dimensions.

The accuracy of $\phi_{\text{fast}}$ in approximating the Shapley value depends on the expressiveness of the model class employed as well as the data available for learning $\phi_{\text{fast}}$ as a post-hoc function.

# 3. ViaSHAP

We introduce ViaSHAP, a method that formulates predictions via Shapley values regression. In contrast to the previous approaches, the Shapley values are not computed in a post-hoc setup. Instead, the learning of Shapley values is integrated into the training of the predictive model and exploits every data example in the training data. Moreover, unlike (Chen et al., 2023b), ViaSHAP does not impose a specific neural network design or constrain the explanation to a subset of input features, as is in (Wang et al., 2021). At inference time, the Shapley values are used directly to generate the prediction. The following subsections outline how ViaSHAP is trained to simultaneously produce accurate predictions and their corresponding Shapley values.

## 3.1. Predicting Shapley Values

Let $X \subseteq \mathbb{R}^n$ and $Y \subseteq \mathbb{R}^d$, respectively, the input and output spaces, and $M = \{1, \cdots, d\}$ the set of output dimensions. We define a model $\mathcal{V}ia^{SHAP} : X \to Y$ which, for a given instance $x$, computes both the Shapley values and the predicted output in a single process.

First, $\phi^{\mathcal{V}ia} : X \to \mathbb{R}^{n \times d}$ computes a matrix of values $\phi^{\mathcal{V}ia}(x; \theta)$. Then, ViaSHAP predicts the output vector as $\mathcal{V}ia^{SHAP}(x) = \mathbf{1}^\top \phi^{\mathcal{V}ia}(x; \theta)$ i.e., summing column-wise. A link function $\sigma$ can be applied to accommodate a valid range of outputs $\left(y = \sigma(\mathbf{1}^\top \phi^{\mathcal{V}ia}(x; \theta))\right)$, e.g., the sigmoid function for binary classification or softmax for multi-class classification.

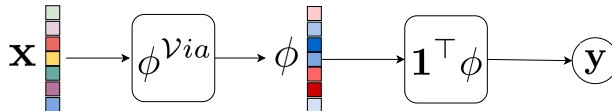

Figure 1. **ViaSHAP** generates predictions by first estimating the Shapley values, whose summation produces the final outcome.

ViaSHAP computes the Shapley values prior to each prediction formulation, as illustrated in Figure 1. Similar to KernelSHAP and FastSHAP (in equation (3) and equation (5)), $\phi^{\mathcal{V}ia}$ is trained by minimizing the weighted least squares loss of the predicted Shapley values, as shown in equation (6). However, no pre-defined black-box model is available beforehand to train the $\phi^{\mathcal{V}ia}$ explainer. Instead, the $\mathcal{V}ia^{SHAP}$ predictor is provided as a black box at each training step.

$$\mathcal{L}_\phi(\theta) = \sum_{x \in X} \sum_{j \in M} \mathop{\mathbb{E}}_{p(S)} \Big[ \Big( \mathcal{V}ia_j^{SHAP}(x^S) - \mathcal{V}ia_j^{SHAP}(\mathbf{0}) \\ - \mathbf{1}_S^\top \phi_j^{\mathcal{V}ia}(x;\theta) \Big)^2 \Big]. \tag{6}$$

Given that the ground truth Shapley values are inaccessible during training, the learning process relies solely on sampling input features, based on the principle that unselected features should be assigned a Shapley value of zero, while the prediction formulated using the selected features should be equal to the sum of their corresponding Shapley values. Since $\phi^{\mathcal{V}ia}$ and $\mathcal{V}ia^{SHAP}$ are essentially the same model, coalition sampling for both functions is performed within the same model but at different locations. For $\mathcal{V}ia^{SHAP}(x^S)$, the sampling occurs on the input features before feeding them to the model. While $\mathbf{1}_S^\top \phi^{\mathcal{V}ia}$ sampling is applied to the predicted Shapley values, given the original set of features as input to the model, as illustrated in Figure 2. In the following, we show that the solution computed by the optimized $\phi^{\mathcal{V}ia}(x;\theta^*)$ function maintains the desirable properties of Shapley values for each output dimension. For ease of notation, we drop the subscript $j$ below and consider one output at a time. All proofs, unless otherwise specified, can be found in the Appendix.

**Lemma 3.1.** $\phi^{\mathcal{V}ia}(x;\theta)$ *satisfies the property of local accuracy wrt* $\mathcal{V}ia^{SHAP}$.

**Lemma 3.2.** *The global minimizer model,* $\phi^{\mathcal{V}ia}(x;\theta^*)$, *of the loss function (6), assigns value zero to features that have no influence on the outcome predicted by* $\mathcal{V}ia^{SHAP}(x)$ *in the distribution* $p(S)$.

**Lemma 3.3.** *Let two ViaSHAP models* $\mathcal{V}$ *and* $\mathcal{V}'$ *whose respective* $\phi^{\mathcal{V}ia}$ *are parameterized by* $\theta^*$ *and* $\theta^{*'}$, *which globally optimize loss function (6) over two possibly differ-*

ent targets $y$ and $y'$. Then, given a feature $i \in N$:

$$\forall S \subseteq N \setminus \{i\}, \mathcal{V}(x^{S \cup \{i\}}) - \mathcal{V}(x^S) \geq \mathcal{V}'(x^{S \cup \{i\}}) - \mathcal{V}'(x^S)$$

$$\Rightarrow \phi_i^{\mathcal{V}ia}(x;\theta^*) \geq \phi_i^{\mathcal{V}ia}(x;\theta^{*'})$$

**Theorem 3.4.** *The global optimizer function* $\phi^{\mathcal{V}ia}(x;\theta^*)$ *computes the exact Shapley values of the predictions of* $\mathcal{V}ia^{SHAP}(x)$.

Theorem 3.4 directly follows from Lemma 3.1, Lemma 3.2, and Lemma 3.3, which demonstrate that $\phi^{\mathcal{V}ia}(x;\theta^*)$ adheres to properties 1 through 3, as well as the fact that Shapley values provide the sole solution for assigning credit to players while satisfying the properties from 1 to 3 (Young, 1985; Lundberg & Lee, 2017).

### 3.2. Predictor Optimization

The parameters of ViaSHAP are optimized with the following dual objective: to learn an optimal function for producing the Shapley values of the predictions and to minimize the prediction loss with respect to the true target. Therefore, the prediction loss is minimized using a function suitable for the specific prediction task, e.g., binary cross-entropy for binary classification or mean squared error for regression tasks. The following presents the loss function for multinomial classification:

$$\mathcal{L}(\theta) = \sum_{x \in X} \sum_{j \in M} \Big( \beta \cdot \mathop{\mathbb{E}}_{p(S)} \Big[ \Big( \mathcal{V}ia_j^{SHAP}(x^S) - \mathcal{V}ia_j^{SHAP}(\mathbf{0}) \\ - \mathbf{1}_S^\top \phi_j^{\mathcal{V}ia}(x;\theta) \Big)^2 \Big] - y_j \log(\hat{y}_j) \Big). \tag{7}$$

where $\beta$ is a predefined scaling hyperparameter and $\hat{y}_j$ is the predicted probability of class $y_j \in Y$ by ViaSHAP. The optimization of ViaSHAP is illustrated in Figure 2 and summarized in Algorithm 1.

The global optimizer of loss function 7 is restricted to predict 0 if all features are marginalized out. However, this approach may not be suitable for all prediction tasks, e.g., regression problems. Therefore, we also propose a relaxed variant of the optimization problem 7, where a ViaSHAP model predicts $y = \mathbf{1}^\top \phi^{\mathcal{V}ia}(x;\theta) + \delta$ (further details on this approach are provided in Appendix G).

### 3.3. ViaSHAP Approximator

According to the universal approximation theorem, a feed-forward network with at least one hidden layer and sufficient units in the hidden layer can approximate any continuous function over a compact input set to an arbitrary degree of accuracy, given a suitable activation function (Hornik et al., 1989; Cybenko, 1989; Hornik, 1991). Consequently, neural networks and multi-layer perceptrons (MLP) can be employed to learn ViaSHAP for prediction tasks where there is

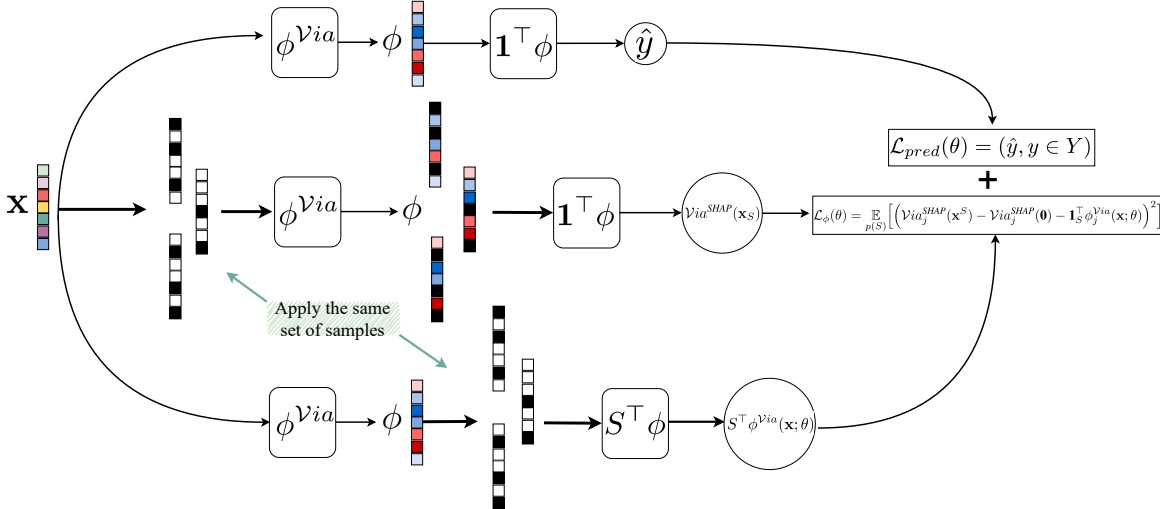

*Figure 2.* **The optimization of ViaSHAP** is conducted using a dual-objective loss function that aims to learn an optimal function for generating the Shapley values while minimizing the prediction loss.

a continuous mapping function from the input dataset to the true targets, which also applies to the true Shapley values as a continuous function.

(Liu et al., 2024) recently proposed Kolmogorov–Arnold Networks (KAN), as an alternative approach to MLPs inspired by the Kolmogorov-Arnold representation theorem. According to the Kolmogorov-Arnold representation theorem, a multivariate continuous function on a bounded domain can be represented by a finite sum of compositions of continuous univariate functions (Kolmogorov, 1956; 1957; Liu et al., 2024), as follows:

$$f(x) = f(x_1, \ldots, x_n) = \sum_{q=1}^{2n+1} \Psi_q \Big( \sum_{p=1}^{n} \psi_{q,p}(x_p) \Big),$$

where $\psi_{q,p} : [0,1] \to \mathbb{R}$ is a univariate function and $\Psi_q : \mathbb{R} \to \mathbb{R}$ is a univariate continuous function. (Liu et al., 2024) defined a KAN layer as a matrix of one-dimensional functions: $\Psi = \{\psi_{q,p}\}$, with $p = 1, 2, \ldots, n_{\text{in}}$ and $q = 1, 2, \ldots, n_{\text{out}}$. Where $n_{\text{in}}$ and $n_{\text{out}}$ represent the dimensions of the layer's input and output, respectively, and $\psi_{q,p}$ are learnable functions parameterized as splines. A KAN network is a composition of L layers stacked together; subsequently, the output of KAN on instance $x$ is given by:

$$y = \text{KAN}(x) = \Psi_{L-1} \circ \Psi_{L-2} \circ \cdots \circ \Psi_1 \circ \Psi_0(x).$$

The degree of each spline and the number of splines for each function are both hyperparameters.

## 4. Empirical Investigation

We evaluate both the predictive performance of ViaSHAP and the feature importance attribution with respect to the

true Shapley values. This section begins with outlining the experimental setup. Then, the predictive performance of ViaSHAP is evaluated. Afterwards, we benchmark the similarity between the feature importance obtained by ViaSHAP and the ground truth Shapley values. We also evaluate the predictive performance and the accuracy of Shapley values on image data. Finally, we summarize the findings of the ablation study.

---

**Algorithm 1** VIASHAP
**Data:** training data $X$, labels $Y$, scalar $\beta$
**Result:** model parameters $\theta$
Initialize $\mathcal{V} : \mathcal{V}ia^{SHAP}(\phi^{\mathcal{V}ia}(x; \theta))$
**while** not converged **do**
    $\mathcal{L} \leftarrow 0$
    **for each** $x \in X$ and $y \in Y$ **do**
        sample $S \sim p(S)$
        $\hat{y} \leftarrow \mathcal{V}(x)$
        $\mathcal{L}_{pred} \leftarrow prediction\ loss(\hat{y}, y)$
        $\mathcal{L}_\phi \leftarrow \Big( \mathcal{V}_y(x^S) - \mathcal{V}_y(\mathbf{0}) - \mathbf{1}_S^\top \phi_y^{\mathcal{V}ia}(x; \theta) \Big)^2$
        $\mathcal{L} \xleftarrow{+} \mathcal{L}_{pred} + \beta \cdot \mathcal{L}_\phi$
    **end**
    Compute gradients $\nabla_\theta \mathcal{L}$
    Update $\theta \leftarrow \theta - \nabla_\theta \mathcal{L}$
**end**

---

### 4.1. Experimental Setup

We employ 25 publicly available datasets in the experiments, each divided into training, validation, and test subsets [1]. The training set is used to train the model, the validation set is

---
[1]The details of the datasets are available in Table 19

used to detect overfitting and determine early stopping, and the test set is used to evaluate the model's performance. All the learning algorithms are trained using default settings without hyperparameter tuning. The training and validation sets are combined into a single training set for algorithms that do not utilize a validation set for performance tracking. During data preprocessing, categorical feature categories are tokenized with numbers starting from one, reserving zero for missing values. We use standard normalization so the feature values are centered around 0. ViaSHAP can be trained using the baseline removal approach or marginal expectations as a value function. However, the baseline removal approach is adopted as the default value function[2]. We experimented with four different implementations of ViaSHAP, using Kolmogorov–Arnold Networks (KANs) and feedforward neural networks[3]:

**1- $KAN^{\mathcal{V}ia}$**: Based on the method proposed by Liu et al. (2024) using a computationally efficient implementation[4]. Uses spline basis functions and consists of an input layer, two hidden layers, and an output layer. Layer dimensions: Input layer maps $n$ features to 64 dimensions, the first hidden layer to 128 dimensions, the second hidden layer to 64 dimensions, and the output layer to $n\times$ (number of classes).

**2- $KAN_{\varrho}^{\mathcal{V}ia}$**: Replaces the spline basis in the original KANs with Radial Basis Functions (RBFs)[5]. The architecture matches that of $KAN^{\mathcal{V}ia}$.

**3- $MLP^{\mathcal{V}ia}$**: A multi-layer perceptron (MLP) with identical input and output dimensions per layer as the KAN-based implementations. Incorporates batch normalization after each layer and uses ReLU activation functions.

**4- $MLP_{\theta}^{\mathcal{V}ia}$**: Similar to $MLP^{\mathcal{V}ia}$, but the number of units in the hidden layers is raised to match the total number of parameters in the $KAN^{\mathcal{V}ia}$ models, as $KAN^{\mathcal{V}ia}$ always results in models with a greater number of parameters compared to the remaining implementations.

The four implementations were trained with the $\beta$ of (7) set to 10 and used 32 sampled coalitions per instance. The above hyperparameters were determined in a quasi-random manner.

For the evaluation of the predictive performance, the four ViaSHAP approximators ($KAN^{\mathcal{V}ia}$, $KAN_{\varrho}^{\mathcal{V}ia}$, $MLP^{\mathcal{V}ia}$, and $MLP_{\theta}^{\mathcal{V}ia}$) are compared against XGBoost, Random Forests, and TabNet (Arik & Pfister, 2021). All the compared algorithms are trained using the default hyperparameters set-

tings without tuning, as it has been shown by (Shwartz-Ziv & Armon, 2022) that deep models typically require more extensive tuning on each tabular dataset to match the performance of tree ensemble models, e.g., XGBoost. If the model's performance varies with different random seeds, it will be trained using five different seeds, and the average result will be reported alongside the standard deviation. In binary classification tasks with imbalanced training data, the minority class in the training subset is randomly over-sampled to match the size of the majority class, a common strategy to address highly imbalanced data (Koziarski et al., 2017; Huang et al., 2022). On the other hand, no oversampling is applied to multinomial classification datasets. The area under the ROC curve (AUC) is used for measuring predictive performance since it is invariant to classification thresholds. For multinomial classification, we compute the AUC for each class versus the rest and then weighting it by the class support. If two algorithms achieve the same AUC score, the model with a smaller standard deviation across five repetitions with different random seeds is considered better. For the explainability evaluation, we generate ground truth Shapley values by running the unbiased KernelSHAP (Covert & Lee, 2021) until convergence. It has been demonstrated that the unbiased KernelSHAP will converge to the true Shapley values when given a sufficiently large number of data samples (Covert & Lee, 2021; Jethani et al., 2022).[6] We measure the similarity of the approximated Shapley values by ViaSHAP to the ground truth using cosine similarity, Spearman rank correlation (Spearman, 1904), and the coefficient of determination ($R^2$), where cosine similarity measures the alignment between two explanation vectors, while Spearman rank correlation measures the consistency in feature rankings. The results are presented as mean values with standard deviations across all data instances in the test set.

For image experiments, we use the CIFAR-10 dataset (Krizhevsky et al., 2014). We provide three ViaSHAP implementations for image classification: $ResNet50^{\mathcal{V}ia}$, $ResNet18^{\mathcal{V}ia}$, and $U\text{-}Net^{\mathcal{V}ia}$ based on ResNet50, ResNet18 (He et al., 2016), and U-Net (Ronneberger et al., 2015), respectively. The accuracy of the Shapley values is estimated by measuring the effect of excluding and including the top important features on the prediction, similar to the approach followed by (Jethani et al., 2022).

### 4.2. Predictive Performance Evaluation

We evaluated the performance of the seven algorithms ($KAN^{\mathcal{V}ia}$, $KAN_{\varrho}^{\mathcal{V}ia}$, $MLP^{\mathcal{V}ia}$, $MLP_{\theta}^{\mathcal{V}ia}$, TabNet, Random Forests, and XGBoost) across the 25 datasets, with detailed results presented in Table 1. The results show that

---

[2]Further details regarding the marginal expectations approach are provided in Appendix H.

[3]The source code is available here: https://github.com/amrmalkhatib/ViaSHAP

[4]https://github.com/Blealtan/efficient-kan

[5]https://github.com/ZiyaoLi/fast-kan

[6]https://github.com/iancovert/shapley-regression

$KAN^{\mathcal{V}ia}$ obtains the highest average rank with respect to AUC. $KAN_{\varrho}^{\mathcal{V}ia}$ came in second place, closely followed by XGBoost, with only a slight difference between them. We employed the Friedman test (Friedman, 1939) to determine whether the observed performance differences are statistically significant. We tested the null hypothesis that there is no difference in predictive performance. The Friedman test allowed the rejection of the null hypothesis, indicating that there is indeed a difference in predictive performance, as measured by AUC, at the 0.05 significance level. Subsequently, the post-hoc Nemenyi (Nemenyi, 1963) test was applied to identify which pairwise differences are significant, again at the 0.05 significance level. The results of the post-hoc test, summarized in Figure 3, indicate that the differences between ViaSHAP using KAN implementations and the tree ensemble models, i.e., XGBoost and Random Forests, are statistically insignificant, given the sample size of 25 datasets. However, the differences in predictive performance between $KAN^{\mathcal{V}ia}$ and MLP variants ($MLP^{\mathcal{V}ia}$ and $MLP_{\theta}^{\mathcal{V}ia}$) are statistically significant. It is also noticeable that the MLP variants of ViaSHAP underperform compared to all other competitors, even when the MLP models have an equivalent number of parameters to $KAN^{\mathcal{V}ia}$. We also evaluated the impact of incorporating Shapley loss on the predictive performance of a KAN model by comparing $KAN^{\mathcal{V}ia}$ to an identical KAN classifier trained without the Shapley loss. The results show that $KAN^{\mathcal{V}ia}$ significantly outperforms identical KAN architecture that is not optimized to compute Shapley values. The detailed results are available in Appendix I.

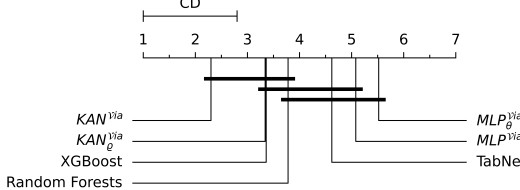

Figure 3. **The average rank** of the 7 predictors on the 25 datasets with respect to the AUC (the lower rank is better). The critical difference (CD) is the largest statistically insignificant difference.

## 4.3. Explainability Evaluation

The explainability of the various ViaSHAP implementations is evaluated by measuring the similarity of ViaSHAP's Shapley values ($\phi^{\mathcal{V}ia}(x;\theta)$ to the ground truth Shapley values ($\phi$), computed by the unbiased KernelSHAP, as discussed in Subsection 4.1, taking ViaSHAP as the black-box model. We present results for models trained with the default values for the hyperparameters. The effect of these settings are further investigated in the ablation study.

The evaluation of the alignment between $\phi^{\mathcal{V}ia}(x;\theta)$ and $\phi$ using cosine similarity generally shows a high degree of sim-

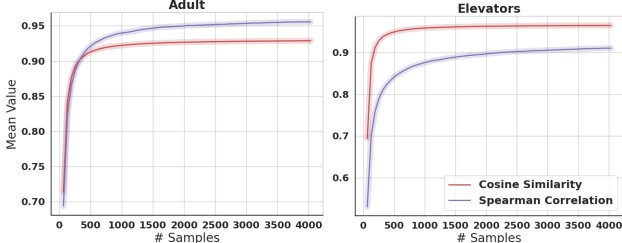

Figure 4. **The similarity between $KAN^{\mathcal{V}ia}$ and KernelSHAP's approximations.** KernelSHAP initially provides approximations that differ remarkably from the values of ViaSHAP. However, as KernelSHAP refines its approximations with more samples, the similarity to ViaSHAP's values grows.

ilarity between the generated Shapley values and the ground truth as illustrated in Figure 4. The ranking of the compared implementations of ViaSHAP with respect to their cosine similarity to the ground truth Shapley values shows that $MLP_{\theta}^{\mathcal{V}ia}$ is ranked first, followed by $KAN^{\mathcal{V}ia}$, $KAN_{\varrho}^{\mathcal{V}ia}$, and $MLP^{\mathcal{V}ia}$, respectively. However, the Friedman test does not indicate any significant difference between the different implementations of ViaSHAP. At the same time, the results of ranking the four implementations of ViaSHAP based on their Spearman rank correlation with the ground truth Shapley values reveal that $KAN^{\mathcal{V}ia}$ ranks first, followed by a tie for second place between $KAN_{\varrho}^{\mathcal{V}ia}$ and $MLP_{\theta}^{\mathcal{V}ia}$, and $MLP^{\mathcal{V}ia}$ placing last. In order to find out whether the differences are significant, the Friedman test is applied once again, which allows for the rejection of the null hypothesis, indicating that there is indeed a difference between the compared models in their $\phi^{\mathcal{V}ia}(x;\theta)$ correlations to the ground truth $\phi$, at 0.05 significance level. The post-hoc Nemenyi test, at 0.05 level, indicates that differences between $MLP^{\mathcal{V}ia}$ and the remaining models are significant, as summarized in Figure 6. Overall, $KAN^{\mathcal{V}ia}$ is found to be a relatively stable approximator across the 25 datasets when both similarity metrics (cosine similarity and Spearman rank correlation) are considered. Detailed results can be found in Tables 2 and 3 in Appendix E. We also compare the accuracy of the Shapley values generated by $KAN^{\mathcal{V}ia}$ to those produced by FastSHAP, with $KAN^{\mathcal{V}ia}$ utilized as black-box within FastSHAP. The results in Appendix K show that $KAN^{\mathcal{V}ia}$ significantly outperforms FastSHAP in terms of similarity to the ground truth.

## 4.4. Image Experiments

We evaluated the predictive performance of $ResNet50^{\mathcal{V}ia}$, $ResNet18^{\mathcal{V}ia}$, and $U\text{-}Net^{\mathcal{V}ia}$ on the CIFAR-10 dataset. All models were trained from scratch (without transfer learning). The results, summarized in Table 4, demonstrate that ViaSHAP can perform accurately in image classification tasks. We also compared the accuracy of the explanations

obtained by ViaSHAP implementations with those obtained by FastSHAP (where ViaSHAP models were treated as black boxes). The results in Table 5 and Figure 8 show that ViaSHAP models consistently provides more accurate Shapley value approximations than the explanations obtained using FastSHAP. Figure 5 provides two examples showing the explanations produced by $ResNet18^{Via}$ and FastSHAP. The experiment details can be found in Appendix F.

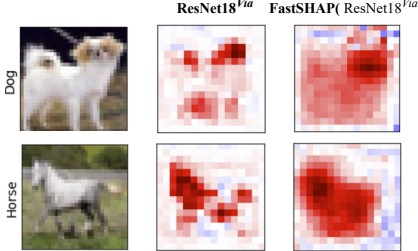

*Figure 5.* The explanations for the predicted class generated by ViaSHAP and FastSHAP using two randomly selected images from the CIFAR-10 dataset.

### 4.5. Ablation Study

The ablation study was conducted after the empirical evaluation to ensure that no prior knowledge of the data or models influenced the experimental setup. The detailed results of the ablation study are provided in Appendex J. We began by examining the effect of $\beta$ on both predictive performance and the accuracy of Shapley values. The results demonstrate that the predictive performance remains robust to changes in $\beta$, unless $\beta$ is raised to an exceptionally large value, e.g., $\geq$200-fold. A more remarkable observation is that the accuracy of the computed Shapley values improves as $\beta$ grows without sacrificing predictive performance. However, the model fails to learn properly with substantially large $\beta$. Afterwards, we evaluated the effect of the number of sampled coalitions. The results indicate that the number of samples has little impact on predictive performance and the accuracy of Shapley values, especially if compared to the impact of $\beta$. We also study the effect of a link function on both predictive performance and the accuracy of Shapley values of ViaSHAP. The results show that removing the link function significantly improves the accuracy of the Shapley values while maintaining the high predictive performance. Then, we assessed the impact of the efficiency constraint. The results indicate that the efficiency constraint has no significant impact on the predictive performance or the accuracy of the explanations of ViaSHAP. Finally, we examined the impact of $\beta$ on the progression of training and validation loss during training. The results indicate that ViaSHAP tends to require a longer time to converge as $\beta$ values increase.

## 5. Related Work

In addition to KernelSHAP and the real-time method Fast-SHAP, alternative approaches have been proposed to reduce the time required for Shapley value approximation. Methods that exploit specific properties of the explained model can provide faster computations, e.g., TreeSHAP (Lundberg et al., 2020) and DASP (Ancona et al., 2019), while others limit the scope to specific problems, e.g., image classifications or text classification (Chen et al., 2019; Teneggi et al., 2022). Additionally, directions to improve Shapley value approximation by enhancing data sampling have also been explored (Frye et al., 2021; Aas et al., 2021; Covert et al., 2021; Mitchell et al., 2022; Chen et al., 2023a; Kolpaczki et al., 2024). Nevertheless, traditional methods for computing Shapley values have typically been considered post-hoc solutions for explaining predictions, requiring additional time, data, and resources to generate explanations. In contrast, ViaSHAP computes Shapley values during inference, eliminating the need for a separate post-hoc explainer.

Research on generating explanations using pre-trained models has explored several approaches. (Chen et al., 2018), (Yoon et al., 2019), and (Jethani et al., 2021) trained models for important features selection. (Schwab & Karlen, 2019) trained a model to estimate the influence of different inputs on the predicted outcome. (Situ et al., 2021) proposed to distill any explanation algorithm for text classification. Pretrained explainers, similar to other post-hoc methods, require further resources for training, and the fidelity of their explanations to the underlying black-box model can vary.

Many approaches for creating explainable neural networks have been proposed. Such approaches not only generate predictions but also include an integrated component that provides explanations, which is trained alongside the predictor (Lei et al., 2016; Alvarez Melis & Jaakkola, 2018; Guo et al., 2021; Al-Shedivat et al., 2022; Sawada & Nakamura, 2022; Guyomard et al., 2022). Explainable graph neural networks (GNNs) have also been studied for graph-structured data, which typically exploit the internal properties of their models to generate explanations, e.g., the similarity between nodes (Dai & Wang, 2021), finding patterns and common graph structures(Feng et al., 2022; Zhang et al., 2022; Cui et al., 2022), or analyzing the behavior of different components of the GNN (Xuanyuan et al., 2023). GNNs have also been employed to learn explainable models for data types beyond graphs, e.g., tabular data (Alkhatib et al., 2024; Alkhatib & Boström, 2025) and images (Chaidos et al., 2025). However, explanations generated by explainable neural networks do not always correspond to Shapley values, in contrast to ViaSHAP. Moreover, the explanations lack fidelity guarantees and do not elaborate on how exactly the predictions are computed, whereas ViaSHAP generates predictions directly from their Shapley values.

## 6. Concluding Remarks

We have proposed ViaSHAP, an algorithm that computes Shapley values during inference. We evaluated the performance of ViaSHAP using implementations based on the universal approximation theorem and the Kolmogorov-Arnold representation theorem. We have presented results from a large-scale empirical investigation, in which ViaSHAP was evaluated with respect to predictive performance and the accuracy of the computed Shapley values. ViaSHAP using Kolmogorov-Arnold Networks showed superior predictive performance compared to multi-layer perceptron variants while competing favorably with state-of-the-art algorithms for tabular data XGBoost and Random Forests. ViaSHAP estimations showed a high similarity to the ground truth Shapley values, which can be controlled through the hyperparameters. One natural direction for future research is to implement ViaSHAP using state-of-the-art algorithms. Another direction is to use ViaSHAP to study possible adversarial attacks on a predictive model.

## Acknowledgements

This work was partially supported by the Wallenberg AI, Autonomous Systems and Software Program (WASP) funded by the Knut and Alice Wallenberg Foundation. The computations (on GPUs) were enabled by resources provided by the National Academic Infrastructure for Supercomputing in Sweden (NAISS), partially funded by the Swedish Research Council through grant agreement no. 2022-06725.

## Impact Statement

This paper presents work whose goal is to advance the field of Explainable Machine Learning. The proposed method could be deployed across a variety of domains where interpretability is crucial. Unlike traditional explanation methods that add latency and can be harder to scale, our approach provides explainable models that are not restricted by a specific design. Furthermore, improvements in explanation accuracy could enhance users' trust and understanding of the employed model. The availability of accurate real-time explanations through one model alongside predictions could also enable its use in resource-constrained settings. However, there are also potential risks, for example, accessible and fast explanations could lead to superficial trust in model predictions, specifically if the explanations are misinterpreted. Moreover, even when accurate in the Shapley value sense, explanations might still fail to capture causal or domain-relevant reasoning and potentially mislead users. More importantly, malicious actors might misuse the availability of explanations to reverse-engineer models or identify vulnerabilities for certain adversarial attacks.

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

## A. Proof of Lemma 3.1

By definition of $\mathcal{V}ia^{\text{SHAP}}$:

$$\mathcal{V}ia^{\text{SHAP}}(x) = \mathbf{1}^\top \phi^{\mathcal{V}ia}(x;\theta) = \sum_{i \in N} \phi_i^{\mathcal{V}ia}(x;\theta)$$

This is the definition of local accuracy for the game $v : S \mapsto \mathcal{V}ia^{\text{SHAP}}(x^S)$.

## B. Proof of Lemma 3.2

Assume that the global minimizer $\phi^{\mathcal{V}ia}(x;\theta^*)$ of the loss function (6) does not satisfy the missingness property, i.e., there exists a feature $i$ that has no impact on the prediction:

$$\mathcal{V}ia^{\text{SHAP}}(x^{S \cup \{i\}}) = \mathcal{V}ia^{\text{SHAP}}(x^S), \ \forall S \subseteq N \setminus \{i\} \tag{8}$$

However, the Shapley value $\phi_i$ assigned by $\phi^{\mathcal{V}ia}(x;\theta^*)$ is not zero ($\phi_i \neq 0$).

We recall the optimized loss function:

$$\mathcal{L}_\phi(\theta) = \sum_{x \in X} \mathbb{E}_{p(S)} \left[ \left( \mathcal{V}ia^{\text{SHAP}}(x^S) - \mathcal{V}ia^{\text{SHAP}}(\mathbf{0}) - \mathbf{1}_S^\top \phi^{\mathcal{V}ia}(x;\theta) \right)^2 \right],$$

This loss is non-negative, and is thus minimized for a value of 0, implying all terms in the expectancy are equal to 0. In particular, for any set $S \subseteq N \setminus \{i\}$, we have:

$$
\begin{aligned}
0 = &\begin{cases} \mathcal{V}ia^{\text{SHAP}}(x^{S \cup \{i\}}) - \mathcal{V}ia^{\text{SHAP}}(\mathbf{0}) - \mathbf{1}_{S \cup \{i\}}^\top \phi^{\mathcal{V}ia}(x;\theta) \\ \mathcal{V}ia^{\text{SHAP}}(x^S) - \mathcal{V}ia^{\text{SHAP}}(\mathbf{0}) - \mathbf{1}_S^\top \phi^{\mathcal{V}ia}(x;\theta) \end{cases} \\
\Rightarrow & \ \mathcal{V}ia^{\text{SHAP}}(x^{S \cup \{i\}}) - \mathbf{1}_{S \cup \{i\}}^\top \phi^{\mathcal{V}ia}(x;\theta) = \mathcal{V}ia^{\text{SHAP}}(x^S) - \mathbf{1}_S^\top \phi^{\mathcal{V}ia}(x;\theta) \\
\Rightarrow & \ \mathcal{V}ia^{\text{SHAP}}(x^S) - \mathbf{1}_{S \cup \{i\}}^\top \phi^{\mathcal{V}ia}(x;\theta) = \mathcal{V}ia^{\text{SHAP}}(x^S) - \mathbf{1}_S^\top \phi^{\mathcal{V}ia}(x;\theta) \\
\Rightarrow & \sum_{j \in S \cup \{i\}} \phi_j^{\mathcal{V}ia}(x;\theta^*) = \sum_{j \in S} \phi_j^{\mathcal{V}ia}(x;\theta^*) \\
\Rightarrow & \ \phi_i^{\mathcal{V}ia}(x;\theta^*) = 0
\end{aligned}
$$

In practice, it is unlikely for a loss to exactly reach its global optimum. Instead, it approximates it. We assume here that the loss has reached a value $\epsilon^2$ for an $\epsilon \geq 0$. We propose an upper bound on $\phi_i^{\mathcal{V}ia}(x;\theta)$ conditioned on $\epsilon$.

Since the loss is composed only of non-negative terms, this means that:

$$\forall S \subseteq N, \left( \mathcal{V}ia^{\text{SHAP}}(x^S) - \mathcal{V}ia^{\text{SHAP}}(\mathbf{0}) - \mathbf{1}_S^\top \phi^{\mathcal{V}ia}(x;\theta) \right)^2 \leq \epsilon^2$$

$$\Rightarrow \left| \mathcal{V}ia^{\text{SHAP}}(x^S) - \mathcal{V}ia^{\text{SHAP}}(\mathbf{0}) - \mathbf{1}_S^\top \phi^{\mathcal{V}ia}(x;\theta) \right| \leq \epsilon$$

$$\epsilon \geq \begin{cases} \left| \mathcal{V}ia^{SHAP}(x^{S \cup \{i\}}) - \mathcal{V}ia^{SHAP}(\mathbf{0}) - \mathbf{1}_{S \cup \{i\}}^{\top} \phi^{\mathcal{V}ia}(x; \theta) \right| \\ \left| \mathcal{V}ia^{SHAP}(x^S) - \mathcal{V}ia^{SHAP}(\mathbf{0}) - \mathbf{1}_S^{\top} \phi^{\mathcal{V}ia}(x; \theta) \right| \end{cases}$$

$$\Rightarrow \left| \mathcal{V}ia^{SHAP}(x^{S \cup \{i\}}) - \mathcal{V}ia^{SHAP}(\mathbf{0}) - \mathbf{1}_{S \cup \{i\}}^{\top} \phi^{\mathcal{V}ia}(x; \theta) - \mathcal{V}ia^{SHAP}(x^S) + \mathcal{V}ia^{SHAP}(\mathbf{0}) + \mathbf{1}_S^{\top} \phi^{\mathcal{V}ia}(x; \theta) \right| \leq 2\epsilon$$

$$\Rightarrow \left| \mathcal{V}ia^{SHAP}(x^S) - \mathbf{1}_{S \cup \{i\}}^{\top} \phi^{\mathcal{V}ia}(x; \theta) - \mathcal{V}ia^{SHAP}(x^S) + \mathbf{1}_S^{\top} \phi^{\mathcal{V}ia}(x; \theta) \right| \leq 2\epsilon \text{ by (8)}$$

$$\Rightarrow \left| \sum_{j \in S \cup \{i\}} \phi_j^{\mathcal{V}ia}(x; \theta) - \sum_{j \in S} \phi_j^{\mathcal{V}ia}(x; \theta) \right| \leq 2\epsilon$$

$$\Rightarrow \left| \phi_i^{\mathcal{V}ia}(x; \theta) \right| \leq 2\epsilon$$

$$\Rightarrow \left| \phi_i^{\mathcal{V}ia}(x; \theta) \right| \leq 2\mathcal{L}_\phi(\theta)$$

Thus, as the loss function converges to 0, so does the importance attributed to features with no influence on the outcome.

## C. Proof of Lemma 3.3

Since both $\mathcal{V}$ and $\mathcal{V}'$ optimize their respective targets, they satisfy efficiency, i.e.:

$$\forall S \subseteq N, \ \ \mathcal{V}(x^S) = \mathbf{1}_S^{\top} \phi^{\mathcal{V}ia}(x; \theta^*); \ \mathcal{V}'(x^S) = \mathbf{1}_S^{\top} \phi'^{\mathcal{V}ia}(x; \theta^{*'}) \tag{9}$$

Then:

$$\forall S \subseteq N \setminus \{i\},$$
$$\mathcal{V}(x^{S \cup \{i\}}) - \mathcal{V}(x^S) \geq \mathcal{V}'(x^{S \cup \{i\}}) - \mathcal{V}'(x^S)$$
$$\Rightarrow \sum_{j \in S \cup \{i\}} \phi_j^{\mathcal{V}ia}(x; \theta^*) - \sum_{j \in S} \phi_j^{\mathcal{V}ia}(x; \theta^*) \geq \sum_{j \in S \cup \{i\}} \phi_j^{\mathcal{V}ia}(x; \theta^{*'}) - \sum_{j \in S} \phi_j^{\mathcal{V}ia}(x; \theta^{*'})$$
$$\Rightarrow \phi_i^{\mathcal{V}ia}(x; \theta^*) \geq \phi_i^{\mathcal{V}ia}(x; \theta^{*'})$$

In the same way as for the Lemma 2, the proof assumes perfect minimization of the loss. Thus, we propose a relaxed variant, where the loss term $\mathcal{L}_\phi(\theta)$ was minimized down to $\epsilon^2$ with $\epsilon \geq 0$. Thus, following similar reasoning as in the proof of Lemma 2, we have that $\forall S$:

$$\left| \mathcal{V}ia^{SHAP}(x^S) - \mathcal{V}ia^{SHAP}(\mathbf{0}) - \mathbf{1}_S^{\top} \phi^{\mathcal{V}ia}(x; \theta) \right| \leq \epsilon$$

We also have:

$$\left| \mathcal{V}ia^{SHAP}(x^S) - \mathbf{1}_S^{\top} \phi^{\mathcal{V}ia}(x; \theta) \right| = \left| \mathcal{V}ia^{SHAP}(x^S) - \mathbf{1}_S^{\top} \phi^{\mathcal{V}ia}(x; \theta) - \mathcal{V}ia^{SHAP}(\mathbf{0}) + \mathcal{V}ia^{SHAP}(\mathbf{0}) \right|$$

By the triangle inequality on the right-hand side:

$$\left| \mathcal{V}ia^{SHAP}(x^S) - \mathbf{1}_S^{\top} \phi^{\mathcal{V}ia}(x; \theta) \right| \leq \left| \mathcal{V}ia^{SHAP}(x^S) - \mathbf{1}_S^{\top} \phi^{\mathcal{V}ia}(x; \theta) - \mathcal{V}ia^{SHAP}(\mathbf{0}) \right| + \left| \mathcal{V}ia^{SHAP}(\mathbf{0}) \right|$$

But observe that all features in $\mathbf{0}$ are non-contributive since, $\forall S \subseteq N, \mathbf{0}^S = \mathbf{0}$ by definition of the masking operation. Thus, by the bound found in Lemma 2: $\forall i \in N, \left| \phi_i(\mathbf{0}, \theta) \right| \leq 2\epsilon$. Thus $\left| \mathcal{V}ia^{SHAP}(\mathbf{0}) \right| \leq 2n\epsilon$.

Thus:

$$\left| \mathcal{V}ia^{SHAP}(x^S) - \mathbf{1}_S^\top \phi^{\mathcal{V}ia}(x;\theta) - \mathcal{V}ia^{SHAP}(\mathbf{0}) \right| + \left| \mathcal{V}ia^{SHAP}(\mathbf{0}) \right| \le \epsilon + 2n\epsilon$$

and we thus derive the following upper bound on the $\phi_i$-wise error as:

$$\left| \mathcal{V}ia^{SHAP}(x^S) - \mathbf{1}_S^\top \phi^{\mathcal{V}ia}(x;\theta) \right| \le \epsilon(2n+1).$$

## D. Predictive Performance

We evaluated the performance of the four variants of ViaSHAP implementations mentioned in the experimental setup, i.e., $KAN^{\mathcal{V}ia}$, $KAN_\varrho^{\mathcal{V}ia}$, $MLP^{\mathcal{V}ia}$, and $MLP_\theta^{\mathcal{V}ia}$, are compared to the following algorithms for structured data: Random Forests, XGBoost, and TabNet, where Random Forests and XGBoost result in black-box models, while TabNet is explainable by visualizing feature selection masks that highlight important features. The predictive performance evaluation is conducted using 25 datasets. The results show that $KAN^{\mathcal{V}ia}$ comes in first place as the best-performing classifier, followed by XGBoost and $KAN_\varrho^{\mathcal{V}ia}$, based on AUC values.

The Friedman test confirmed that the differences in predictive performance are statistically significant at the 0.05 level. A subsequent post-hoc Nemenyi test revealed that while the differences between KAN-based implementations and tree ensemble models (XGBoost and Random Forests) are statistically insignificant, the performance differences between $KAN^{\mathcal{V}ia}$ and MLP variants are significant. Moreover, the differences between KANVia and TabNet are also statistically significant. The ranking of the seven models on the 25 datasets and the results of the post-hoc Nemenyi test are illustrated in Figure 3. The detailed results on the 25 datasets are shown in Table 1.

While the MLP implementations of ViaSHAP significantly underperformed compared to the KAN variants, their performance can still be enhanced by using, for instance, deeper and more expressive models, particularly for datasets with high dimensionality and large training sets. However, we defer the task of improving MLP-based ViaSHAP implementations to future work, as the core concept of ViaSHAP can be integrated with any deep learning model. More importantly, ViaSHAP is not limited to structured data and can be incorporated easily into the training loop of models in computer vision and natural language processing.

*Table 1.* The AUC of $KAN^{\mathcal{V}ia}$, $KAN_{\varrho}^{\mathcal{V}ia}$, $MLP^{\mathcal{V}ia}$, and $MLP_{\theta}^{\mathcal{V}ia}$, TabNet, Random Forests, and XGBoost. The best-performing model is colored in light green, and the second best-performing is colored in light blue.

| Dataset | $KAN^{\mathcal{V}ia}$ | $KAN_{\varrho}^{\mathcal{V}ia}$ | $MLP^{\mathcal{V}ia}$ | $MLP_{\theta}^{\mathcal{V}ia}$ | TabNet | Random Forests | XGBoost |
|---|---|---|---|---|---|---|---|
| Abalone | 0.87 ± 0.003 | 0.871 ± 0.003 | 0.877 ± 0.003 | 0.878 ± 0.004 | 0.873 ± 0.01 | 0.875 ± 0.001 | 0.868 |
| Ada Prior | 0.89 ± 0.005 | 0.899 ± 0.002 | 0.887 ± 0.005 | 0.878 ± 0.005 | 0.82 ± 0.062 | 0.885 ± 0.001 | 0.887 |
| Adult | 0.914 ± 0.003 | 0.916 ± 0.001 | 0.91 ± 0.007 | 0.912 ± 0.006 | 0.911 ± 0.001 | 0.913 ± 0.001 | 0.928 |
| Bank32nh | 0.878 ± 0.001 | 0.876 ± 0.005 | 0.873 ± 0.002 | 0.872 ± 0.005 | 0.862 ± 0.005 | 0.875 ± 0.003 | 0.875 |
| Electricity | 0.93 ± 0.004 | 0.92 ± 0.004 | 0.864 ± 0.003 | 0.87 ± 0.008 | 0.88 ± 0.009 | 0.97 ± 0.0004 | 0.973 |
| Elevators | 0.935 ± 0.002 | 0.939 ± 0.001 | 0.922 ± 0.031 | 0.938 ± 0.003 | 0.942 ± 0.002 | 0.909 ± 0.001 | 0.935 |
| Fars | 0.96 ± 0.0003 | 0.955 ± 0.003 | 0.953 ± 0.001 | 0.952 ± 0.002 | 0.953 ± 0.0008 | 0.951 ± 0.0003 | 0.963 |
| Helena | 0.884 ± 0.0001 | 0.881 ± 0.0005 | 0.88 ± 0.002 | 0.881 ± 0.002 | 0.883 ± 0.002 | 0.855 ± 0.001 | 0.874 |
| Heloc | 0.788 ± 0.002 | 0.784 ± 0.002 | 0.775 ± 0.006 | 0.773 ± 0.004 | 0.774 ± 0.007 | 0.779 ± 0.001 | 0.767 |
| Higgs | 0.801 ± 0.001 | 0.805 ± 0.003 | 0.786 ± 0.006 | 0.786 ± 0.005 | 0.801 ± 0.003 | 0.79 ± 0.001 | 0.799 |
| LHC Identify Jet | 0.944 ± 0.0001 | 0.942 ± 0.0003 | 0.931 ± 0.001 | 0.932 ± 0.002 | 0.942 ± 0.0008 | 0.935 ± 0.0002 | 0.941 |
| House 16H | 0.949 ± 0.0007 | 0.944 ± 0.001 | 0.929 ± 0.005 | 0.932 ± 0.01 | 0.94 ± 0.003 | 0.955 ± 0.0004 | 0.952 |
| Indian Pines | 0.985 ± 0.0004 | 0.885 ± 0.086 | 0.918 ± 0.014 | 0.692 ± 0.191 | 0.983 ± 0.003 | 0.979 ± 0.0005 | 0.987 |
| Jannis | 0.864 ± 0.001 | 0.861 ± 0.0005 | 0.571 ± 0.159 | 0.569 ± 0.151 | 0.864 ± 0.003 | 0.861 ± 0.0002 | 0.871 |
| JM1 | 0.732 ± 0.003 | 0.74 ± 0.006 | 0.719 ± 0.01 | 0.717 ± 0.005 | 0.726 ± 0.004 | 0.746 ± 0.004 | 0.708 |
| Magic Telescope | 0.929 ± 0.001 | 0.927 ± 0.001 | 0.917 ± 0.004 | 0.917 ± 0.004 | 0.929 ± 0.001 | 0.933 ± 0.0005 | 0.935 |
| MC1 | 0.94 ± 0.003 | 0.93 ± 0.013 | 0.909 ± 0.014 | 0.896 ± 0.01 | 0.916 ± 0.021 | 0.845 ± 0.002 | 0.934 |
| Microaggregation2 | 0.783 ± 0.002 | 0.765 ± 0.003 | 0.746 ± 0.005 | 0.733 ± 0.029 | 0.759 ± 0.014 | 0.768 ± 0.0008 | 0.781 |
| Mozilla4 | 0.968 ± 0.0008 | 0.962 ± 0.001 | 0.948 ± 0.002 | 0.947 ± 0.001 | 0.96 ± 0.004 | 0.988 ± 0.0007 | 0.989 |
| Satellite | 0.996 ± 0.001 | 0.992 ± 0.002 | 0.997 ± 0.001 | 0.996 ± 0.001 | 0.991 ± 0.002 | 0.998 ± 0.0004 | 0.992 |
| PC2 | 0.827 ± 0.009 | 0.818 ± 0.032 | 0.685 ± 0.088 | 0.662 ± 0.016 | 0.631 ± 0.16 | 0.631 ± 0.05 | 0.646 |
| Phonemes | 0.946 ± 0.003 | 0.936 ± 0.004 | 0.904 ± 0.004 | 0.894 ± 0.032 | 0.918 ± 0.014 | 0.965 ± 0.002 | 0.951 |
| Pollen | 0.515 ± 0.006 | 0.496 ± 0.007 | 0.504 ± 0.007 | 0.5 ± 0.014 | 0.493 ± 0.012 | 0.485 ± 0.006 | 0.475 |
| Telco Customer Churn | 0.854 ± 0.003 | 0.852 ± 0.003 | 0.843 ± 0.003 | 0.839 ± 0.004 | 0.832 ± 0.004 | 0.847 ± 0.002 | 0.846 |
| 1st order theorem proving | 0.822 ± 0.002 | 0.695 ± 0.024 | 0.737 ± 0.013 | 0.64 ± 0.093 | 0.727 ± 0.01 | 0.855 ± 0.001 | 0.858 |

# E. Explanations Accuracy Evaluation

The explainability of the four implementations of ViaSHAP, based on MLP and KAN, were evaluated by comparing their Shapley values ($\phi^{\text{Via}}(x;\theta)$) to the ground truth Shapley values ($\phi$). As mentioned in the experimental set, the ground truth Shapley values were generated by KernelSHAP after convergence on each example in the test set. In the explainability evaluation, we used the models trained with default hyperparameters in the predictive performance evaluation, which generally showed high similarity to the ground truth, as demonstrated by the cosine similarity measurements. The Friedman test found no significant differences in the cosine similarity between the compared algorithms over the 25 datasets. The detailed results are available in Table 2.

*Table 2.* The cosine similarity of the ground truth Shapley values to the Shapley values obtained from $KAN^{\mathcal{V}ia}$, $KAN_{\varrho}^{\mathcal{V}ia}$, $MLP^{\mathcal{V}ia}$, and $MLP_{\theta}^{\mathcal{V}ia}$. The best-performing model is colored in light green.

| Dataset | $KAN^{\mathcal{V}ia}$ | $KAN_{\varrho}^{\mathcal{V}ia}$ | $MLP^{\mathcal{V}ia}$ | $MLP_{\theta}^{\mathcal{V}ia}$ |
|---|---|---|---|---|
| Abalone | 0.969 ± 0.0166 | 0.966 ± 0.013 | 0.647 ± 0.21 | 0.807 ± 0.214 |
| Ada Prior | 0.935 ± 0.046 | 0.982 ± 0.006 | 0.663 ± 0.142 | 0.908 ± 0.045 |
| Adult | 0.931 ± 0.049 | 0.992 ± 0.011 | 0.574 ± 0.16 | 0.947 ± 0.032 |
| Bank32nh | 0.779 ± 0.163 | 0.713 ± 0.187 | 0.794 ± 0.166 | 0.876 ± 0.084 |
| Electricity | 0.970 ± 0.02 | 0.971 ± 0.017 | 0.912 ± 0.131 | 0.913 ± 0.09 |
| Elevators | 0.966 ± 0.024 | 0.966 ± 0.026 | 0.976 ± 0.025 | 0.976 ± 0.02 |
| Fars | 0.886 ± 0.253 | 0.886 ± 0.28 | 0.95 ± 0.104 | 0.943 ± 0.058 |
| Helena | 0.856 ± 0.092 | 0.715 ± 0.157 | 0.840 ± 0.099 | 0.789 ± 0.104 |
| Heloc | 0.844 ± 0.111 | 0.671 ± 0.182 | 0.759 ± 0.176 | 0.832 ± 0.125 |
| Higgs | 0.917 ± 0.068 | 0.925 ± 0.062 | 0.92 ± 0.093 | 0.912 ± 0.097 |
| LHC Identify Jet | 0.971 ± 0.021 | 0.952 ± 0.065 | 0.97 ± 0.042 | 0.972 ± 0.041 |
| House 16H | 0.919 ± 0.048 | 0.922 ± 0.043 | 0.927 ± 0.06 | 0.944 ± 0.048 |
| Indian Pines | 0.796 ± 0.121 | 0.241 ± 0.07 | 0.304 ± 0.077 | 0.325 ± 0.084 |
| Jannis | 0.852 ± 0.141 | 0.546 ± 0.189 | 0.675 ± 0.13 | 0.439 ± 0.164 |
| JM1 | 0.88 ± 0.044 | 0.667 ± 0.217 | 0.795 ± 0.203 | 0.839 ± 0.159 |
| Magic Telescope | 0.922 ± 0.067 | 0.935 ± 0.058 | 0.973 ± 0.035 | 0.962 ± 0.058 |
| MC1 | 0.466 ± 0.268 | 0.794 ± 0.084 | 0.777 ± 0.127 | 0.887 ± 0.055 |
| Microaggregation2 | 0.938 ± 0.049 | 0.610 ± 0.149 | 0.840 ± 0.099 | 0.81 ± 0.096 |
| Mozilla4 | 0.953 ± 0.023 | 0.948 ± 0.016 | 0.975 ± 0.018 | 0.979 ± 0.022 |
| Satellite | 0.841 ± 0.116 | 0.870 ± 0.077 | 0.766 ± 0.159 | 0.861 ± 0.093 |
| PC2 | 0.534 ± 0.183 | 0.905 ± 0.053 | 0.786 ± 0.137 | 0.827 ± 0.098 |
| Phonemes | 0.811 ± 0.162 | 0.868 ± 0.082 | 0.873 ± 0.126 | 0.916 ± 0.083 |
| Pollen | 0.952 ± 0.059 | 0.945 ± 0.023 | 0.464 ± 0.476 | 0.592 ± 0.439 |
| Telco Customer Churn | 0.81 ± 0.108 | 0.904 ± 0.051 | 0.43 ± 0.189 | 0.592 ± 0.231 |
| 1st order theorem proving | 0.725 ± 0.179 | 0.464 ± 0.517 | 0.387 ± 0.182 | 0.539 ± 0.144 |

We also measured similarity in ranking the important features between the computed Shapley values ($\phi^{\text{Via}}(x;\theta)$) and the ground truth Shapley values ($\phi$) using the Spearman rank correlation coefficient. $KAN^{\mathcal{V}ia}$ is ranked first with respect to the correlation values across the 25 datasets, followed by both $KAN^{\mathcal{V}ia}\varrho$ and $MLP^{\mathcal{V}ia}\theta$ in the second place, and $MLP^{\mathcal{V}ia}$ in the last place. The Spearman rank test revealed that the observed differences are significant. Subsequently, the post-hoc Nemenyi test confirmed that $MLP^{\mathcal{V}ia}$ significantly underperformed the compared algorithms, while the differences between the remaining algorithms are insignificant. Overall, if both the cosine similarity and the Spearman rank are considered, $KAN^{\mathcal{V}ia}$ proved to be a more stable approximator, as detailed in Tables 2 and 3.

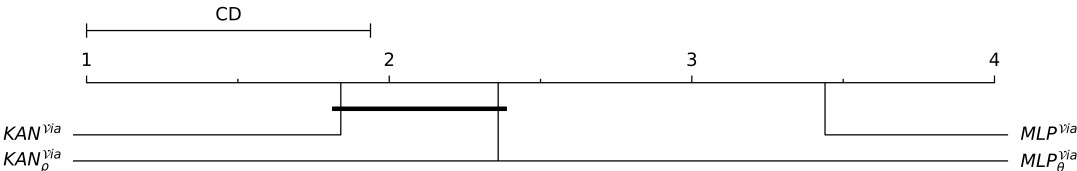

*Figure 6.* **The average rank** of $KAN^{\mathcal{V}ia}$, $KAN_{\varrho}^{\mathcal{V}ia}$, $MLP^{\mathcal{V}ia}$, and $MLP_{\theta}^{\mathcal{V}ia}$ on the 25 datasets with respect to the Spearman correlation between the ground truth Shapley values and the values obtained from the compared models. A lower rank is better and the critical difference (CD) represents the largest difference that is not statistically significant.

*Table 3.* The Spearman rank correlation between the ground truth Shapley values and the Shapley values obtained from $KAN^{\mathcal{V}ia}$, $KAN_{\varrho}^{\mathcal{V}ia}$, and $MLP^{\mathcal{V}ia}$. The best-performing model is colored in light green.

| Dataset | $KAN^{\mathcal{V}ia}$ | $KAN_{\varrho}^{\mathcal{V}ia}$ | $MLP^{\mathcal{V}ia}$ | $MLP_{\theta}^{\mathcal{V}ia}$ |
|---|---|---|---|---|
| Abalone | $0.663 \pm 0.234$ | $0.879 \pm 0.14$ | $0.529 \pm 0.246$ | $0.649 \pm 0.236$ |
| Ada Prior | $0.876 \pm 0.088$ | $0.962 \pm 0.025$ | $0.576 \pm 0.163$ | $0.869 \pm 0.081$ |
| Adult | $0.959 \pm 0.035$ | $0.932 \pm 0.034$ | $0.398 \pm 0.214$ | $0.864 \pm 0.084$ |
| Bank32nh | $0.432 \pm 0.151$ | $0.433 \pm 0.139$ | $0.349 \pm 0.15$ | $0.486 \pm 0.129$ |
| Electricity | $0.798 \pm 0.183$ | $0.838 \pm 0.142$ | $0.751 \pm 0.206$ | $0.848 \pm 0.137$ |
| Elevators | $0.920 \pm 0.064$ | $0.888 \pm 0.072$ | $0.883 \pm 0.07$ | $0.902 \pm 0.06$ |
| Fars | $0.347 \pm 0.328$ | $0.106 \pm 0.133$ | $0.512 \pm 0.164$ | $0.491 \pm 0.115$ |
| Helena | $0.669 \pm 0.152$ | $0.475 \pm 0.188$ | $0.656 \pm 0.159$ | $0.660 \pm 0.168$ |
| Heloc | $0.741 \pm 0.147$ | $0.673 \pm 0.159$ | $0.589 \pm 0.173$ | $0.701 \pm 0.143$ |
| Higgs | $0.674 \pm 0.12$ | $0.718 \pm 0.112$ | $0.535 \pm 0.143$ | $0.568 \pm 0.139$ |
| LHC Identify Jet | $0.857 \pm 0.119$ | $0.726 \pm 0.184$ | $0.737 \pm 0.164$ | $0.724 \pm 0.146$ |
| House 16H | $0.888 \pm 0.092$ | $0.858 \pm 0.102$ | $0.823 \pm 0.112$ | $0.864 \pm 0.095$ |
| Indian Pines | $0.699 \pm 0.116$ | $0.057 \pm 0.054$ | $0.099 \pm 0.07$ | $0.181 \pm 0.056$ |
| Jannis | $0.477 \pm 0.131$ | $0.314 \pm 0.174$ | $0.343 \pm 0.132$ | $0.227 \pm 0.137$ |
| JM1 | $0.756 \pm 0.202$ | $0.682 \pm 0.223$ | $0.59 \pm 0.188$ | $0.715 \pm 0.189$ |
| Magic Telescope | $0.9 \pm 0.098$ | $0.91 \pm 0.087$ | $0.882 \pm 0.098$ | $0.828 \pm 0.141$ |
| MC1 | $0.621 \pm 0.157$ | $0.885 \pm 0.088$ | $0.619 \pm 0.169$ | $0.716 \pm 0.108$ |
| Microaggregation2 | $0.876 \pm 0.096$ | $0.411 \pm 0.183$ | $0.656 \pm 0.159$ | $0.705 \pm 0.2$ |
| Mozilla4 | $0.942 \pm 0.092$ | $0.971 \pm 0.063$ | $0.909 \pm 0.161$ | $0.913 \pm 0.137$ |
| Satellite | $0.746 \pm 0.212$ | $0.786 \pm 0.151$ | $0.677 \pm 0.208$ | $0.8 \pm 0.132$ |
| PC2 | $0.733 \pm 0.161$ | $0.924 \pm 0.09$ | $0.675 \pm 0.154$ | $0.737 \pm 0.135$ |
| Phonemes | $0.941 \pm 0.103$ | $0.954 \pm 0.083$ | $0.807 \pm 0.213$ | $0.862 \pm 0.159$ |
| Pollen | $0.285 \pm 0.442$ | $0.171 \pm 0.484$ | $0.297 \pm 0.498$ | $0.407 \pm 0.545$ |
| Telco Customer Churn | $0.848 \pm 0.098$ | $0.938 \pm 0.043$ | $0.262 \pm 0.297$ | $0.471 \pm 0.211$ |
| 1st order theorem proving | $0.623 \pm 0.188$ | $0.082 \pm 0.145$ | $0.183 \pm 0.146$ | $0.367 \pm 0.14$ |

# F. Image Experiments

We implemented ViaSHAP for image classification using three architectures: ResNet50 (He et al., 2016) ($ResNet50^{\mathcal{V}ia}$), ResNet18 ($ResNet18^{\mathcal{V}ia}$), and U-Net (Ronneberger et al., 2015) ($U\text{-}Net^{\mathcal{V}ia}$). The predictive performance of these models was evaluated using Top-1 Accuracy, with the results summarized in Table 4. All models were trained on the CIFAR-10 (Krizhevsky et al., 2014) dataset without transfer learning or pre-trained weights (i.e., trained from scratch) using four masks (samples) per data instance. The training incorporated early stopping, terminating after ten epochs without improvement on a validation split (10% of the training data). The results of evaluating the performance of the trained models on the test set demonstrate that ViaSHAP can achieve high predictive performance on standard image classification tasks.

Table 4. A comparison of the predictive performance of $ResNet50^{\mathcal{V}ia}$, $ResNet18^{\mathcal{V}ia}$, and $U\text{-}Net^{\mathcal{V}ia}$ measured in AUC.

|  | AUC | 0.95 Confidence Interval |
|---|---|---|
| $U\text{-}Net^{\mathcal{V}ia}$ | **0.983** | **(0.981, 0.986)** |
| $ResNet18^{\mathcal{V}ia}$ | 0.968 | (0.964, 0.971) |
| $ResNet50^{\mathcal{V}ia}$ | 0.96 | (0.956, 0.964) |

In order to assess the accuracy of the Shapley values computed by ViaSHAP implementations, we followed a methodology similar to (Jethani et al., 2022). Specifically, we selected the top 50% most important features identified by the explainer and evaluated the predictive performance of the explained model under two conditions: using only the selected top features (Inclusion Accuracy) and excluding the top features (Exclusion Accuracy).

We compared the accuracy of Shapley value approximations of the three models ($ResNet50^{\mathcal{V}ia}$, $ResNet18^{\mathcal{V}ia}$, and $U\text{-}Net^{\mathcal{V}ia}$). We also evaluated the accuracy of FastSHAP's approximations where the three ViaSHAP implementations for image classification are provided as black boxes to FastSHAP. The results indicate that the ViaSHAP implementations consistently provide more accurate Shapley value approximations than those generated by FastSHAP, as shown in Table 5. Figure 7 presents two examples illustrating the explanations generated by ViaSHAP models and FastSHAP, where the latter treats the ViaSHAP models as black-box predictors. We also show the effects of using different percentages of the top features considered for inclusion and exclusion on the top-1 accuracy in Figure 8.

Table 5. The accuracy of the Shapley values is evaluated using the top 50% of the most important features (according to their Shapley values). The Inclusion AUC (higher values are better) and the Exclusion AUC (lower values are better) are computed using the top 1 accuracy.

| Dataset | Exclusion AUC | 0.95 Confidence Interval | Inclusion AUC | 0.95 Confidence Interval |
|---|---|---|---|---|
| $U\text{-}Net^{\mathcal{V}ia}$ | 0.773 | (0.747, 0.799) | 0.988 | (0.981, 0.995) |
| FastSHAP($U\text{-}Net^{\mathcal{V}ia}$) | 0.864 | (0.843, 0.885) | 0.978 | (0.969, 0.987) |
| $ResNet18^{\mathcal{V}ia}$ | 0.611 | (0.581, 0.642) | 0.99 | (0.983, 0.996) |
| FastSHAP($ResNet18^{\mathcal{V}ia}$) | 0.755 | (0.728, 0.782) | 0.954 | (0.941, 0.967) |
| $ResNet50^{\mathcal{V}ia}$ | **0.554** | **(0.523, 0.585)** | **0.997** | **(0.994, 1.0)** |
| FastSHAP($ResNet50^{\mathcal{V}ia}$) | 0.778 | (0.753, 0.804) | 0.978 | (0.969, 0.987) |

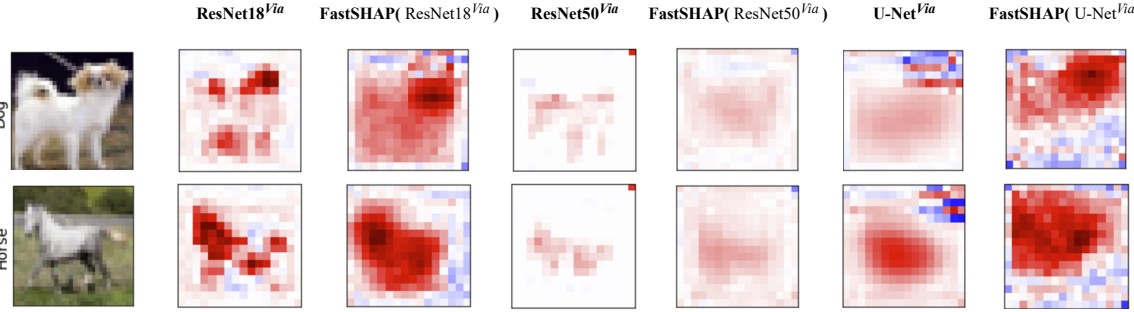

Figure 7. The explanations generated by ViaSHAP models and FastSHAP using two randomly selected images from the CIFAR-10 dataset.

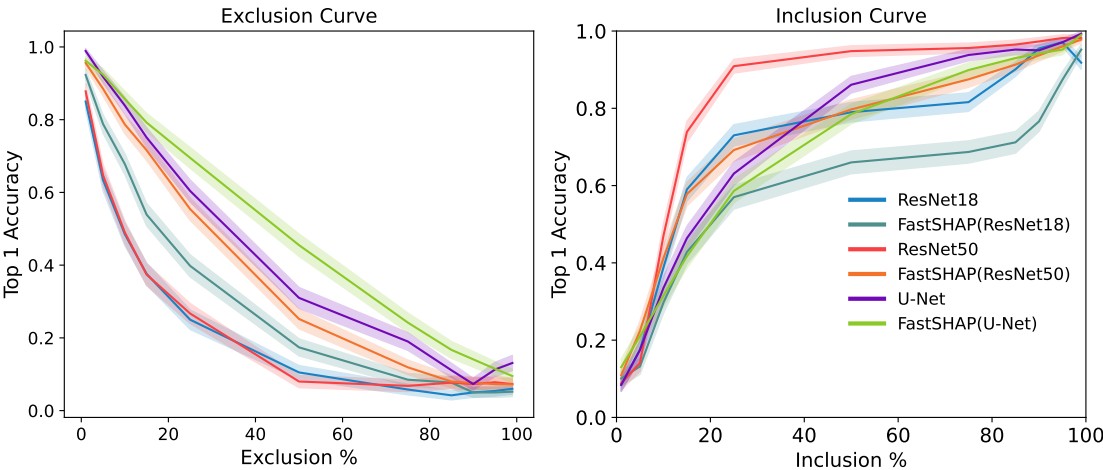

*Figure 8.* The inclusion and exclusion curves of ViaSHAP implementations as well as their FastSHAP explainers. We show how the top-1 accuracy of the predictive model changes as we exclude or include an increasing share of the important features, where the important features are determined by each explainer in the comparison.

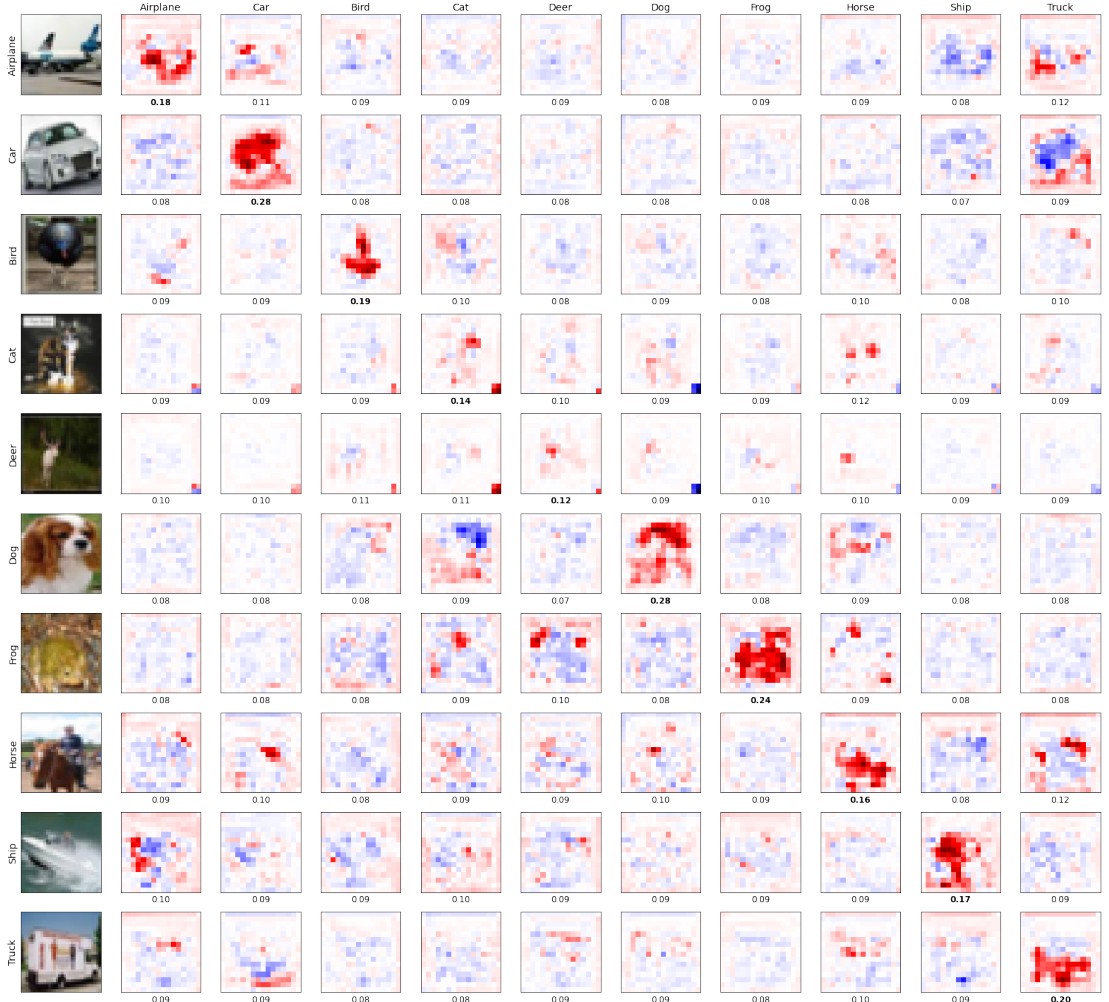

*Figure 9.* The explanations of *ResNet18*$^{\mathcal{V}ia}$ for 10 randomly selected predictions on the CIFAR-10 dataset. Each column corresponds to a CIFAR-10 class, and the predicted probability by *ResNet18*$^{\mathcal{V}ia}$ displayed beneath each image.

# G. Relaxed Expected Prediction

ViaSHAP is optimized to minimize the loss function (7) and predict 0 if all features are marginalized out ($Via^{SHAP}(\mathbf{0}) = 0$), which can be a limitation if the baseline removal approach (Sundararajan & Najmi, 2020), for instance, is applied to mask features using their mean values, i.e., $Via^{SHAP}(\mathbb{E}(x)) = 0$. In some cases, particularly with heavily imbalanced data, the "average values" $\mathbb{E}(x)$ might strongly represent one class over others. Moreover, in regression problems, the prediction of an accurate estimator using the expected values is unlikely to be 0. Therefore, we propose a relaxed variant of the optimization problem, where $Via^{SHAP}(\mathbf{0})$ is not obliged to predict 0 in order to minimize the Shapley loss. We introduce a bias term $\delta$ to ViaSHAP, modifying the predictions such that $y = \mathbf{1}^\top \phi^{Via}(x; \theta) + \delta$. Accordingly, the relaxed loss function is formulated as follows:

$$
\mathcal{L}_\phi(\theta) = \sum_{x \in X} \sum_{j \in M} \left( \beta \cdot \left( \underset{p(S)}{\mathbb{E}} \left[ \left( (\mathbf{1}^\top \phi^{Via}(x^S; \theta) + \delta) - (\mathbf{1}^\top \phi^{Via}(\mathbf{0}; \theta) + \delta) - \mathbf{1}_S^\top \phi_j^{Via}(x; \theta) \right)^2 \right] \right) \right.
$$
$$
\left. - \left( y_j \log(\mathbf{1}^\top \phi_j^{Via}(x; \theta) + \delta) \right) \right), \quad (10)
$$

Consequently, ViaSHAP explains $Via^{SHAP}(x) - \delta$, where $\delta$ is optimized through the prediction loss function to minimize the overall prediction error. In the case of baseline removal using the expected values, $Via^{SHAP}(\mathbb{E}(x)) \approx \delta$.

To measure the effect of the relaxed loss function on the predictive performance as well as the accuracy of the Shapley value approximations, we conduct an experiment where we compare the performance of two identical KAN implementations: one trained with the default loss function (7) and the other trained with the relaxed variant of the loss function (10). The two implementations are trained using the default setting without a link function applied to the predicted outcome. The detailed results of predictive performance are presented in Table 6, while the accuracy of the Shapley value approximations is provided in Table 7.

We test the null hypothesis that no significant difference exists in predictive performance, as measured by AUC, between ViaSHAP trained using the default architecture and its relaxed variant. Given that only two approaches are compared, the Wilcoxon signed-rank test (Wilcoxon, 1945) is employed. The results indicate that the null hypothesis cannot be rejected at the 0.05 significance level, i.e., there is no significant difference in the predictive performance of the compared approaches. Although the datasets used in the experiments include 19 imbalanced datasets, the results of both variants of ViaSHAP are performing remarkably well.

The results regarding the similarity of the approximated Shapley values to the ground truth show no significant difference between the compared approaches, as measured by the three similarity metrics: cosine similarity, Spearman's rank, and $R^2$.

*Table 6.* The predictive performance of $KAN^{\mathcal{V}ia}$ optimized for $\mathcal{V}ia^{SHAP}(\mathbf{0}) = 0$ vs. the relaxed version of the optimization function ($\mathcal{V}ia^{SHAP}(\mathbf{0}) = \delta$). The predictive performance is measured by AUC. The best-performing model is colored in light green .

| Dataset | $KAN^{\mathcal{V}ia}$ ($\mathcal{V}ia^{SHAP}(\mathbf{0}) = 0$) | $KAN^{\mathcal{V}ia}$ ($\mathcal{V}ia^{SHAP}(\mathbf{0}) = \delta$) |
|---|---|---|
| Abalone | $0.883 \pm 0.0002$ | $0.883 \pm 0.0002$ |
| Ada Prior | $0.898 \pm 0.003$ | $0.900 \pm 0.003$ |
| Adult | $0.919 \pm 0.0005$ | $0.919 \pm 0.003$ |
| Bank32nh | $0.883 \pm 0.003$ | $0.887 \pm 0.002$ |
| Electricity | $0.934 \pm 0.004$ | $0.936 \pm 0.006$ |
| Elevators | $0.936 \pm 0.003$ | $0.937 \pm 0.001$ |
| Fars | $0.958 \pm 0.002$ | $0.958 \pm 0.001$ |
| Helena | $0.868 \pm 0.006$ | $0.873 \pm 0.003$ |
| Heloc | $0.792 \pm 0.001$ | $0.793 \pm 0.001$ |
| Higgs | $0.801 \pm 0.001$ | $0.801 \pm 0.001$ |
| LHC Identify Jet | $0.939 \pm 0.0005$ | $0.938 \pm 0.003$ |
| House 16H | $0.949 \pm 0.001$ | $0.951 \pm 0.001$ |
| Indian Pines | $0.982 \pm 0.001$ | $0.981 \pm 0.002$ |
| Jannis | $0.861 \pm 0.001$ | $0.858 \pm 0.003$ |
| JM1 | $0.686 \pm 0.025$ | $0.703 \pm 0.025$ |
| Magic Telescope | $0.921 \pm 0.002$ | $0.925 \pm 0.002$ |
| MC1 | $0.952 \pm 0.011$ | $0.942 \pm 0.013$ |
| Microaggregation2 | $0.764 \pm 0.008$ | $0.766 \pm 0.011$ |
| Mozilla4 | $0.965 \pm 0.001$ | $0.965 \pm 0.001$ |
| Satellite | $0.944 \pm 0.010$ | $0.964 \pm 0.022$ |
| PC2 | $0.659 \pm 0.060$ | $0.689 \pm 0.031$ |
| Phonemes | $0.923 \pm 0.003$ | $0.922 \pm 0.002$ |
| Pollen | $0.501 \pm 0.002$ | $0.502 \pm 0.004$ |
| Telco Customer Churn | $0.857 \pm 0.003$ | $0.852 \pm 0.006$ |
| 1st order theorem proving | $0.810 \pm 0.006$ | $0.761 \pm 0.003$ |

Table 7. A comparison to evaluate the similarity to the ground truth explanations between $KAN^{Via}$ optimized for $\mathcal{V}ia^{SHAP}(\mathbf{0}) = 0$ and the relaxed version of the optimization function ($\mathcal{V}ia^{SHAP}(\mathbf{0}) = \delta$). The similarity is measured using cosine similarity, Spearman's rank, and $R^2$. The best-performing model is colored in light green.

| Dataset | Cosine Similarity | | Spearman's Rank | | $R^2$ | |
|---|---|---|---|---|---|---|
| | $\mathcal{V}ia^{SHAP}(\mathbf{0}) = 0$ | $\mathcal{V}ia^{SHAP}(\mathbf{0}) = \delta$ | $\mathcal{V}ia^{SHAP}(\mathbf{0}) = 0$ | $\mathcal{V}ia^{SHAP}(\mathbf{0}) = \delta$ | $\mathcal{V}ia^{SHAP}(\mathbf{0}) = 0$ | $\mathcal{V}ia^{SHAP}(\mathbf{0}) = \delta$ |
| Abalone | 0.999 ± 0.0008 | 0.999 ± 0.001 | 0.971 ± 0.052 | 0.972 ± 0.054 | 0.999 ± 0.002 | 0.998 ± 0.003 |
| Ada Prior | 0.963 ± 0.037 | 0.945 ± 0.053 | 0.909 ± 0.068 | 0.886 ± 0.088 | 0.9 ± 0.095 | 0.858 ± 0.132 |
| Adult | 0.981 ± 0.03 | 0.984 ± 0.023 | 0.931 ± 0.074 | 0.941 ± 0.056 | 0.948 ± 0.079 | 0.956 ± 0.066 |
| Bank32nh | 0.948 ± 0.045 | 0.948 ± 0.043 | 0.648 ± 0.114 | 0.646 ± 0.113 | 0.87 ± 0.142 | 0.874 ± 0.126 |
| Electricity | 0.998 ± 0.004 | 0.997 ± 0.005 | 0.967 ± 0.043 | 0.965 ± 0.043 | 0.992 ± 0.012 | 0.992 ± 0.016 |
| Elevators | 0.997 ± 0.004 | 0.995 ± 0.006 | 0.969 ± 0.026 | 0.963 ± 0.031 | 0.993 ± 0.009 | 0.988 ± 0.016 |
| Fars | 0.962 ± 0.036 | 0.995 ± 0.012 | 0.882 ± 0.073 | 0.724 ± 0.119 | 0.895 ± 0.073 | 0.988 ± 0.029 |
| Helena | 0.874 ± 0.095 | 0.873 ± 0.101 | 0.702 ± 0.148 | 0.701 ± 0.143 | 0.677 ± 0.204 | 0.689 ± 0.197 |
| Heloc | 0.962 ± 0.036 | 0.973 ± 0.029 | 0.882 ± 0.073 | 0.885 ± 0.078 | 0.895 ± 0.105 | 0.934 ± 0.065 |
| Higgs | 0.991 ± 0.006 | 0.989 ± 0.01 | 0.87 ± 0.057 | 0.833 ± 0.076 | 0.977 ± 0.014 | 0.974 ± 0.023 |
| LHC Identify Jet | 0.999 ± 0.002 | 0.999 ± 0.005 | 0.974 ± 0.032 | 0.961 ± 0.047 | 0.998 ± 0.005 | 0.997 ± 0.012 |
| House 16H | 0.988 ± 0.015 | 0.987 ± 0.018 | 0.952 ± 0.044 | 0.948 ± 0.051 | 0.961 ± 0.057 | 0.965 ± 0.04 |
| Indian Pines | 0.683 ± 0.171 | 0.648 ± 0.147 | 0.553 ± 0.18 | 0.511 ± 0.163 | 0.333 ± 0.192 | 0.266 ± 0.205 |
| Jannis | 0.898 ± 0.072 | 0.884 ± 0.086 | 0.624 ± 0.113 | 0.607 ± 0.125 | 0.722 ± 0.183 | 0.717 ± 0.186 |
| JM1 | 0.965 ± 0.042 | 0.965 ± 0.029 | 0.916 ± 0.085 | 0.917 ± 0.077 | 0.901 ± 0.094 | 0.907 ± 0.09 |
| Magic Telescope | 0.994 ± 0.006 | 0.997 ± 0.004 | 0.959 ± 0.042 | 0.968 ± 0.039 | 0.98 ± 0.02 | 0.991 ± 0.012 |
| MC1 | 0.951 ± 0.093 | 0.957 ± 0.089 | 0.881 ± 0.139 | 0.891 ± 0.121 | 0.873 ± 0.332 | 0.894 ± 0.236 |
| Microaggregation2 | 0.982 ± 0.021 | 0.985 ± 0.021 | 0.957 ± 0.049 | 0.959 ± 0.056 | 0.929 ± 0.114 | 0.956 ± 0.057 |
| Mozilla4 | 0.9998 ± 0.0003 | 0.9998 ± 0.0005 | 0.967 ± 0.074 | 0.968 ± 0.071 | 0.9996 ± 0.0007 | 0.9995 ± 0.001 |
| Satellite | 0.976 ± 0.033 | 0.961 ± 0.052 | 0.894 ± 0.102 | 0.856 ± 0.131 | 0.814 ± 0.296 | 0.83 ± 0.203 |
| PC2 | 0.956 ± 0.087 | 0.945 ± 0.096 | 0.875 ± 0.127 | 0.857 ± 0.133 | 0.895 ± 0.223 | 0.853 ± 0.606 |
| Phonemes | 0.993 ± 0.013 | 0.996 ± 0.005 | 0.951 ± 0.094 | 0.972 ± 0.058 | 0.975 ± 0.076 | 0.987 ± 0.022 |
| Pollen | 0.994 ± 0.013 | 0.988 ± 0.02 | 0.959 ± 0.076 | 0.933 ± 0.11 | 0.929 ± 0.212 | 0.86 ± 0.913 |
| Telco Customer Churn | 0.978 ± 0.025 | 0.973 ± 0.027 | 0.934 ± 0.052 | 0.913 ± 0.068 | 0.939 ± 0.054 | 0.93 ± 0.06 |
| 1st order theorem proving | 0.778 ± 0.123 | 0.814 ± 0.131 | 0.66 ± 0.146 | 0.771 ± 0.142 | 0.429 ± 0.479 | 0.612 ± 0.239 |

# H. ViaSHAP with Marginal Expectations

We evaluate the predictive performance and the similarity of the approximate Shapley values to the ground truth values when ViaSHAP is trained using marginal expectations (Chen et al., 2020) as the strategy for marginalizing out features $\left(\mathcal{V}ia^{SHAP}(x^S) = \mathbb{E}_{x^S}\left[\mathbf{1}^\top \phi^{\mathcal{V}ia}(x^S, X^{N\setminus S}; \theta)\right]\right)$.

In the experiment, we compare ViaSHAP trained using baseline removal vs. marginal expectations. The compared models are trained using the default settings of ViaSHAP without link functions. The marginal expectations models employ 128 data instances as background data selected from the validation set. The results of the experiment, presented in Tables 8 and 9, indicate that ViaSHAP trained using the marginal expectations value function significantly underperforms the baseline removal approach. Furthermore, ViaSHAP trained with marginal expectations fails to accurately approximate the solutions provided by the unbiased KernelSHAP that employs marginal expectations. The results indicate that employing the marginal expectations value function can hinder the learning of an accurate predictor.

*Table 8.* The predictive performance of $KAN^{\mathcal{V}ia}$ optimized using marginal expectations for masking features vs. using baseline removal. The predictive performance is measured by AUC. The best-performing model is colored in light green .

| Dataset | Marginal Expectations | Baseline Removal |
|---------|----------------------|------------------|
| Abalone | $0.860 \pm 0.005$ | $0.883 \pm 0.0002$ |
| Ada Prior | $0.833 \pm 0.010$ | $0.898 \pm 0.003$ |
| Adult | $0.888 \pm 0.004$ | $0.919 \pm 0.000$ |
| Bank32nh | $0.833 \pm 0.007$ | $0.883 \pm 0.003$ |
| Elevators | $0.875 \pm 0.006$ | $0.936 \pm 0.003$ |
| Helena | $0.850 \pm 0.003$ | $0.868 \pm 0.006$ |
| House 16H | $0.929 \pm 0.003$ | $0.949 \pm 0.001$ |
| Indian Pines | $0.939 \pm 0.039$ | $0.982 \pm 0.001$ |
| JM1 | $0.711 \pm 0.009$ | $0.686 \pm 0.025$ |
| MC1 | $0.936 \pm 0.019$ | $0.952 \pm 0.011$ |
| Microaggregation2 | $0.758 \pm 0.017$ | $0.764 \pm 0.008$ |
| Mozilla4 | $0.924 \pm 0.006$ | $0.965 \pm 0.001$ |
| Phonemes | $0.907 \pm 0.006$ | $0.923 \pm 0.003$ |
| Telco Customer Churn | $0.824 \pm 0.010$ | $0.857 \pm 0.003$ |

*Table 9.* The similarity to the ground truth explanations when $KAN^{\mathcal{V}ia}$ optimized using marginal expectations for masking features vs. using baseline removal. The similarity is measured using cosine similarity, Spearman's rank, and $R^2$. The best-performing model is colored in light green.

| Dataset | Cosine Similarity | | Spearman's Rank | | $R^2$ | |
|---|---|---|---|---|---|---|
| | Marginal | Baseline | Marginal | Baseline | Marginal | Baseline |
| Abalone | $0.104 \pm 0.629$ | $0.999 \pm 0.001$ | $0.697 \pm 0.253$ | $0.971 \pm 0.052$ | $-14.28 \pm 13.19$ | $0.999 \pm 0.002$ |
| Ada Prior | $0.331 \pm 0.286$ | $0.963 \pm 0.037$ | $0.526 \pm 0.223$ | $0.909 \pm 0.068$ | $-0.87 \pm 1.071$ | $0.9 \pm 0.095$ |
| Adult | $0.448 \pm 0.375$ | $0.981 \pm 0.03$ | $0.665 \pm 0.176$ | $0.931 \pm 0.074$ | $-0.778 \pm 1.261$ | $0.948 \pm 0.079$ |
| Bank32nh | $0.811 \pm 0.116$ | $0.948 \pm 0.045$ | $0.426 \pm 0.143$ | $0.648 \pm 0.114$ | $0.544 \pm 0.301$ | $0.87 \pm 0.142$ |
| Elevators | $0.497 \pm 0.362$ | $0.997 \pm 0.004$ | $0.559 \pm 0.224$ | $0.969 \pm 0.026$ | $-1.498 \pm 1.427$ | $0.993 \pm 0.009$ |
| Helena | $0.286 \pm 0.149$ | $0.874 \pm 0.095$ | $-0.013 \pm 0.187$ | $0.702 \pm 0.148$ | $-0.221 \pm 0.533$ | $0.677 \pm 0.204$ |
| House 16H | $0.926 \pm 0.065$ | $0.988 \pm 0.015$ | $0.809 \pm 0.125$ | $0.952 \pm 0.044$ | $0.752 \pm 0.188$ | $0.961 \pm 0.057$ |
| Indian Pines | $0.021 \pm 0.067$ | $0.683 \pm 0.171$ | $-0.0004 \pm 0.066$ | $0.553 \pm 0.18$ | $-0.26 \pm 0.328$ | $0.333 \pm 0.192$ |
| JM1 | $0.691 \pm 0.229$ | $0.965 \pm 0.042$ | $0.596 \pm 0.198$ | $0.916 \pm 0.085$ | $-0.809 \pm 1.686$ | $0.901 \pm 0.094$ |
| MC1 | $0.568 \pm 0.186$ | $0.951 \pm 0.093$ | $0.449 \pm 0.177$ | $0.881 \pm 0.139$ | $-0.839 \pm 0.177$ | $0.873 \pm 0.332$ |
| Microaggregation2 | $0.366 \pm 0.236$ | $0.982 \pm 0.021$ | $0.177 \pm 0.231$ | $0.957 \pm 0.049$ | $-0.242 \pm 0.48$ | $0.929 \pm 0.114$ |
| Mozilla4 | $0.884 \pm 0.147$ | $0.9998 \pm 0.0003$ | $0.909 \pm 0.141$ | $0.967 \pm 0.074$ | $0.124 \pm 1.551$ | $0.9996 \pm 0.001$ |
| Phonemes | $0.347 \pm 0.578$ | $0.993 \pm 0.013$ | $0.845 \pm 0.188$ | $0.951 \pm 0.094$ | $-8.204 \pm 14.423$ | $0.975 \pm 0.076$ |
| Telco Cust. Churn | $0.646 \pm 0.186$ | $0.978 \pm 0.025$ | $0.607 \pm 0.154$ | $0.934 \pm 0.052$ | $-0.156 \pm 0.486$ | $0.939 \pm 0.054$ |

# I. A Comparison Between ViaSHAP and a KAN Model with the Same Architecture

We conducted an experiment to assess the impact of incorporating Shapley loss in the optimization process on predictive performance of a KAN model. Consequently, we compared $KAN^{Via}$ to a KAN model with an identical architecture that does not compute Shapley values. As summarized in Table 10, the results indicate that $KAN^{Via}$ generally outperforms the KAN model with the same architecture. In order to determine the statistical significance of these results, the Wilcoxon signed-rank test (Wilcoxon, 1945) was employed to test the null hypothesis that no difference exists in predictive performance, as measured by AUC, between $KAN^{Via}$ and the identical KAN model without Shapley values. The test results allowed for the rejection of the null hypothesis, indicating that $KAN^{Via}$ significantly outperforms the KAN architecture that is not optimized to compute Shapley values with respect to the predictive performance as measured by the AUC.

*Table 10.* A comparison between the predictive performance of $KAN^{Via}$ and a KAN model with an identical architecture to $KAN^{Via}$ but does not compute the Shapley values. The results are reported in AUC.

| Dataset | $KAN$ | $KAN^{Via}$ |
|---|---|---|
| Abalone | $0.882 \pm 0.001$ | $0.87 \pm 0.003$ |
| Ada Prior | $0.895 \pm 0.005$ | $0.89 \pm 0.005$ |
| Adult | $0.917 \pm 0.001$ | $0.914 \pm 0.003$ |
| Bank32nh | $0.886 \pm 0.001$ | $0.878 \pm 0.001$ |
| Electricity | $0.924 \pm 0.005$ | $0.93 \pm 0.004$ |
| Elevators | $0.935 \pm 0.003$ | $0.935 \pm 0.002$ |
| Fars | $0.957 \pm 0.001$ | $0.96 \pm 0.0003$ |
| Helena | $0.883 \pm 0.001$ | $0.884 \pm 0.0001$ |
| Heloc | $0.793 \pm 0.002$ | $0.788 \pm 0.002$ |
| Higgs | $0.801 \pm 0.002$ | $0.801 \pm 0.001$ |
| LHC Identify Jet | $0.944 \pm 0.0003$ | $0.944 \pm 0.0001$ |
| House 16H | $0.948 \pm 0.001$ | $0.949 \pm 0.0007$ |
| Indian Pines | $0.935 \pm 0.001$ | $0.985 \pm 0.0004$ |
| Jannis | $0.860 \pm 0.002$ | $0.864 \pm 0.001$ |
| JM1 | $0.725 \pm 0.008$ | $0.732 \pm 0.003$ |
| Magic Telescope | $0.931 \pm 0.001$ | $0.929 \pm 0.001$ |
| MC1 | $0.933 \pm 0.019$ | $0.94 \pm 0.003$ |
| Microaggregation2 | $0.783 \pm 0.002$ | $0.783 \pm 0.002$ |
| Mozilla4 | $0.967 \pm 0.001$ | $0.968 \pm 0.0008$ |
| Satellite | $0.987 \pm 0.003$ | $0.996 \pm 0.001$ |
| PC2 | $0.458 \pm 0.049$ | $0.827 \pm 0.009$ |
| Phonemes | $0.945 \pm 0.002$ | $0.946 \pm 0.003$ |
| Pollen | $0.491 \pm 0.005$ | $0.515 \pm 0.006$ |
| Telco Customer Churn | $0.848 \pm 0.005$ | $0.854 \pm 0.003$ |
| 1st order theorem proving | $0.805 \pm 0.005$ | $0.822 \pm 0.002$ |

# J. Ablation Study

In this section, we explore the influence of key hyperparameters on the performance and behavior of ViaSHAP. Specifically, we investigate the effects of the scaling hyperparameter $\beta$ and the number of sampled coalitions per data instance. We begin by analyzing how variations in $\beta$ impact both predictive performance and the accuracy of the Shapley values generated by ViaSHAP. We then examine the role of the number of sampled coalitions in model performance, followed by an evaluation of how changes in $\beta$ affect the progress of the computed loss values during training. The findings provide valuable insights into the robustness and efficiency of ViaSHAP under different hyperparameter settings.

### J.1. The Impact of Scaling Hyperparameter $\beta$ on the Performance of ViaSHAP

We evaluated the performance of the models trained with different $\beta$ values (in equation 7), where exponentially increasing values are tested. The models were trained using the default hyperparameter settings described in the experimental setup, except for the values of $\beta$. The AUC of the trained models is measured on the test set, as well as the similarity of the predicted Shapley values to the ground truth. The results indicate that the predictive performance of ViaSHAP, as measured by the area under the ROC curve, remains largely unaffected by the value of $\beta$, even when $\beta$ is increased exponentially. On the other hand, the similarity between the computed Shapley values and the ground truth improves as $\beta$ increases. However, the model struggles to learn effectively after $\beta$ exceeds 200, as shown in Figures 10 and 11.

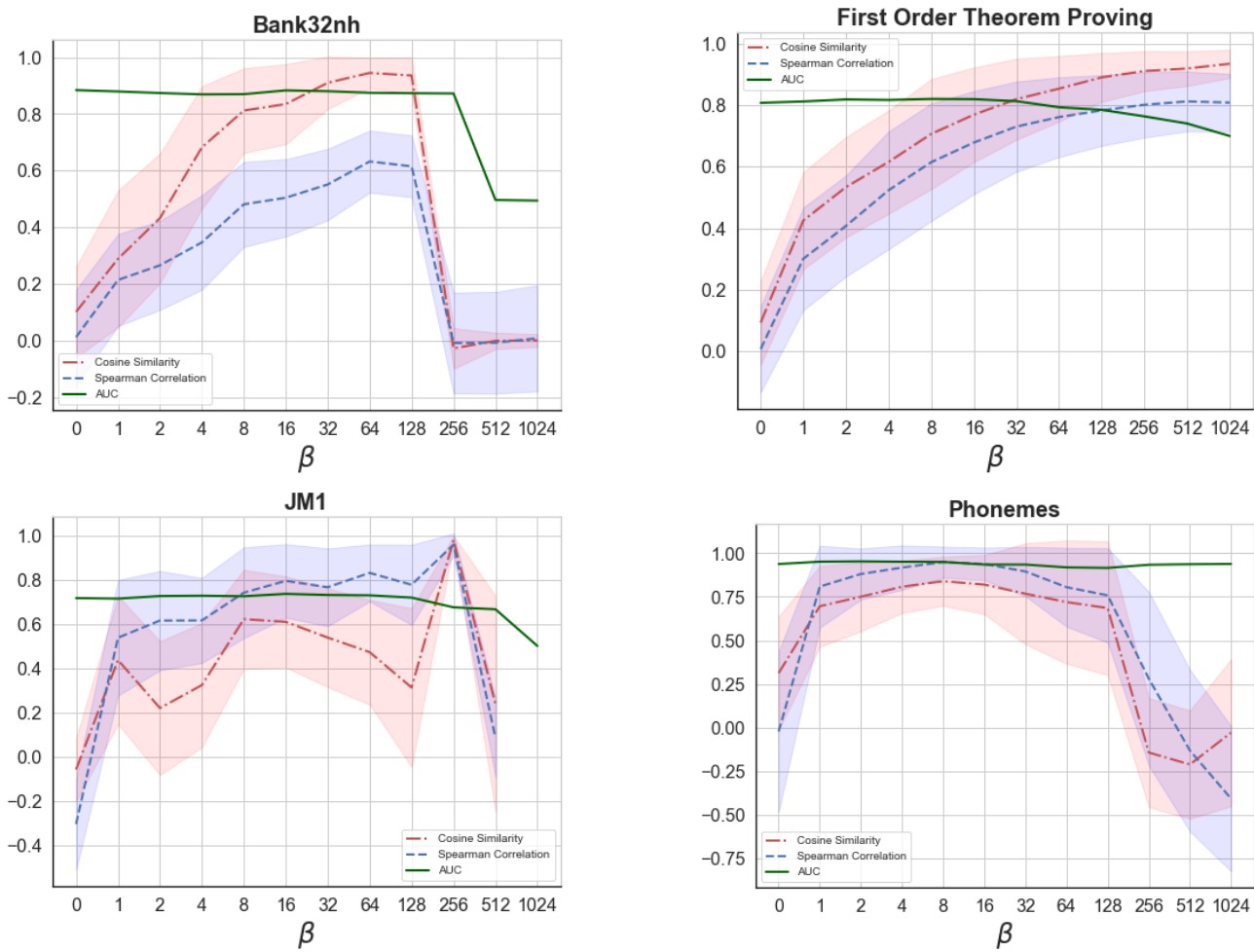

Figure 10. The effect of different values of $\beta$ on the predictive performance (AUC), alignment with the true Shapley values (cosine similarity), and the similarity in the order of features to the ground truth (Spearman rank).

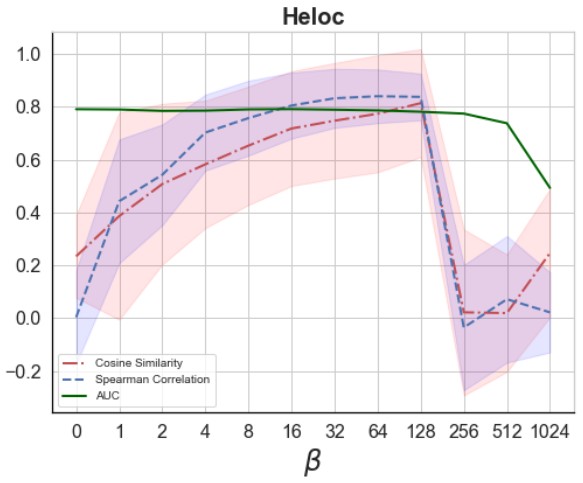 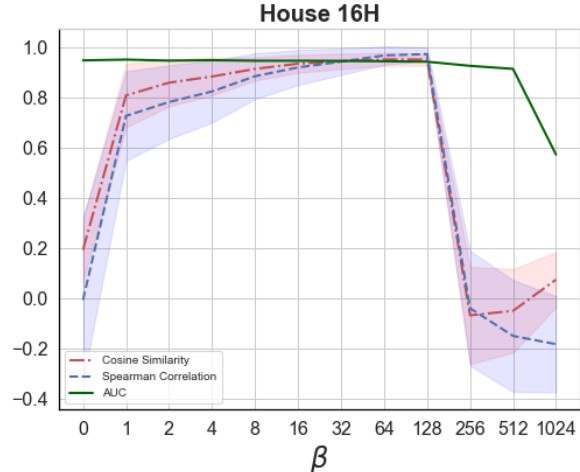

*Figure 11.* The effect of different values of $\beta$ on the predictive performance (AUC), alignment with the true Shapley values (cosine similarity), and the similarity in the order of features to the ground truth (Spearman rank).

## J.2. The Number of Samples

We assessed the impact of the number of sampled coalitions per data example on the performance of ViaSHAP, retraining the model using the default hyperparameters with the exception of the sample size. We investigated an exponentially increasing range of sample sizes ($2^s$), from 1 to 128. The findings suggest that the number of samples has a smaller effect on the performance of the trained models compared to $\beta$, which allows for effective training of ViaSHAP models with as few as one sample per data instance. The results are illustrated in Figures 12 and 13.

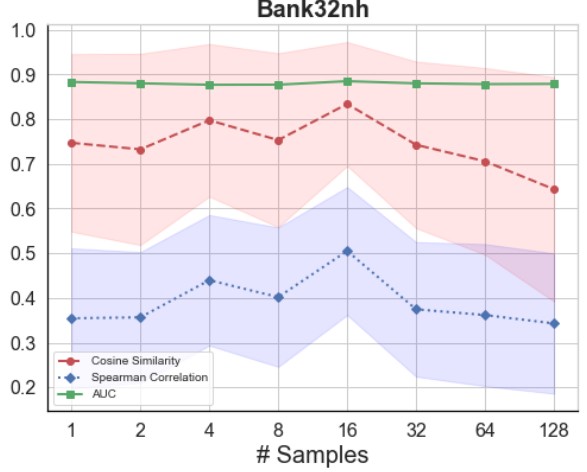 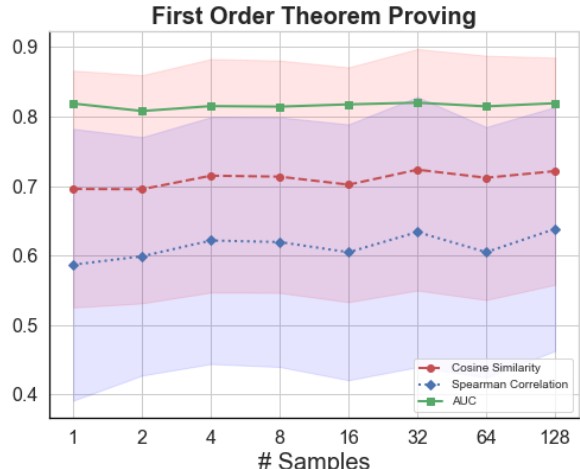

*Figure 12.* The effect of different number of samples on the predictive performance (AUC), alignment with the true Shapley values (cosine similarity), and the similarity in the order of features to the ground truth (Spearman rank).

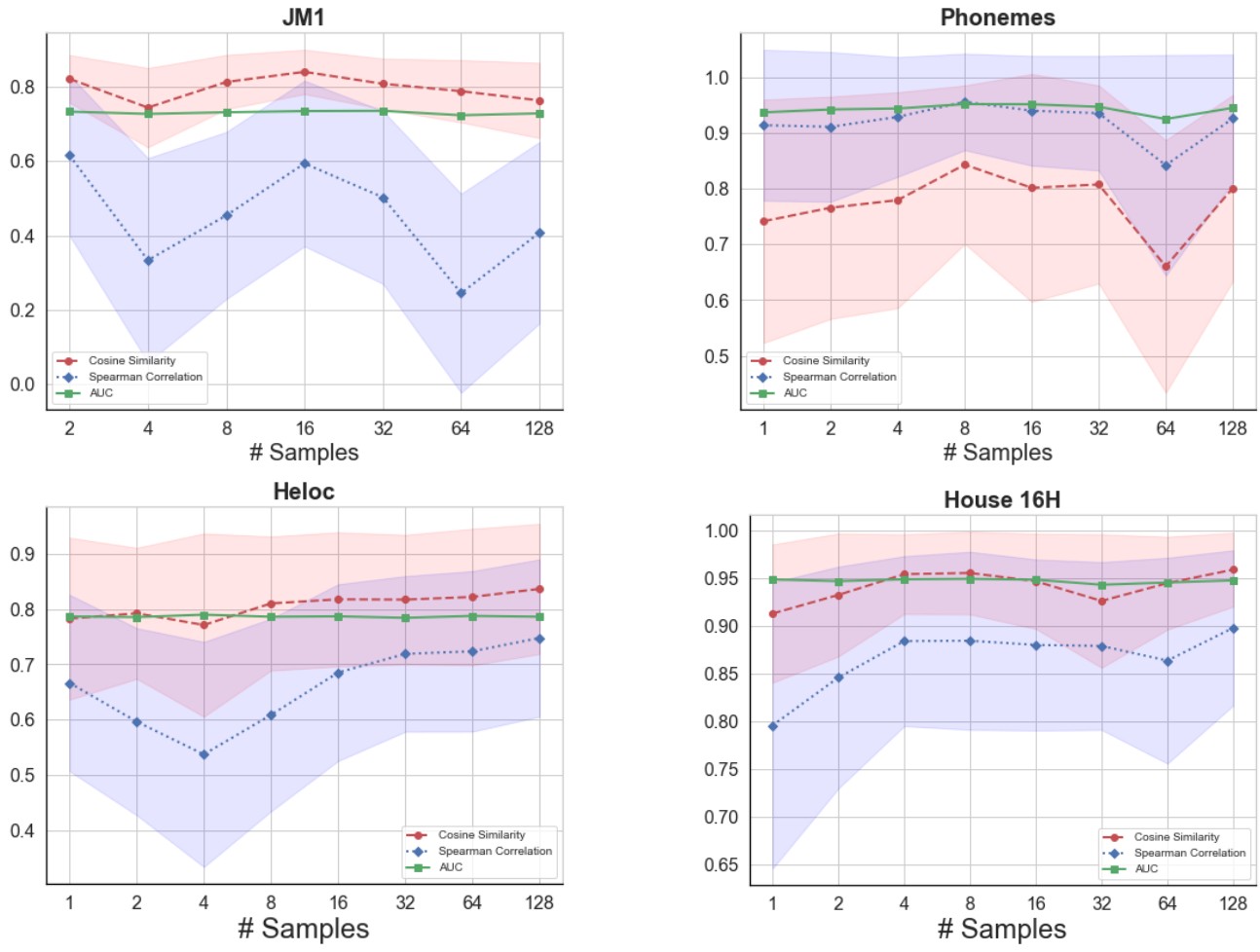

*Figure 13.* The effect of different number of samples on the predictive performance (AUC), alignment with the true Shapley values (cosine similarity), and the similarity in the order of features to the ground truth (Spearman rank).

### J.3. The Effect of Applying a Link Function to the Predicted Outcome

To examine the impact of employing a link function on the predictive performance of ViaSHAP and the accuracy of its Shapley value approximations, we trained $KAN^{\mathcal{V}ia}$ without applying a link function at the output layer and compared the predictive performance to that of $KAN^{\mathcal{V}ia}$ with the default settings mentioned in the experimental setup. The results of the predictive comparison are summarized in Table 11. To evaluate the null hypothesis that there is no difference in predictive performance, measured by the AUC, between $KAN^{\mathcal{V}ia}$ with and without a link function, the Wilcoxon signed-rank test was employed, given that only two methods were compared. The results indicate that the null hypothesis can be rejected at the 0.05 significance level. Therefore, the results indicate that the presence of a link function does not significantly influence predictive performance in general.

The similarity between the ground truth and the approximated Shapley values by $KAN^{\mathcal{V}ia}$, both with and without link functions, are reported in Table 12. The similarity of $KAN^{\mathcal{V}ia}$'s approximations to the ground truth is measured using the cosine similarity and the Spearman's Rank as described in the experimental setup, which allow for measuring the similarity even if two explanations are not on the same scale, since ViaSHAP allows for applying a link function to accommodate a valid range of outcomes which can lead ViaSHAP's approximations to be on a different scale than the ground truth obtained using the unbiased KernelSHAP. However, since we measure the effect of using the link function on the accuracy of Shapley values, we can also apply a metric that measures the similarity on the same scale for models without a link function. Therefore, we also apply $R^2$ as a similarity metric to the ground truth Shapley values for models without link functions. The results presented in Table 12 demonstrate that ViaSHAP without a link function significantly outperforms its counterpart with a link function. In order to test the null hypothesis that no difference exists in the accuracy of Shapley value approximations by $KAN^{\mathcal{V}ia}$ with and without a link function, the Wilcoxon signed-rank test was applied. The test results confirm that the null hypothesis can be rejected in both cases, whether Spearman's rank or cosine similarity is used as the similarity metric. Furthermore, the results show that $R^2$ as a similarity metric is consistent with both Spearman's rank and cosine similarity.

*Table 11.* The effect of the link function on the predictive performance of $KAN^{\mathcal{V}ia}$ as measured by AUC. The best-performing model is colored in light green .

| Dataset | $KAN^{\mathcal{V}ia}$ (without a link function) | $KAN^{\mathcal{V}ia}$ (default settings) |
|---|---|---|
| Abalone | $0.883 \pm 0.0002$ | $0.87 \pm 0.003$ |
| Ada Prior | $0.898 \pm 0.003$ | $0.89 \pm 0.005$ |
| Adult | $0.919 \pm 0.0005$ | $0.914 \pm 0.003$ |
| Bank32nh | $0.883 \pm 0.003$ | $0.878 \pm 0.001$ |
| Electricity | $0.934 \pm 0.004$ | $0.93 \pm 0.004$ |
| Elevators | $0.936 \pm 0.002$ | $0.935 \pm 0.002$ |
| Fars | $0.958 \pm 0.001$ | $0.96 \pm 0.0003$ |
| Helena | $0.868 \pm 0.006$ | $0.884 \pm 0.0001$ |
| Heloc | $0.792 \pm 0.001$ | $0.788 \pm 0.002$ |
| Higgs | $0.801 \pm 0.001$ | $0.801 \pm 0.001$ |
| LHC Identify Jet | $0.939 \pm 0.0005$ | $0.944 \pm 0.0001$ |
| House 16H | $0.949 \pm 0.001$ | $0.949 \pm 0.0007$ |
| Indian Pines | $0.982 \pm 0.001$ | $0.985 \pm 0.0004$ |
| Jannis | $0.861 \pm 0.001$ | $0.864 \pm 0.001$ |
| JM1 | $0.686 \pm 0.024$ | $0.732 \pm 0.003$ |
| Magic Telescope | $0.921 \pm 0.002$ | $0.929 \pm 0.001$ |
| MC1 | $0.952 \pm 0.011$ | $0.94 \pm 0.003$ |
| Microaggregation2 | $0.764 \pm 0.008$ | $0.783 \pm 0.002$ |
| Mozilla4 | $0.965 \pm 0.001$ | $0.968 \pm 0.0008$ |
| Satellite | $0.944 \pm 0.01$ | $0.996 \pm 0.001$ |
| PC2 | $0.659 \pm 0.06$ | $0.827 \pm 0.009$ |
| Phonemes | $0.923 \pm 0.003$ | $0.946 \pm 0.003$ |
| Pollen | $0.501 \pm 0.002$ | $0.515 \pm 0.006$ |
| Telco Customer Churn | $0.857 \pm 0.003$ | $0.854 \pm 0.003$ |
| 1st order theorem proving | $0.810 \pm 0.006$ | $0.822 \pm 0.002$ |

*Table 12.* The effect of the link function on the similarity of the approximated Shapley values by $KAN^{Via}$. The best-performing model is colored in light green .

| Dataset | ViaSHAP with default settings | | ViaSHAP without a link function | | |
| --- | --- | --- | --- | --- | --- |
| | Cosine Similarity | Spearman's Rank | Cosine Similarity | Spearman's Rank | $R^2$ |
| Abalone | 0.969 ± 0.017 | 0.6635 ± 0.234 | 0.999 ± 0.0008 | 0.971 ± 0.052 | 0.999 ± 0.002 |
| Ada Prior | 0.935 ± 0.046 | 0.8763 ± 0.088 | 0.963 ± 0.037 | 0.909 ± 0.068 | 0.9 ± 0.095 |
| Adult | 0.931 ± 0.049 | 0.9594 ± 0.035 | 0.981 ± 0.03 | 0.931 ± 0.074 | 0.948 ± 0.079 |
| Bank32nh | 0.779 ± 0.163 | 0.432 ± 0.151 | 0.948 ± 0.045 | 0.648 ± 0.114 | 0.87 ± 0.142 |
| Electricity | 0.970 ± 0.02 | 0.7983 ± 0.183 | 0.998 ± 0.004 | 0.967 ± 0.043 | 0.992 ± 0.012 |
| Elevators | 0.966 ± 0.024 | 0.9203 ± 0.064 | 0.997 ± 0.004 | 0.969 ± 0.026 | 0.993 ± 0.009 |
| Fars | 0.886 ± 0.253 | 0.347 ± 0.328 | 0.962 ± 0.036 | 0.882 ± 0.073 | 0.895 ± 0.073 |
| Helena | 0.856 ± 0.092 | 0.669 ± 0.152 | 0.874 ± 0.095 | 0.702 ± 0.148 | 0.677 ± 0.204 |
| Heloc | 0.844 ± 0.111 | 0.7409 ± 0.147 | 0.962 ± 0.036 | 0.882 ± 0.073 | 0.895 ± 0.105 |
| Higgs | 0.917 ± 0.068 | 0.674 ± 0.12 | 0.991 ± 0.006 | 0.87 ± 0.057 | 0.977 ± 0.014 |
| LHC Identify Jet | 0.971 ± 0.021 | 0.8575 ± 0.119 | 0.999 ± 0.002 | 0.974 ± 0.032 | 0.998 ± 0.005 |
| House 16H | 0.919 ± 0.048 | 0.8876 ± 0.092 | 0.988 ± 0.015 | 0.952 ± 0.044 | 0.961 ± 0.057 |
| Indian Pines | 0.796 ± 0.121 | 0.6991 ± 0.116 | 0.683 ± 0.171 | 0.553 ± 0.18 | 0.333 ± 0.192 |
| Jannis | 0.852 ± 0.141 | 0.4775 ± 0.131 | 0.898 ± 0.072 | 0.624 ± 0.113 | 0.722 ± 0.183 |
| JM1 | 0.88 ± 0.044 | 0.7561 ± 0.202 | 0.965 ± 0.042 | 0.916 ± 0.085 | 0.901 ± 0.094 |
| Magic Telescope | 0.922 ± 0.067 | 0.9 ± 0.098 | 0.994 ± 0.006 | 0.959 ± 0.042 | 0.98 ± 0.02 |
| MC1 | 0.466 ± 0.268 | 0.6212 ± 0.157 | 0.951 ± 0.093 | 0.881 ± 0.139 | 0.873 ± 0.332 |
| Microaggregation2 | 0.938 ± 0.049 | 0.8756 ± 0.096 | 0.982 ± 0.021 | 0.957 ± 0.049 | 0.929 ± 0.114 |
| Mozilla4 | 0.953 ± 0.023 | 0.9423 ± 0.092 | 0.9998 ± 0.0003 | 0.967 ± 0.074 | 0.9996 ± 0.0007 |
| Satellite | 0.841 ± 0.116 | 0.746 ± 0.212 | 0.976 ± 0.033 | 0.894 ± 0.102 | 0.814 ± 0.296 |
| PC2 | 0.534 ± 0.183 | 0.7326 ± 0.161 | 0.956 ± 0.087 | 0.875 ± 0.127 | 0.895 ± 0.223 |
| Phonemes | 0.811 ± 0.162 | 0.9407 ± 0.103 | 0.993 ± 0.013 | 0.951 ± 0.094 | 0.975 ± 0.076 |
| Pollen | 0.952 ± 0.059 | 0.372 ± 0.429 | 0.994 ± 0.013 | 0.959 ± 0.076 | 0.929 ± 0.212 |
| Telco Customer Churn | 0.81 ± 0.108 | 0.8476 ± 0.098 | 0.978 ± 0.025 | 0.934 ± 0.052 | 0.939 ± 0.054 |
| 1st order theorem proving | 0.725 ± 0.179 | 0.6228 ± 0.188 | 0.778 ± 0.123 | 0.66 ± 0.146 | 0.429 ± 0.479 |

## J.4. The Effect of the Efficiency Constraint

We investigate the impact of the efficiency constraint (4) on the predictive performance of ViaSHAP and the similarity of its approximate Shapley values to the ground truth. The experimental results, presented in Tables 13 and 14, indicate that imposing the efficiency constraint has no significant effect on either the predictive performance of ViaSHAP or the accuracy of its explanations. To formally test this, we evaluate the null hypothesis that no significant difference exists in predictive performance, measured by AUC, between models trained with or without the efficiency constraint. Since only two approaches are compared, the Wilcoxon signed-rank test (Wilcoxon, 1945) is employed. The results confirm that the null hypothesis cannot be rejected at the 0.05 significance level, indicating no significant difference in predictive performance between the two approaches. Significance tests are also applied to evaluate the similarity of the approximated Shapley values to the ground truth based on cosine similarity, Spearman's rank, and $R^2$. The results indicate no significant difference between ViaSHAP models trained with and without the efficiency constraint.

*Table 13*. The effect of the efficiency constraint on the predictive performance of $KAN^{\mathcal{V}ia}$ as measured by AUC. The best-performing model is colored in light green .

| Dataset | Unconstrained | Constrained |
|---|---|---|
| Abalone | $0.883 \pm 0.0003$ | $0.883 \pm 0.0002$ |
| Ada Prior | $0.897 \pm 0.003$ | $0.898 \pm 0.003$ |
| Adult | $0.919 \pm 0.0007$ | $0.919 \pm 0.0005$ |
| Bank32nh | $0.884 \pm 0.002$ | $0.883 \pm 0.003$ |
| Electricity | $0.936 \pm 0.004$ | $0.934 \pm 0.004$ |
| Elevators | $0.933 \pm 0.002$ | $0.936 \pm 0.003$ |
| Fars | $0.959 \pm 0.001$ | $0.958 \pm 0.002$ |
| Helena | $0.870 \pm 0.005$ | $0.868 \pm 0.006$ |
| Heloc | $0.792 \pm 0.002$ | $0.792 \pm 0.001$ |
| Higgs | $0.800 \pm 0.002$ | $0.801 \pm 0.001$ |
| LHC Identify Jet | $0.939 \pm 0.0006$ | $0.939 \pm 0.0005$ |
| House 16H | $0.948 \pm 0.001$ | $0.949 \pm 0.001$ |
| Indian Pines | $0.982 \pm 0.002$ | $0.982 \pm 0.001$ |
| Jannis | $0.860 \pm 0.003$ | $0.861 \pm 0.001$ |
| JM1 | $0.691 \pm 0.026$ | $0.686 \pm 0.025$ |
| Magic Telescope | $0.921 \pm 0.002$ | $0.921 \pm 0.002$ |
| MC1 | $0.942 \pm 0.011$ | $0.952 \pm 0.011$ |
| Microaggregation2 | $0.763 \pm 0.009$ | $0.764 \pm 0.008$ |
| Mozilla4 | $0.965 \pm 0.001$ | $0.965 \pm 0.001$ |
| Satellite | $0.926 \pm 0.006$ | $0.944 \pm 0.010$ |
| PC2 | $0.670 \pm 0.046$ | $0.659 \pm 0.060$ |
| Phonemes | $0.919 \pm 0.006$ | $0.923 \pm 0.003$ |
| Pollen | $0.499 \pm 0.002$ | $0.501 \pm 0.002$ |
| Telco Customer Churn | $0.853 \pm 0.004$ | $0.857 \pm 0.003$ |
| 1st order theorem proving | $0.809 \pm \pm 0.007$ | $0.810 \pm 0.006$ |

*Table 14.* The similarity to the ground truth explanations when $KAN^{\mathcal{V}ia}$ optimized with and without the efficiency constraint. The similarity is measured using cosine similarity, Spearman's rank, and $R^2$. The best-performing model is colored in light green.

| Dataset | Cosine Similarity | | Spearman's Rank | | $R^2$ | |
|---|---|---|---|---|---|---|
| | Unconstrained | Constrained | Unconstrained | Constrained | Unconstrained | Constrained |
| Abalone | 0.999 ± 0.001 | 0.999 ± 0.001 | 0.972 ± 0.056 | 0.971 ± 0.052 | 0.999 ± 0.002 | 0.999 ± 0.002 |
| Ada Prior | 0.949 ± 0.047 | 0.963 ± 0.037 | 0.909 ± 0.067 | 0.909 ± 0.068 | 0.8461 ± 0.138 | 0.9 ± 0.095 |
| Adult | 0.987 ± 0.023 | 0.981 ± 0.03 | 0.955 ± 0.048 | 0.931 ± 0.074 | 0.966 ± 0.062 | 0.948 ± 0.079 |
| Bank32nh | 0.945 ± 0.045 | 0.948 ± 0.045 | 0.654 ± 0.117 | 0.648 ± 0.114 | 0.849 ± 0.125 | 0.87 ± 0.142 |
| Electricity | 0.997 ± 0.005 | 0.998 ± 0.004 | 0.964 ± 0.048 | 0.967 ± 0.043 | 0.991 ± 0.015 | 0.992 ± 0.012 |
| Elevators | 0.996 ± 0.004 | 0.997 ± 0.004 | 0.967 ± 0.03 | 0.969 ± 0.026 | 0.991 ± 0.011 | 0.993 ± 0.009 |
| Fars | 0.998 ± 0.008 | 0.962 ± 0.036 | 0.908 ± 0.076 | 0.882 ± 0.073 | 0.994 ± 0.022 | 0.895 ± 0.073 |
| Helena | 0.888 ± 0.085 | 0.874 ± 0.095 | 0.719 ± 0.139 | 0.702 ± 0.148 | 0.713 ± 0.187 | 0.677 ± 0.204 |
| Heloc | 0.96 ± 0.037 | 0.962 ± 0.036 | 0.862 ± 0.084 | 0.882 ± 0.073 | 0.894 ± 0.097 | 0.895 ± 0.105 |
| Higgs | 0.993 ± 0.004 | 0.991 ± 0.006 | 0.88 ± 0.055 | 0.87 ± 0.057 | 0.984 ± 0.011 | 0.977 ± 0.014 |
| LHC Identify Jet | 0.999 ± 0.002 | 0.999 ± 0.002 | 0.97 ± 0.038 | 0.974 ± 0.032 | 0.998 ± 0.006 | 0.998 ± 0.005 |
| House 16H | 0.99 ± 0.012 | 0.988 ± 0.015 | 0.959 ± 0.037 | 0.952 ± 0.044 | 0.969 ± 0.032 | 0.961 ± 0.057 |
| Indian Pines | 0.663 ± 0.164 | 0.683 ± 0.171 | 0.518 ± 0.184 | 0.553 ± 0.18 | 0.297 ± 0.178 | 0.333 ± 0.192 |
| Jannis | 0.903 ± 0.071 | 0.898 ± 0.072 | 0.649 ± 0.122 | 0.624 ± 0.113 | 0.727 ± 0.191 | 0.722 ± 0.183 |
| JM1 | 0.95 ± 0.041 | 0.965 ± 0.042 | 0.888 ± 0.102 | 0.916 ± 0.085 | 0.851 ± 0.123 | 0.901 ± 0.094 |
| Magic Telescope | 0.995 ± 0.005 | 0.994 ± 0.006 | 0.964 ± 0.039 | 0.959 ± 0.042 | 0.985 ± 0.017 | 0.98 ± 0.02 |
| MC1 | 0.941 ± 0.103 | 0.951 ± 0.093 | 0.862 ± 0.145 | 0.881 ± 0.139 | 0.869 ± 0.239 | 0.873 ± 0.332 |
| Microaggregation2 | 0.986 ± 0.021 | 0.982 ± 0.021 | 0.963 ± 0.05 | 0.957 ± 0.049 | 0.956 ± 0.061 | 0.929 ± 0.114 |
| Mozilla4 | 0.9998 ± 0.001 | 0.9998 ± 0.0003 | 0.965 ± 0.074 | 0.967 ± 0.074 | 0.9996 ± 0.002 | 0.9996 ± 0.001 |
| Satellite | 0.965 ± 0.043 | 0.976 ± 0.033 | 0.858 ± 0.129 | 0.894 ± 0.102 | 0.823 ± 0.195 | 0.814 ± 0.296 |
| PC2 | 0.961 ± 0.085 | 0.956 ± 0.087 | 0.889 ± 0.134 | 0.875 ± 0.127 | 0.913 ± 0.206 | 0.895 ± 0.223 |
| Phonemes | 0.994 ± 0.016 | 0.993 ± 0.013 | 0.964 ± 0.086 | 0.951 ± 0.094 | 0.98 ± 0.063 | 0.975 ± 0.076 |
| Pollen | 0.996 ± 0.016 | 0.994 ± 0.013 | 0.954 ± 0.095 | 0.959 ± 0.076 | 0.967 ± 0.107 | 0.929 ± 0.212 |
| Telco Customer Churn | 0.974 ± 0.02 | 0.978 ± 0.025 | 0.911 ± 0.051 | 0.934 ± 0.052 | 0.923 ± 0.051 | 0.939 ± 0.054 |
| 1st order theorem proving | 0.786 ± 0.128 | 0.778 ± 0.123 | 0.669 ± 0.156 | 0.66 ± 0.146 | 0.464 ± 0.21 | 0.429 ± 0.479 |

## J.5. The Progress of Training and Validation Losses

In this subsection, we report the progression of training and validation losses with different values of the hyperparameter $\beta$ using six datasets. A common trend observed across models trained on the six datasets is that, with different values of $\beta$, the Shapley loss (scaled by $\beta$) consistently decreases quickly below the level of the classification loss, except for the First Order Theorem Proving dataset (Figure 15), which is a multinomial classification dataset. For the First Order Theorem Proving dataset, the Shapley loss remains at a scale determined by the $\beta$ factor throughout the training time. However, the model for the First Order Theorem Proving dataset can still learn a function that estimates Shapley values with good accuracy, as shown in Tables 2 and 3. Moreover, it benefits from larger $\beta$ values to achieve accurate Shapley value approximations, as illustrated in Figure 10. Additionally, the results indicate that ViaSHAP generally tends to take longer to converge as $\beta$ values increase.

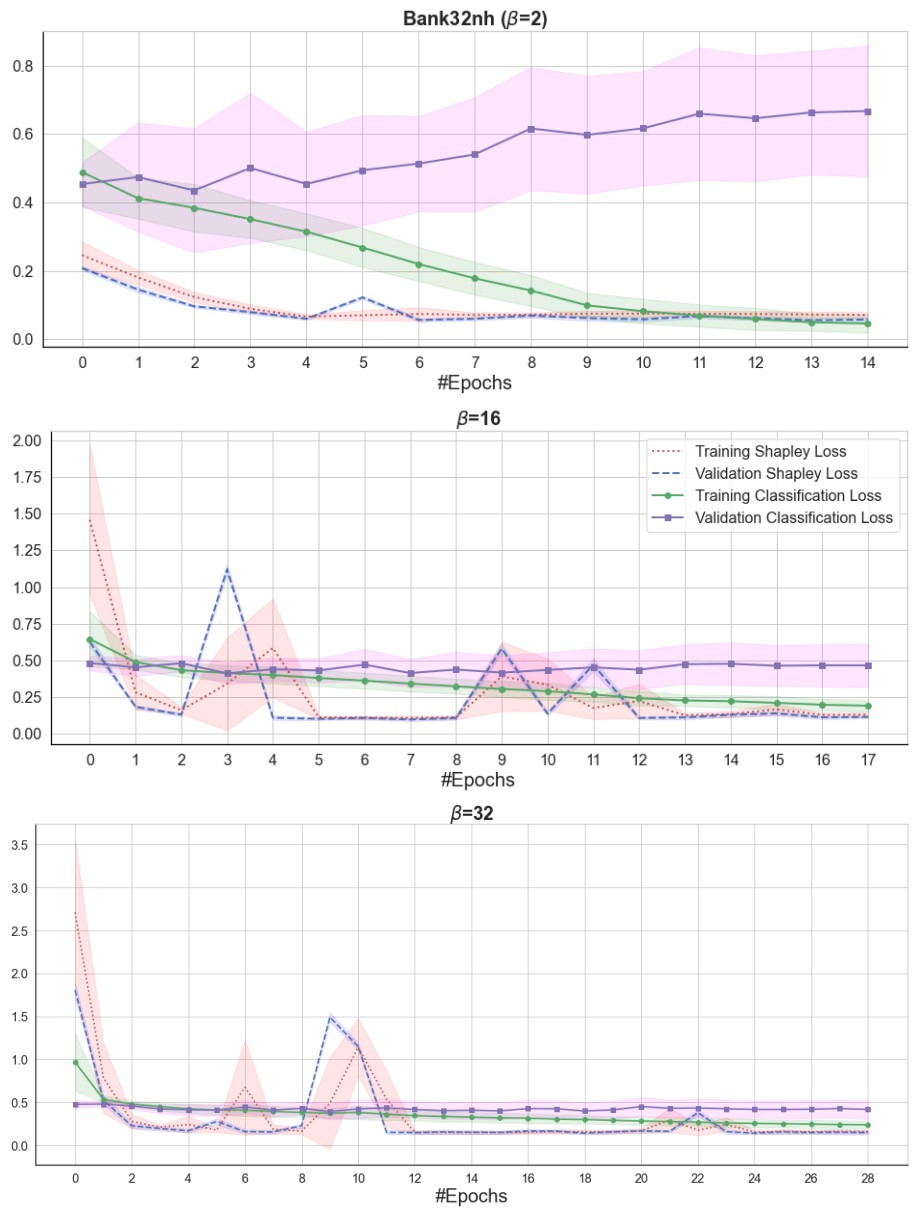

*Figure 14.* The effect of $\beta$ value on the progress of the training and the validation loss values.

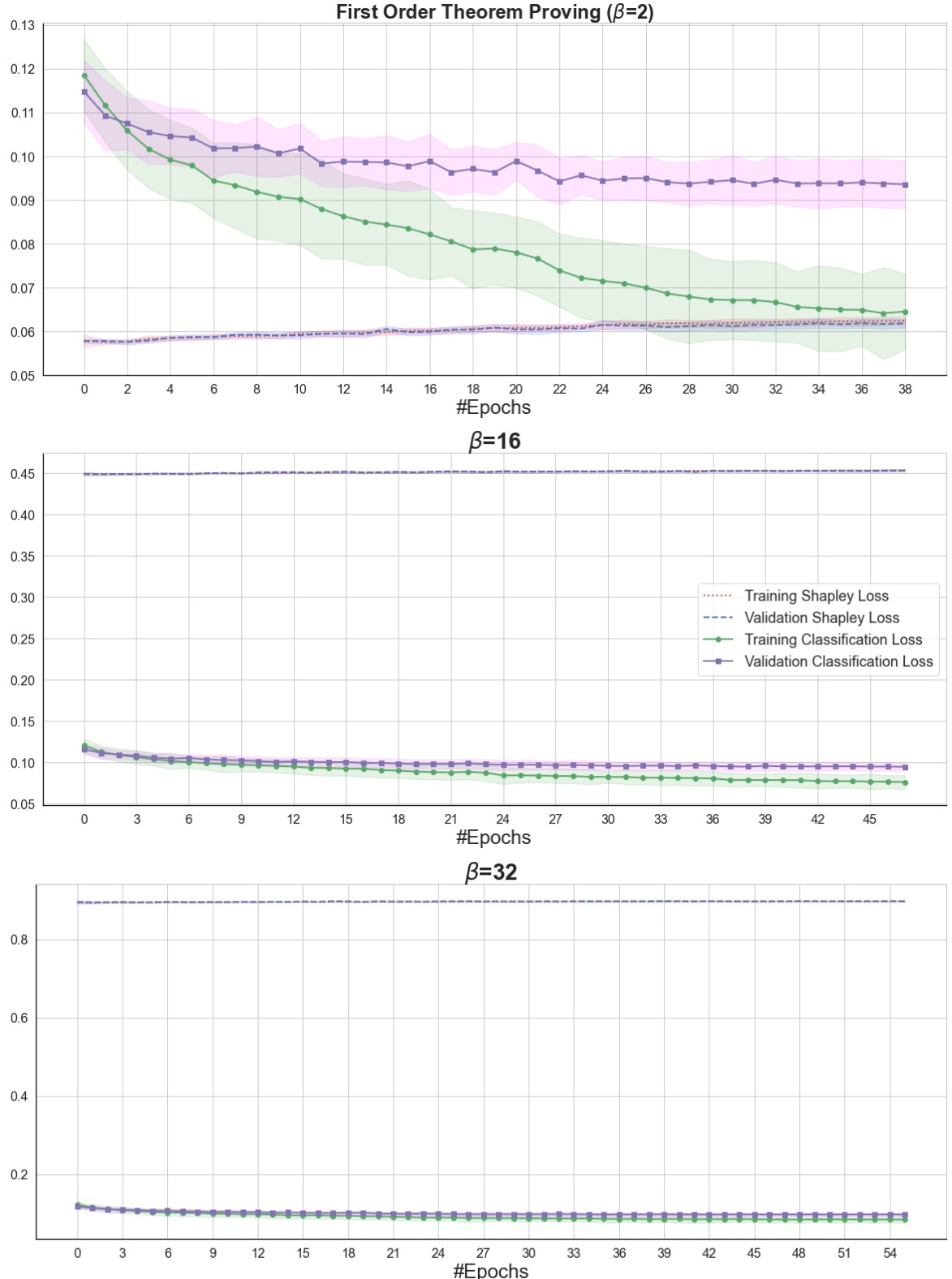

*Figure 15.* The effect of $\beta$ value on the progress of the training and the validation loss values.

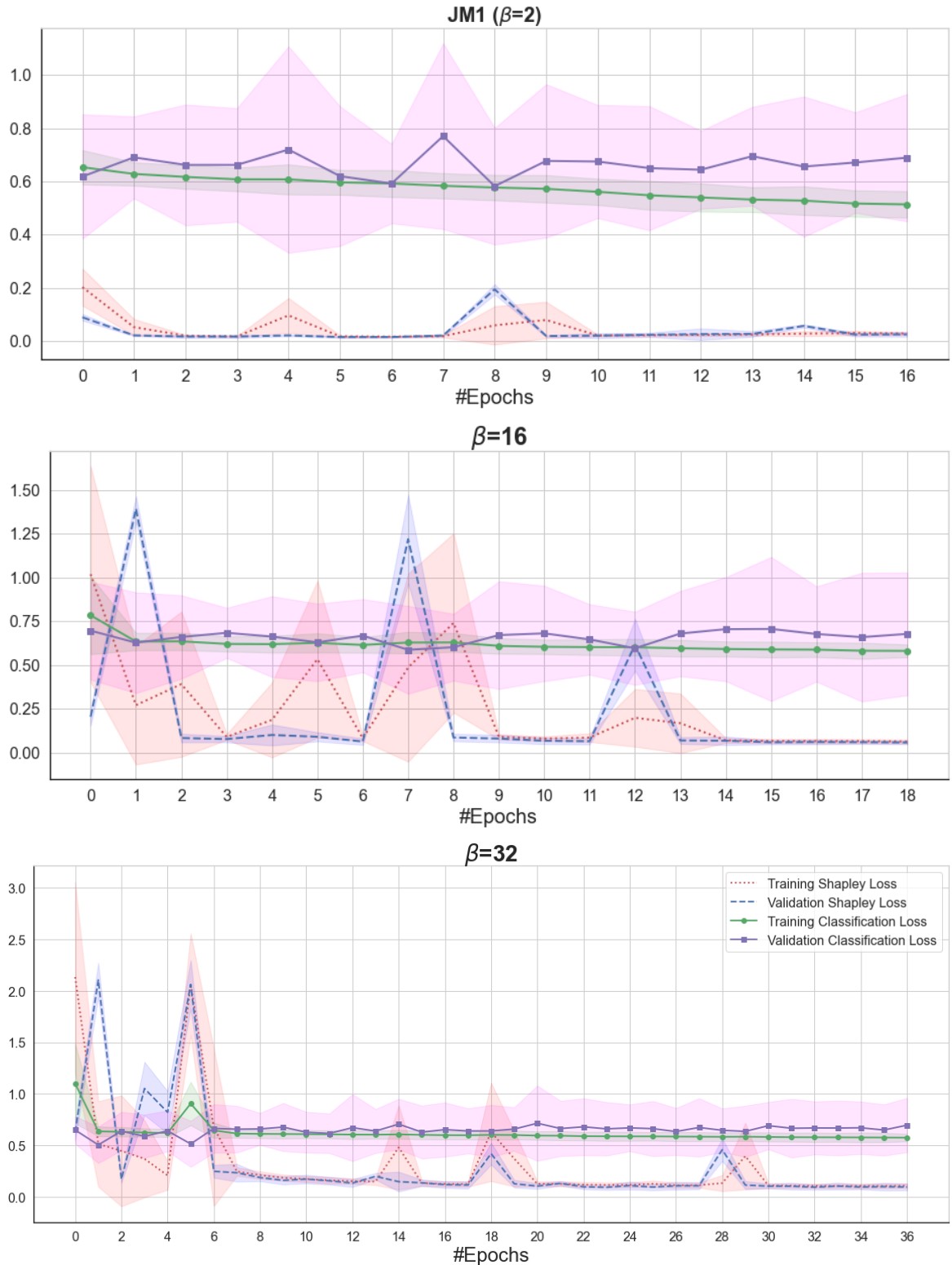

*Figure 16.* The effect of $\beta$ value on the progress of the training and the validation loss values.

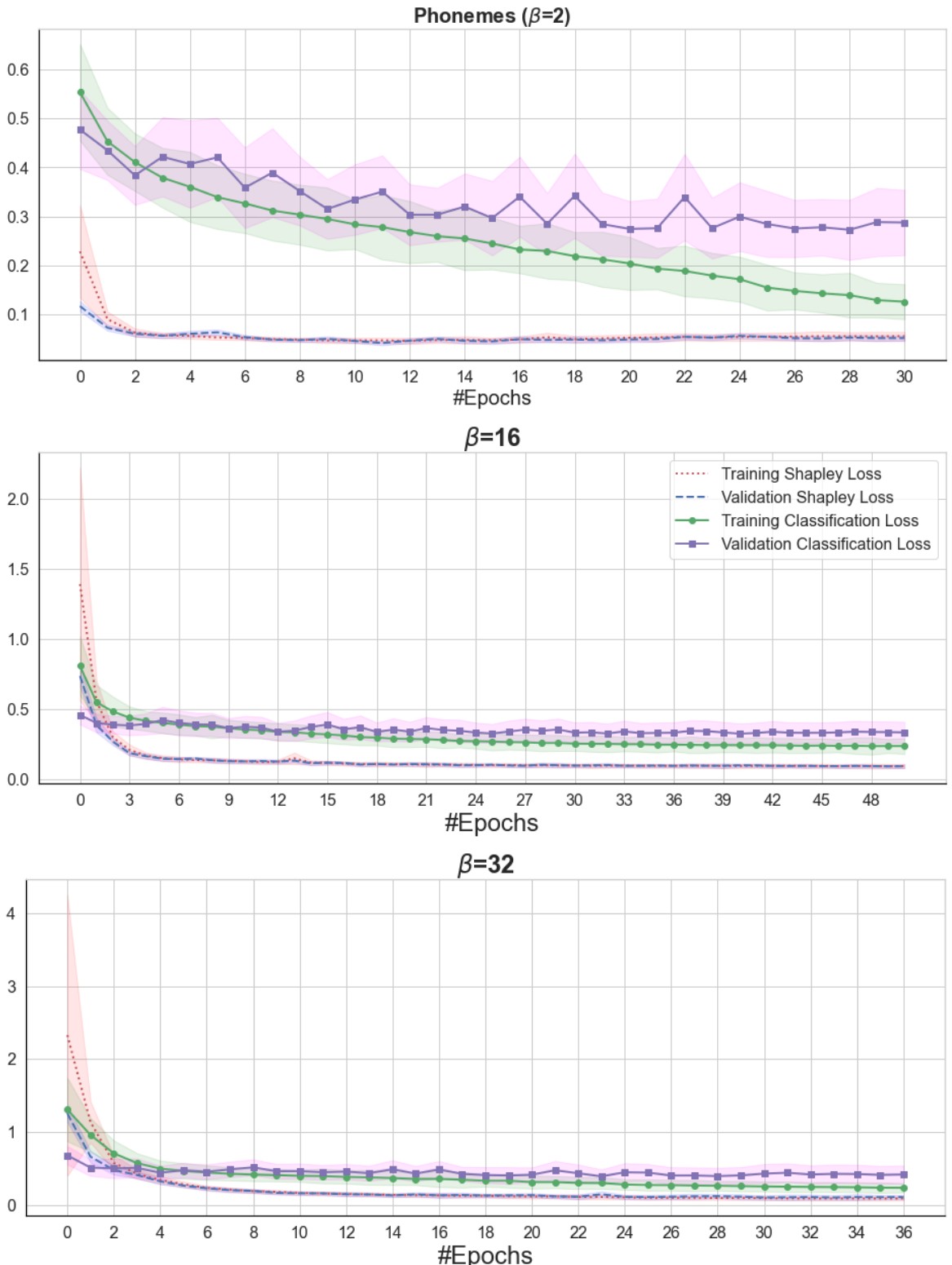

*Figure 17.* The effect of $\beta$ value on the progress of the training and the validation loss values.

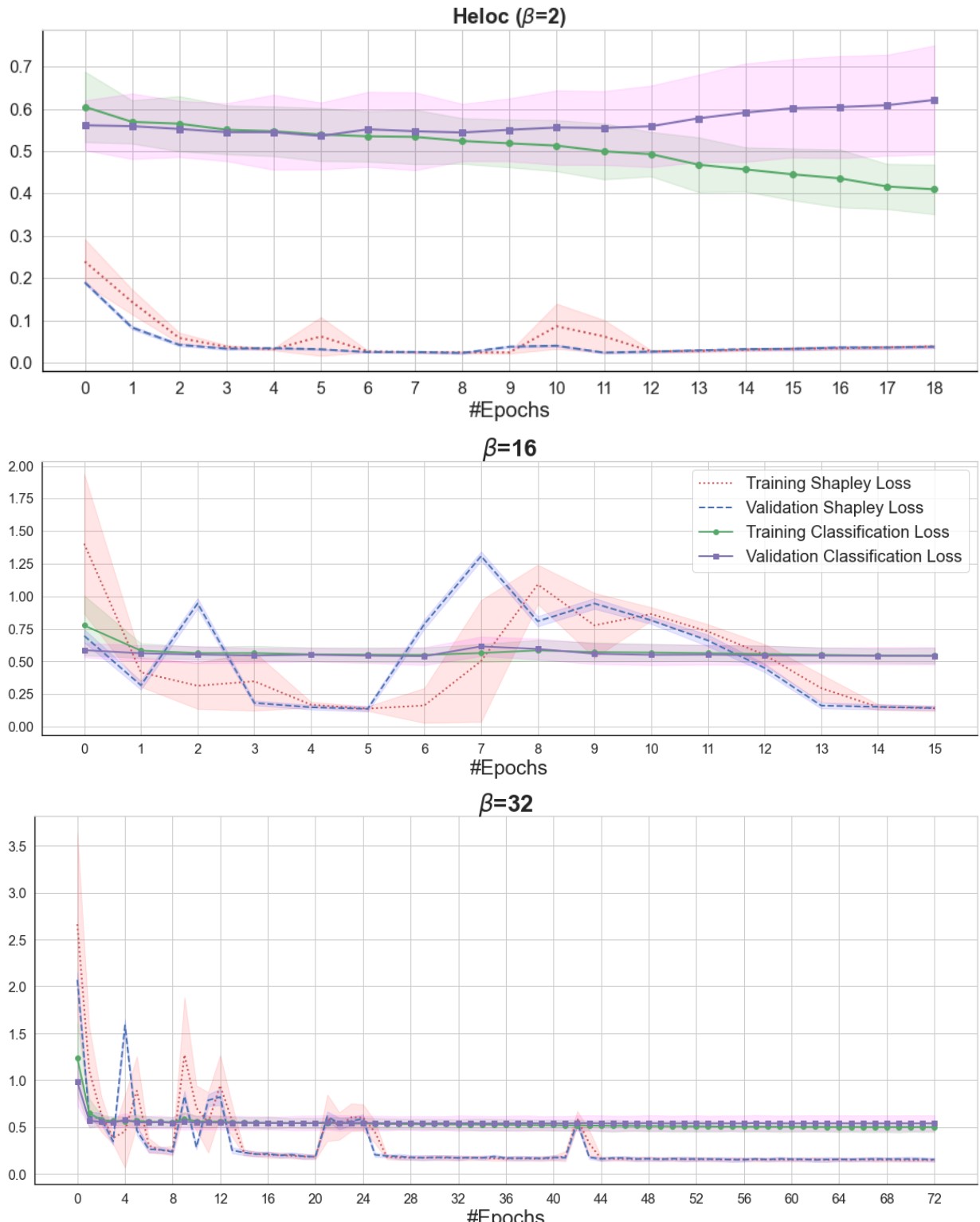

*Figure 18.* The effect of $\beta$ value on the progress of the training and the validation loss values.

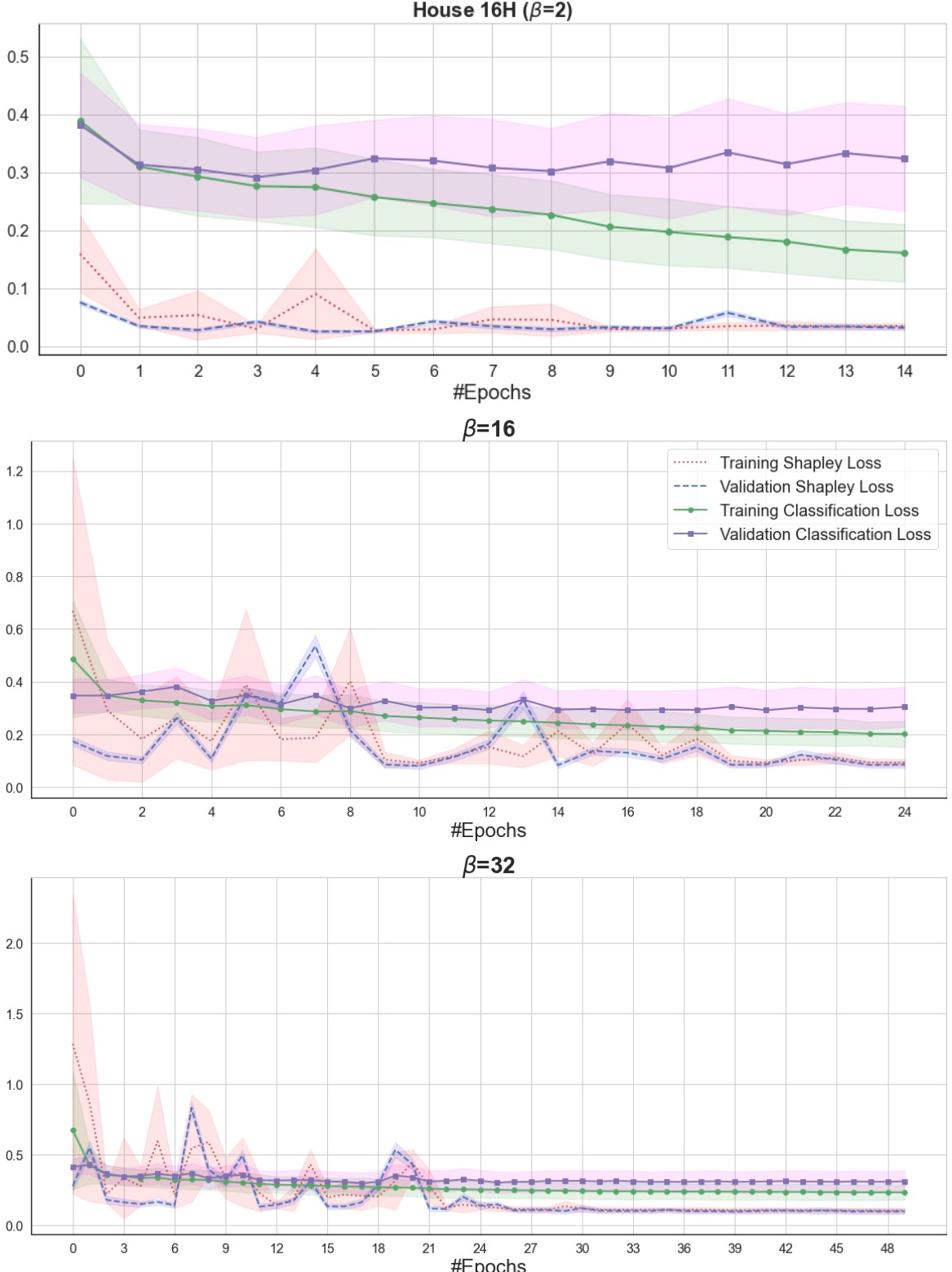

*Figure 19.* The effect of $\beta$ value on the progress of the training and the validation loss values.

## K. A Comparison Between ViaSHAP and FastSHAP

We compared the accuracy of ViaSHAP's Shapley value approximations to FastSHAP, using ViaSHAP as a black-box model within the FastSHAP framework. ViaSHAP is implemented using $KAN^{\mathcal{V}ia}$ without a link function, while FastSHAP is using the default settings. The evaluation employed metrics such as $R^2$, cosine similarity, and Spearman's rank correlation to measure the similarity between the computed Shapley values and the ground truth. The results demonstrate that ViaSHAP achieves significantly higher similarity to the ground truth compared to FastSHAP. This conclusion is supported by the Wilcoxon signed-rank test, which enabled rejection of the null hypothesis that there is no difference in similarity to the ground truth Shapley values between ViaSHAP and FastSHAP. The test confirmed significant differences using all evaluated similarity metrics, including $R^2$, cosine similarity, and Spearman's rank correlation. The detailed results are available in Table 16.

## L. A Comparison Between the Inference Time of ViaSHAP and KernelSHAP

In Table 15, we report the time required to explain 1000 instances using KernelSHAP and ViaSHAP ($KAN^{\mathcal{V}ia}$) on six datasets using an NVIDIA Tesla V100f GPU and 16 cores of an Intel Xeon Gold 6338 processor.

*Table 15.* The time (in seconds) required to explain 1000 predictions from 6 different datasets using KernelSHAP and ViaSHAP.

| Dataset | KernelSHAP | $KAN^{\mathcal{V}ia}$ |
|---|---|---|
| Adult | 56.92 | 0.0026 |
| Elevators | 54.22 | 0.0021 |
| House 16 | 53.12 | 0.0052 |
| Indian Pines | 43124.66 | 0.0023 |
| Microaggregation 2 | 79.97 | 0.0022 |
| First order proving theorem | 436.25 | 0.0022 |

## M. Limitations of ViaSHAP

ViaSHAP operates under the assumption that the selected base model can be optimized using backpropagation. Hence, models that employ other optimization algorithms, such as decision trees, are not suitable for this approach. Nevertheless, ViaSHAP can be extended to work with methods that are not based on backpropagation. For example, we can train one regressor per dimension of $\phi^{\mathcal{V}ia}(x)$.

The empirical results presented in Appendix H indicate that ViaSHAP does not yield accurate models when trained using the marginal expectations value function, which requires further investigation. Furthermore, as demonstrated in Appendix D, ViaSHAP does not produce accurate predictors when trained using a small-sized MLP consisting of two hidden layers.

*Table 16.* A comparison between ViaSHAP and FastSHAP with respect to the similarity of the approximated Shapley values to the ground truth values. The best-performing model is  colored in light green .

| Dataset | Cosine Similarity | | Spearman's Rank | | $R^2$ | |
|---|---|---|---|---|---|---|
| | ViaSHAP | FastSHAP | ViaSHAP | FastSHAP | ViaSHAP | FastSHAP |
| Abalone | 0.999 ± 0.0008 | 0.999 ± 0.002 | 0.971 ± 0.052 | 0.966 ± 0.055 | 0.999 ± 0.002 | 0.996 ± 0.008 |
| Ada Prior | 0.963 ± 0.037 | 0.703 ± 0.25 | 0.909 ± 0.068 | 0.64 ± 0.24 | 0.887 ± 0.105 | 0.042 ± 1.359 |
| Adult | 0.981 ± 0.03 | 0.956 ± 0.072 | 0.931 ± 0.074 | 0.893 ± 0.115 | 0.952 ± 0.072 | 0.853 ± 0.298 |
| Bank32nh | 0.948 ± 0.045 | 0.897 ± 0.079 | 0.648 ± 0.114 | 0.527 ± 0.133 | 0.852 ± 0.161 | 0.728 ± 0.29 |
| Electricity | 0.998 ± 0.004 | 0.978 ± 0.06 | 0.967 ± 0.043 | 0.921 ± 0.101 | 0.993 ± 0.011 | 0.914 ± 0.306 |
| Elevators | 0.997 ± 0.004 | 0.994 ± 0.006 | 0.969 ± 0.026 | 0.941 ± 0.047 | 0.993 ± 0.009 | 0.983 ± 0.023 |
| Fars | 0.997 ± 0.008 | 0.997 ± 0.021 | 0.849 ± 0.098 | 0.834 ± 0.124 | 0.994 ± 0.022 | 0.991 ± 0.028 |
| Helena | 0.874 ± 0.095 | 0.822 ± 0.139 | 0.702 ± 0.148 | 0.6 ± 0.193 | 0.677 ± 0.204 | 0.532 ± 0.29 |
| Heloc | 0.962 ± 0.036 | 0.935 ± 0.064 | 0.882 ± 0.073 | 0.826 ± 0.111 | 0.894 ± 0.098 | 0.824 ± 0.177 |
| Higgs | 0.991 ± 0.006 | 0.994 ± 0.004 | 0.87 ± 0.057 | 0.899 ± 0.049 | 0.977 ± 0.014 | 0.986 ± 0.01 |
| LHC Identify Jet | 0.999 ± 0.002 | 0.999 ± 0.003 | 0.974 ± 0.032 | 0.971 ± 0.035 | 0.998 ± 0.005 | 0.997 ± 0.016 |
| House 16H | 0.988 ± 0.015 | 0.964 ± 0.035 | 0.952 ± 0.044 | 0.891 ± 0.09 | 0.964 ± 0.039 | 0.89 ± 0.107 |
| Indian Pines | 0.683 ± 0.171 | 0.423 ± 0.154 | 0.553 ± 0.18 | 0.204 ± 0.122 | 0.333 ± 0.192 | -0.615 ± 0.912 |
| Jannis | 0.898 ± 0.072 | 0.92 ± 0.064 | 0.624 ± 0.113 | 0.673 ± 0.106 | 0.722 ± 0.183 | 0.776 ± 0.179 |
| JM1 | 0.965 ± 0.042 | 0.98 ± 0.042 | 0.916 ± 0.085 | 0.934 ± 0.083 | 0.887 ± 0.206 | 0.925 ± 0.37 |
| Magic Telescope | 0.994 ± 0.006 | 0.984 ± 0.023 | 0.959 ± 0.042 | 0.918 ± 0.084 | 0.98 ± 0.021 | 0.946 ± 0.094 |
| MC1 | 0.951 ± 0.093 | 0.789 ± 0.254 | 0.881 ± 0.139 | 0.638 ± 0.297 | 0.881 ± 0.346 | -0.024 ± 9.964 |
| Microaggregation2 | 0.982 ± 0.021 | 0.99 ± 0.017 | 0.957 ± 0.049 | 0.97 ± 0.041 | 0.944 ± 0.061 | 0.966 ± 0.054 |
| Mozilla4 | 0.9998 ± 0.0003 | 0.994 ± 0.017 | 0.967 ± 0.074 | 0.921 ± 0.141 | 0.9996 ± 0.0007 | 0.984 ± 0.049 |
| Satellite | 0.976 ± 0.033 | 0.858 ± 0.114 | 0.894 ± 0.102 | 0.55 ± 0.25 | 0.873 ± 0.151 | 0.126 ± 0.793 |
| PC2 | 0.956 ± 0.087 | 0.786 ± 0.234 | 0.875 ± 0.127 | 0.619 ± 0.249 | 0.891 ± 0.272 | 0.274 ± 1.616 |
| Phonemes | 0.993 ± 0.013 | 0.981 ± 0.036 | 0.951 ± 0.094 | 0.946 ± 0.096 | 0.971 ± 0.071 | 0.925 ± 0.165 |
| Pollen | 0.994 ± 0.013 | 0.984 ± 0.024 | 0.959 ± 0.076 | 0.905 ± 0.129 | 0.933 ± 0.276 | 0.855 ± 0.23 |
| Telco Customer Churn | 0.978 ± 0.025 | 0.963 ± 0.045 | 0.934 ± 0.052 | 0.892 ± 0.087 | 0.924 ± 0.085 | 0.899 ± 0.109 |
| 1st order theorem proving | 0.778 ± 0.123 | 0.776 ± 0.174 | 0.66 ± 0.146 | 0.658 ± 0.206 | 0.429 ± 0.479 | 0.367 ± 2.832 |

# N. Computational Cost

The experiments were conducted using an NVIDIA Tesla V100f GPU and 16 cores of an Intel Xeon Gold 6338 processor. The training time required for both $KAN^{Via}$ and $MLP^{Via}$ are recorded on 1,000 data examples with varying numbers of coalitions (Table 17). The inference time is also recorded on 1,000 data example for both $KAN^{Via}$ and $MLP^{Via}$ as shown in Table 18. All the results are reported as the mean and standard deviation across five different runs. Generally, $MLP^{Via}$ is faster than $KAN^{Via}$ in both training and inference. Additionally, while the number of samples per data example increased exponentially, the computational cost during training did not rise at the same rate, as depicted in Figure 20.

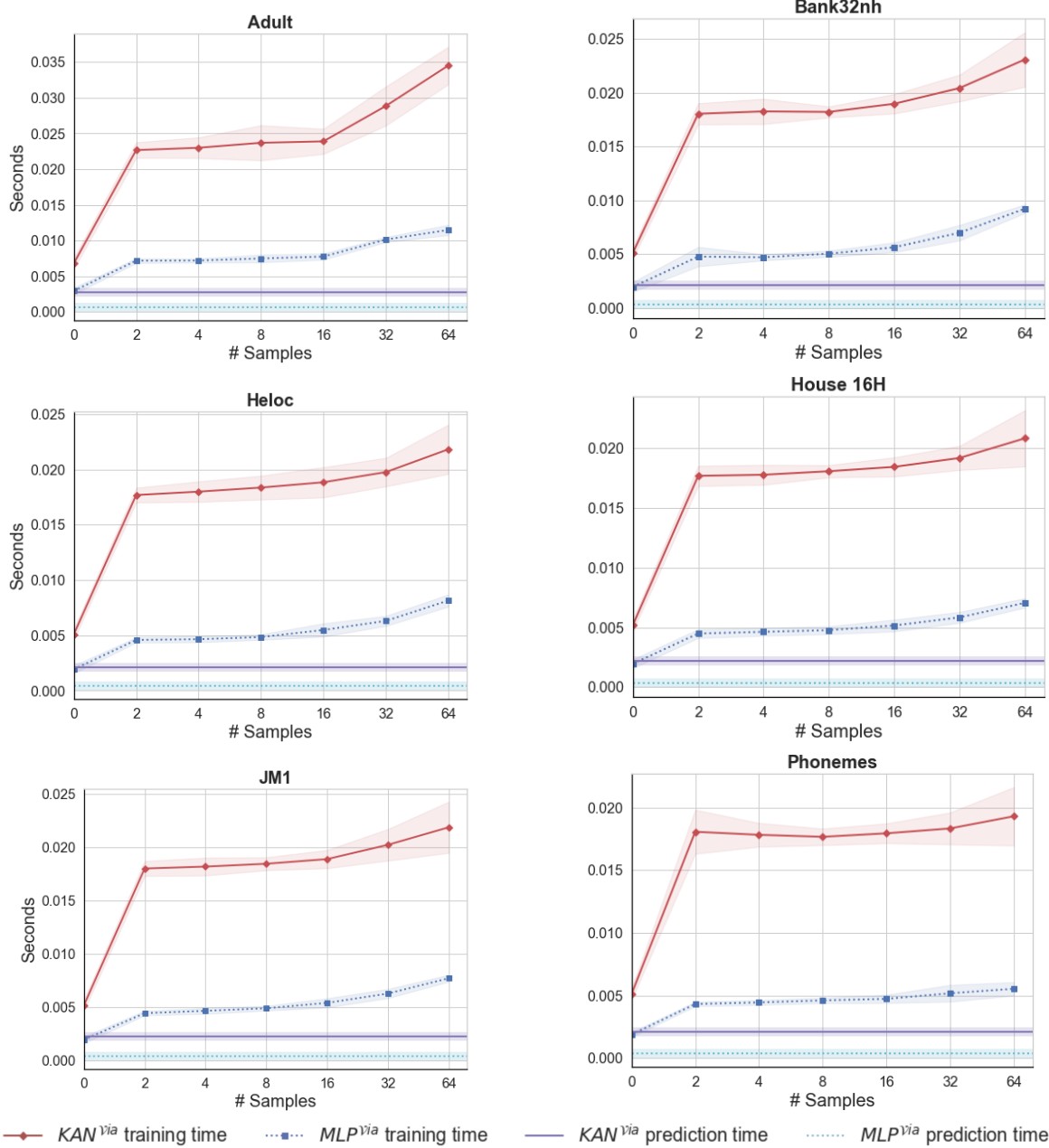

Figure 20. The training time and prediction time on 1000 data instance of $KAN^{Via}$ and $MLP^{Via}$.

Table 17. The training time in seconds for 1000 data instances using $KAN^{\mathcal{V}ia}$ and $MLP^{\mathcal{V}ia}$.

| Dataset | $KAN^{\mathcal{V}ia}$ | | | | $MLP^{\mathcal{V}ia}$ | | | |
|---|---|---|---|---|---|---|---|---|
| | No Sampling | 2 Samples | 16 Samples | 32 Samples | No Sampling | 2 Samples | 16 Samples | 32 Samples |
| Abalone | 0.0058 ± 0.0005 | 0.0208 ± 0.0013 | 0.0209 ± 0.0009 | 0.0228 ± 0.0016 | 0.002 ± 0.0002 | 0.0053 ± 0.0007 | 0.0056 ± 0.0005 | 0.0062 ± 0.0006 |
| Ada Prior | 0.0068 ± 0.0005 | 0.024 ± 0.0014 | 0.0235 ± 0.0017 | 0.0284 ± 0.0021 | 0.0028 ± 0.0003 | 0.0071 ± 0.0003 | 0.0079 ± 0.0005 | 0.0104 ± 0.0007 |
| Adult | 0.0068 ± 0.0008 | 0.0227 ± 0.0011 | 0.0239 ± 0.0018 | 0.0288 ± 0.0027 | 0.003 ± 0.0005 | 0.0072 ± 0.0003 | 0.0078 ± 0.0004 | 0.0101 ± 0.0004 |
| Bank32nh | 0.0051 ± 0.0005 | 0.018 ± 0.001 | 0.019 ± 0.0009 | 0.0204 ± 0.0012 | 0.002 ± 0.0005 | 0.0048 ± 0.0009 | 0.0056 ± 0.0005 | 0.007 ± 0.0007 |
| Electricity | 0.0059 ± 0.0005 | 0.0209 ± 0.0012 | 0.0211 ± 0.0013 | 0.023 ± 0.0017 | 0.0021 ± 0.0004 | 0.0051 ± 0.0004 | 0.0056 ± 0.0005 | 0.0061 ± 0.0005 |
| Elevators | 0.0051 ± 0.0005 | 0.0177 ± 0.0007 | 0.0187 ± 0.0013 | 0.0193 ± 0.0011 | 0.0019 ± 0.0004 | 0.0044 ± 0.0003 | 0.0052 ± 0.0005 | 0.0059 ± 0.0004 |
| Fars | 0.009 ± 0.0006 | 0.0262 ± 0.0009 | 0.0274 ± 0.0017 | 0.0358 ± 0.0017 | 0.0064 ± 0.0004 | 0.0102 ± 0.0004 | 0.0114 ± 0.0005 | 0.0153 ± 0.0005 |
| Helena | 0.0054 ± 0.0007 | 0.0184 ± 0.0012 | 0.0195 ± 0.0014 | 0.0208 ± 0.002 | 0.0021 ± 0.0005 | 0.0048 ± 0.0004 | 0.0061 ± 0.0008 | 0.0073 ± 0.0008 |
| Heloc | 0.0051 ± 0.0005 | 0.0177 ± 0.0007 | 0.0188 ± 0.0014 | 0.0198 ± 0.0013 | 0.0019 ± 0.0004 | 0.0046 ± 0.0002 | 0.0055 ± 0.0006 | 0.0063 ± 0.0004 |
| Higgs | 0.0067 ± 0.0003 | 0.019 ± 0.0005 | 0.0198 ± 0.0006 | 0.0211 ± 0.0013 | 0.0021 ± 0.0002 | 0.005 ± 0.0002 | 0.0067 ± 0.0004 | 0.0075 ± 0.0005 |
| LHC Identify Jet | 0.0055 ± 0.0003 | 0.0184 ± 0.0007 | 0.0187 ± 0.0008 | 0.0198 ± 0.0012 | 0.0022 ± 0.0003 | 0.0045 ± 0.0004 | 0.0058 ± 0.0004 | 0.0064 ± 0.0003 |
| House 16H | 0.0052 ± 0.0006 | 0.0176 ± 0.0008 | 0.0238 ± 0.0075 | 0.0195 ± 0.0018 | 0.002 ± 0.0004 | 0.0058 ± 0.0004 | 0.0052 ± 0.0005 | 0.0058 ± 0.0005 |
| Indian Pines | 0.0054 ± 0.0006 | 0.0194 ± 0.0011 | 0.0268 ± 0.0022 | 0.0355 ± 0.0026 | 0.0021 ± 0.0004 | 0.0053 ± 0.0002 | 0.0129 ± 0.0006 | 0.0208 ± 0.0009 |
| Jannis | 0.0073 ± 0.0006 | 0.0195 ± 0.0006 | 0.0214 ± 0.001 | 0.0235 ± 0.0016 | 0.0022 ± 0.0003 | 0.0044 ± 0.0002 | 0.0073 ± 0.0005 | 0.0096 ± 0.0002 |
| JM1 | 0.0051 ± 0.0005 | 0.018 ± 0.0007 | 0.0189 ± 0.0009 | 0.0202 ± 0.0015 | 0.0019 ± 0.0003 | 0.0046 ± 0.0007 | 0.0054 ± 0.0004 | 0.0063 ± 0.0004 |
| MagicTelescope | 0.0051 ± 0.0005 | 0.0178 ± 0.0009 | 0.0183 ± 0.001 | 0.0188 ± 0.0012 | 0.0019 ± 0.0002 | 0.0045 ± 0.0003 | 0.005 ± 0.0005 | 0.0054 ± 0.0005 |
| MC1 | 0.0051 ± 0.0005 | 0.0182 ± 0.0008 | 0.0194 ± 0.0008 | 0.0209 ± 0.0014 | 0.0021 ± 0.0007 | 0.0045 ± 0.0002 | 0.0063 ± 0.0004 | 0.0076 ± 0.0007 |
| Microaggregation 2 | 0.0053 ± 0.0007 | 0.0187 ± 0.0011 | 0.0189 ± 0.001 | 0.0198 ± 0.0013 | 0.0019 ± 0.0003 | 0.0045 ± 0.0002 | 0.0055 ± 0.0004 | 0.0062 ± 0.0004 |
| Mozilla4 | 0.0051 ± 0.0006 | 0.0178 ± 0.0007 | 0.0184 ± 0.0015 | 0.0188 ± 0.0016 | 0.0019 ± 0.0003 | 0.0044 ± 0.0003 | 0.0048 ± 0.0003 | 0.005 ± 0.0004 |
| Satellite | 0.0051 ± 0.0005 | 0.018 ± 0.001 | 0.0197 ± 0.002 | 0.0207 ± 0.0015 | 0.002 ± 0.0004 | 0.0046 ± 0.0003 | 0.006 ± 0.0007 | 0.0073 ± 0.0005 |
| PC2 | 0.005 ± 0.0004 | 0.0178 ± 0.0007 | 0.0192 ± 0.0009 | 0.0209 ± 0.0019 | 0.0019 ± 0.0003 | 0.0045 ± 0.0002 | 0.0059 ± 0.0005 | 0.0073 ± 0.0006 |
| Phonemes | 0.0051 ± 0.0005 | 0.0181 ± 0.0018 | 0.0179 ± 0.0008 | 0.0183 ± 0.0013 | 0.0019 ± 0.0002 | 0.0043 ± 0.0002 | 0.0047 ± 0.0003 | 0.0052 ± 0.0007 |
| Pollen | 0.005 ± 0.0004 | 0.018 ± 0.0015 | 0.0181 ± 0.0013 | 0.0185 ± 0.0015 | 0.0019 ± 0.0004 | 0.0044 ± 0.0003 | 0.0048 ± 0.0004 | 0.005 ± 0.0004 |
| Telco Customer Churn | 0.0076 ± 0.0008 | 0.0247 ± 0.0014 | 0.0256 ± 0.0016 | 0.0338 ± 0.002 | 0.0038 ± 0.0005 | 0.0093 ± 0.0004 | 0.0103 ± 0.0006 | 0.0138 ± 0.0004 |
| 1st Ord. Theorem Prov. | 0.0052 ± 0.0006 | 0.0186 ± 0.0013 | 0.0294 ± 0.0197 | 0.0224 ± 0.002 | 0.0019 ± 0.0003 | 0.0048 ± 0.0003 | 0.0065 ± 0.0003 | 0.0087 ± 0.0005 |

*Table 18.* The prediction running time in seconds for 1000 data instances using $KAN^{Via}$ and $MLP^{Via}$.

| Dataset | $KAN^{Via}$ | $MLP^{Via}$ |
|---|---|---|
| Abalone | $0.0024 \pm 0.0003$ | $0.0004 \pm 0.00003$ |
| Ada Prior | $0.003 \pm 0.0008$ | $0.0006 \pm 0.000005$ |
| Adult | $0.0026 \pm 0.0004$ | $0.0006 \pm 0.000005$ |
| Bank32nh | $0.0021 \pm 0.0002$ | $0.0004 \pm 0.0001$ |
| Electricity | $0.0024 \pm 0.0003$ | $0.0005 \pm 0.0002$ |
| Elevators | $0.0021 \pm 0.0002$ | $0.0005 \pm 0.0003$ |
| Fars | $0.0031 \pm 0.0005$ | $0.0009 \pm 0.0001$ |
| Helena | $0.0023 \pm 0.0004$ | $0.0004 \pm 0.0001$ |
| Heloc | $0.0022 \pm 0.0002$ | $0.0003 \pm 0.000005$ |
| Higgs | $0.0022 \pm 0.0002$ | $0.0003 \pm ´0.00001$ |
| LHC Identify Jet | $0.0023 \pm 0.0004$ | $0.0004 \pm 0.00001$ |
| House 16H | $0.0052 \pm 0.0005$ | $0.0004 \pm 0.0001$ |
| Indian Pines | $0.0023 \pm 0.0003$ | $0.0004 \pm 0.0001$ |
| Jannis | $0.0023 \pm 0.0003$ | $0.0004 \pm 0.00001$ |
| JM1 | $0.0026 \pm 0.0012$ | $0.0003 \pm 0.00001$ |
| MagicTelescope | $0.0022 \pm 0.0002$ | $0.0003 \pm 0.00001$ |
| MC1 | $0.0023 \pm 0.0003$ | $0.0004 \pm 0.0001$ |
| Microaggregation 2 | $0.0022 \pm 0.0002$ | $0.0004 \pm 0.00001$ |
| Mozilla 4 | $0.0022 \pm 0.0002$ | $0.0004 \pm 0.0001$ |
| Satellite | $0.0022 \pm 0.0003$ | $0.0004 \pm 0.0001$ |
| PC2 | $0.0021 \pm 0.0003$ | $0.0003 \pm 0.00001$ |
| Phonemes | $0.0021 \pm 0.0001$ | $0.0003 \pm 0.000005$ |
| Pollen | $0.0022 \pm 0.0003$ | $0.0004 \pm 0.0001$ |
| Telco Customer Churn | $0.003 \pm 0.0005$ | $0.0009 \pm 0.0001$ |
| 1st Order Theorem Proving | $0.0022 \pm 0.0003$ | $0.0004 \pm 0.000004$ |

# O. Dataset Details

Table 19 presents an overview of the datasets used in the experiments. The table includes the number of classes, number of features, dataset size, training, validation, and test split sizes. Additionally, the table provides the corresponding dataset ID from OpenML.

Table 19. The dataset information.

| Dataset | # Features | # Classes | Dataset Size | Train. Set | Val. Set | Test Set | OpenML ID |
|---|---|---|---|---|---|---|---|
| Abalone | 8 | 2 | 4177 | 2506 | 836 | 835 | 720 |
| Ada Prior | 14 | 2 | 4562 | 2737 | 913 | 912 | 1037 |
| Adult | 14 | 2 | 48842 | 43957 | 2443 | 2442 | 1590 |
| Bank32nh | 32 | 2 | 8,192 | 5,734 | 1,229 | 1,229 | 833 |
| Electricity | 8 | 2 | 45,312 | 36,249 | 4,532 | 4,531 | 151 |
| Elevators | 18 | 2 | 16,599 | 11,619 | 2,490 | 2,490 | 846 |
| Fars | 29 | 8 | 100,968 | 80,774 | 10,097 | 10,097 | 40672 |
| Helena | 27 | 100 | 65,196 | 41,724 | 10,432 | 13,040 | 41169 |
| Heloc | 22 | 2 | 10,000 | 7,500 | 1,250 | 1,250 | 45023 |
| Higgs | 28 | 2 | 98,050 | 88,245 | 4,903 | 4,902 | 23512 |
| LHC Identify Jet | 16 | 5 | 830,000 | 749,075 | 39,425 | 41,500 | 42468 |
| House 16H | 16 | 2 | 22,784 | 18,227 | 2,279 | 2,278 | 821 |
| Indian Pines | 220 | 8 | 9,144 | 5,852 | 1,463 | 1,829 | 41972 |
| Jannis | 54 | 4 | 83,733 | 53,588 | 13,398 | 16,747 | 41168 |
| JM1 | 21 | 2 | 10,885 | 8,708 | 1,089 | 1,088 | 1053 |
| MagicTelescope | 10 | 2 | 19,020 | 15,216 | 1,902 | 1,902 | 1120 |
| MC1 | 38 | 2 | 9,466 | 7,478 | 994 | 994 | 1056 |
| Microaggregation 2 | 20 | 5 | 20,000 | 12,800 | 3,200 | 4,000 | 41671 |
| Mozilla 4 | 5 | 2 | 15,545 | 12,436 | 1,555 | 1,554 | 1046 |
| Satellite | 36 | 2 | 5,100 | 2,805 | 1,148 | 1,147 | 40900 |
| PC2 | 36 | 2 | 5,589 | 3,353 | 1,118 | 1,118 | 1069 |
| Phonemes | 5 | 2 | 5,404 | 3,782 | 811 | 811 | 1489 |
| Pollen | 5 | 2 | 3,848 | 2,308 | 770 | 770 | 871 |
| Telco Customer Churn | 19 | 2 | 7,043 | 4,930 | 1,057 | 1,056 | 42178 |
| 1st Order Theorem Proving | 51 | 6 | 6,118 | 3,915 | 979 | 1,224 | 1475 |

