# OpenReview forum: "Prediction via Shapley Value Regression"
_ICML.cc/2025/Conference — ICML 2025 poster_

### Official Review · Reviewer_1ExE · 2025-02-28

**Overall Recommendation:** 2

**Summary:**

This paper introduces a framework for estimating Shapley values in explainable AI. Typically, a single observation $\mathbf{x}$ is considered for attribution and gives rise to a set function $v: 2^{[n]} \to \mathbb{R}$. Then we compute the Shapley values for just the set function $v$ i.e., $\phi_i = \frac1{n} \sum_{S \subseteq [n] \setminus \{i\}} \frac{v(S \cup \{i\}) - v(S)}{\binom{n-1}{|S|}}$. If we have many observations $\mathbf{x}$ that we want to explain, this is clearly a computational issue. In prior work (FastSHAP) and this work (ViaSHAP), the goal is to learn one function that simultaneously outputs the Shapley values for many observations $\mathbf{x}$ and the induced set functions $v$.

The authors propose a training method with two loss functions:
• the prediction loss which is the standard measure of how accurately the model can recover the label $y$ for a given input $\mathbf{x}$
• the loss on the Shapley estimates produced by the model, the goal is for the Shapley values to sum to the correct value on a given coalition $S$

They evaluate their method on 25 (impressive!) datasets and four different networks. However, they don't seem to use the ground truth Shapley values and instead estimate them with approximation methods like Kernel SHAP.

**Claims And Evidence:**

They claim "we generate ground truth Shapley values by running KernelSHAP until it converges since it has been demonstrated that KernelSHAP will converge to the true Shapley values when given a sufficiently large number of data samples". But, KernelSHAP produces estimates which are slightly biased so the "sufficiently large number of data samples" required to recover the true Shapley values is $2^n$ since KernelSHAP samples without replacement.

**Essential References Not Discussed:**

I don't think so.

**Experimental Designs Or Analyses:**

I think an obvious and simple baseline is missing: Train a decision tree model (e.g., XGBoost). Then for a given input $\mathbf{x}$, exactly (and efficiently) compute the Shapley values using Tree SHAP. This approach is far simpler than creating their model to produce Shapley values, and the computation of the Shapley values is exact; the only approximation is from the original explainer model to the decision tree model.

**Methods And Evaluation Criteria:**

As mentioned above, I'm concerned about how they compute ground truth Shapley values. I would recommend running experiments using all of the following approaches:
• Use a small value of $n$ where the Shapley values can be computed exactly
• Use a linear model as the model to be explained where the Shapley values are simply the coefficients of the model
• Use a decision tree or forest model as the model to be explained where the Shapley values can be exactly computed using Tree SHAP

**Other Comments Or Suggestions:**

Miscellaneous comments:
• Third paragraph in 2.2 looks like it was supposed to be commented out?
• 2.3 "There are $2^n-1$ possible coalitions", I guess this is semantic but doesn't the empty set trivially count as a coalition?
• Notation of $p(S)$ and $p(x)$ in 2.4 is confusing because two different distributions but written like one function that takes both input data $x$ and coalitions $S$.
• 3.3 second paragraph "bounded domain can [be] represented by a finite sum"
• Figure 3 is very strange to me. Could you represent as a table? Or bar plots?

**Other Strengths And Weaknesses:**

N/A

**Questions For Authors:**

• In the image experiments, are you computing the Shapley values for each pixel in the image input?
• The title "Prediction via Shapley Value Regression" is very broad, and could include e.g., Kernel SHAP. Can you make it more specific and relevant to your work e.g., "A Framework for Simultaneously Predicting Shapley Values on Many Observations"?

**Relation To Broader Scientific Literature:**

Inspired by and a follow up work to Fast SHAP.

**Theoretical Claims:**

They state three lemmas and a theorem, but I would describe these as "decorative theory" that are straightforward and of marginal value.

---

> ### Author Rebuttal · Authors · 2025-03-28
>
> We appreciate the reviewer's time and feedback. Please find our responses below.
>
> > A) It's unclear to me what it means to run KernelSHAP until convergence...
> > B) KernelSHAP produces estimates which are slightly biased...
>
> We agree with the reviewer that KernelSHAP can be biased, therefore, we employed the unbiased KernelSHAP [1]. For the tabular datasets, we allowed unbiased KernelSHAP to continue sampling and updating the learned values iteratively until convergence, meaning that the sampling is not restricted to a fixed number of coalitions but continues until the estimates are stable.
>
> The bias of KernelSHAP shrinks to zero as the sample size grows [1,2]. Furthermore, it has been shown that the unbiased KernelSHAP converges to the true Shapley values given a sufficiently large number of samples [1,3], which we provided for the tabular data. We employ the same evaluation setup that has been employed in [3].
>
> > I'm concerned about how they compute ground truth...Use a small value of $2^n$...
>
> We included datasets with a small number of features $n$, such as Phonemes (5 $n$), Pollen (5 $n$), Mozilla 4 (5 $n$), Abalone (8 $n$), Electricity (8 $n$), and MagicTelescope (10 $n$). For these datasets, the total number of possible coalitions is small enough that the unbiased KernelSHAP solution provides the exact Shapley values, as all possible coalitions are generated.
>
> > ... Use a linear model.... Use a decision tree
>
> We appreciate the reviewer's perspective. However, training a linear model or TreeSHAP to obtain the exact Shapley values may not be applicable to evaluating the explainability of ViaSHAP, as ViaSHAP explains its own predictions rather than the predictions of a separate model, e.g., a decision tree where exact Shapley values can be obtained. Also, ViaSHAP, in the current form, cannot be implemented using decision trees.
>
> > They state three lemmas and a theorem, but I would describe these as "decorative theory"...
>
> We respectfully disagree that the lemmas and the theorem are of marginal value for the following reasons:
>
> 1-The proofs of Lemmas 3.2 and 3.3 **provide upper bounds on the error associated with satisfying the missingness and consistency** properties under realistic optimization assumptions.
>
> 2-Without the theoretical results, there would be no formal guarantee that ViaSHAP computes Shapley values, given it differs from post-hoc explainers and computes the Shapley values before making predictions.
>
> > simple baseline is missing: Train a decision tree model...
>
> We want to clarify again that ViaSHAP explains itself rather than acting as a post-hoc explainer. Since ViaSHAP, as currently proposed, can only be implemented with algorithms that can be optimized using backpropagation (i.e., cannot be implemented using decision trees), we cannot see how training a decision tree-based model and explaining it could help evaluate ViaSHAP's explainability.
>
> > "There are $2^n - 1$ possible coalitions", I guess this is semantic but doesn't the empty set trivially count as a coalition?
>
> We mean that there are $2^n - 1$ possible coalitions that can be used to compute the exact Shapley values, since we cannot add a new player $i$ to the grand coalition $S$ of all players as follows: $\frac{|S|! (n - |S| - 1)!}{n!} (v(S \cup \\{i\\}) - v(S))$. We will update the sentence to be clear.
>
> > Notation of $p(S)$ and $p(x)$ in 2.4 is confusing..
>
> We agree with the reviewer and will update the notation to eliminate confusion.
>
> > Figure 3 is very strange...
>
> Figure 3 is a standard visualization for summarizing the results of Friedman-Nemenyi statistical significance tests. Similar plots have been used in [4,5,6]. For a detailed breakdown of the results, please refer to Table 1.
>
> > are you computing the Shapley values for each pixel...
>
> We computed Shapley values using super-pixels of size 2×2. However, this is a hyperparameter, and users can choose between pixel-wise explanation or larger super-pixel sizes.
>
> > title "Prediction via Shapley Value Regression" is very broad...
>
> Methods like KernelSHAP indeed involve Shapley value regression. However, a key difference is that they do not formulate predictions using the regressed Shapley values, which is central to our approach. We appreciate the reviewer’s suggestion for an alternative title and are open to modifying the title to be more specific.
>
> [1]-Covert, I., et al. Improving kernelshap: Practical shapley value estimation using linear regression. AISTATS 2021.
>
> [2]-Covert, I., et al. Stochastic Amortization: A Unified Approach to Accelerate Feature and Data Attribution. NeurIPS 2024.
>
> [3]-Jethani, N., et al. FastSHAP: Real-time shapley value estimation. ICLR, 2022
>
> [4]-Dedja, K., et al. BELLATREX: Building Explanations through a LocaLly AccuraTe Rule EXtractor.
>
> [5]-Werner, H., et al. Evaluating Different Approaches to Calibrating Conformal Predictive Systems.
>
> [6]-Pugnana, A., et al. Deep Neural Network Benchmarks for Selective Classification

---

### Official Review · Reviewer_86f8 · 2025-03-07

**Overall Recommendation:** 2

**Summary:**

The paper presents a method called ViaSHAP which aims to learn a function to compute the Shapley Values as the model trains. This function predicts the Shapley Values from inputs, directly uses those values to form the model output and bypasses the need for post-hoc computation (i.e. to fit a KernelSHAP to the model). ViaSHAP is implemented as an constraint on a network for an MLP and KAN architecture. The Shapley results are shown to perform comparably to KernelSHAP and FastSHAP across several experiments without a significant loss in model performance.

**Claims And Evidence:**

The claims are quite clear and the experiments are quite exhaustive.

**Essential References Not Discussed:**

The citations are quite exhaustive

**Experimental Designs Or Analyses:**

I'm a bit concerned with how the comparisons between ViaSHAP and the ground truth are made, as noted in lines 358-360, the ground truth KernelSHAP values are computed using the ViaSHAP network as a black box. Is the comparison then how different the output of a model are to a (weighted) least squares approximation of that model itself?

Other than that, the experimental design all made sense to me.

**Methods And Evaluation Criteria:**

Comparisons with KernelSHAP and FastSHAP across the tested datasets makes sense for this application. The performance of ViaSHAP is then compared with XGBoost, TabNet, and Random Forests to validate the model performance.

**Other Comments Or Suggestions:**

Overall I quite like the idea of the paper, some minor issues in below sections.

Minor Comments:
- The paper focuses on the classification case with regression being briefly discussed in the appendices, which doesn't seem to reflect the paper title very well

**Other Strengths And Weaknesses:**

Posed in questions section.

**Questions For Authors:**

Questions:
- I would suggest incorporating more discussion about when this method would be best used, and why one would choose it over other
methods.
- Why is it the case that the results are particularly weak over certain datasets (such as pollen)?
- In Table 1, why is it the case that the AUC of XGBoost does not have a +- value associated? Additionally, what exactly do the +- values indicate, are they variations over random (seed) initialization of the model, or over different train/test splits?
- What exactly does Figure 5 show? I'm not sure how much this illustration contributes to the main paper.

**Relation To Broader Scientific Literature:**

This paper contributes a method of significantly reducing the cost of computing Shapley Values by removing the need for post-hoc fitting.

**Theoretical Claims:**

I don't see any particular issues with the proofs presented.

---

> ### Author Rebuttal · Authors · 2025-03-28
>
> We appreciate the reviewer's time and feedback on our paper. Below, we provide our responses and clarifications.
>
> > I'm a bit concerned with how the comparisons between ViaSHAP and the ground truth are made...
>
> We appreciate the reviewer’s concern and would like to clarify our experimental setup. In the explainability evaluation, we treated ViaSHAP models as standard black boxes and computed the ground truth Shapley values using the unbiased KernelSHAP [1], which involves solving an optimization problem for each prediction. We then compared the explanations generated by ViaSHAP to the ground truth obtained from the unbiased KernelSHAP.
>
> As established in previous work, the bias of KernelSHAP shrinks to zero as the sample size grows [1, 2], and with a sufficiently large number of samples, it converges to the true Shapley values [1, 3]. For the tabular datasets, we allowed the unbiased KernelSHAP to continue sampling and updating the learned values until convergence. We followed the same evaluation setup using the unbiased KernelSHAP that has been employed in [3].
>
> > The paper focuses on the classification case with regression being briefly discussed in the appendices...
>
> We appreciate the reviewer’s feedback and acknowledge that incorporating additional regression datasets could further strengthen the experiments. However, due to the constraints of the rebuttal phase, we are unable to add new results at this stage.
>
> We would also like to clarify that the term *regression* in the paper title, *Prediction via Shapley Value Regression*, specifically refers to the fact that we solve the regression task of Shapley values and use Shapley values in the prediction task itself. In other words, the title is intended to convey *"(Classification and Regression) via Shapley Value Regression."*
>
> > incorporating more discussion about when this method would be best used...
>
> We sincerely appreciate the reviewer’s suggestion and will update the manuscript accordingly.
> ViaSHAP is particularly advantageous in settings where computational resources or time are limited, as it removes the need to train and run separate models for prediction and explanation. Moreover, it is well-suited for scenarios where high-fidelity explanations are crucial, as the model’s predictions are inherently tied to the explanations.
>
> Post-hoc methods might be more suitable in cases where a pre-trained black-box model is already employed or when users cannot access or modify the black-box model.
>
> > Why is it the case that the results are particularly weak over certain datasets (such as pollen)?
>
> Similar to other machine learning models, ViaSHAP's performance can vary depending on the characteristics of the dataset. Certain datasets pose inherent challenges, such as a limited number of training examples, a high-dimensional feature space, complex data distributions, or noisy data, all of which can impact model performance.
> While ViaSHAP's performance on the Pollen dataset is relatively weaker compared to other datasets, it is noteworthy that it also outperforms XGBoost, Random Forests, and TabNet on this dataset, which suggests that the Pollen dataset is inherently challenging, rather than an issue specific to ViaSHAP.
>
> > ...why is it the case that the AUC of XGBoost does not have a +- value...
>
> All compared models are trained and evaluated using the same data splits to ensure a fair comparison. The $\pm$ values indicate variations in performance due to different random initializations, as stated in the experimental setup (Lines [312, 315]):
> *"If the model’s performance varies with different random seeds, it will be trained using five different seeds, and the average result will be reported alongside the standard deviation."*
> Since XGBoost with the default settings is deterministic and its performance does not vary across different random seeds, its AUC is reported without a $\pm$ value.
>
> > What exactly does Figure 5 show? I'm not sure how much this illustration contributes to the main paper.
>
> Figure 5 illustrates that the explanations generated by ViaSHAP are more precise compared to those from FastSHAP when applied to the same image. Specifically, FastSHAP tends to highlight broader regions, whereas ViaSHAP primarily focuses on the shape of the classified instance within the image, providing a more precise and sparse explanation.
>
> [1] Covert, I. and Lee, S.-I. Improving kernelshap: Practical shapley value estimation using linear regression. In Proceedings of The 24th International Conference on Artificial Intelligence and Statistics.
>
> [2] Covert, I., Kim, C., Lee, S., Zou, J., Hashimoto, T. Stochastic Amortization: A Unified Approach to Accelerate Feature and Data Attribution. In The Thirty-eighth Annual Conference on Neural Information Processing Systems 2024.
>
> [3] Jethani, N., Sudarshan, M., Covert, I. C., Lee, S.-I., and Ranganath, R. FastSHAP: Real-time shapley value estimation. In the International Conference on Learning Representations, 2022

---

> > ### Comment · Reviewer_86f8 · 2025-04-03
> >
> > Thanks answering my questions.
> >
> > I have concerns about the evaluation methodology used in this paper. As answered in the question response, the results are averaged over initialization (i.e. random seeds), but not choices of train/test splits which means this comparison would not hold more generally, is there any reason that this comparison was not made?

---

> > > ### Author Response · Authors · 2025-04-03
> > >
> > > We thank the reviewer for pointing out their concern in a direct question. Our experimental setup is motivated by the following reasons:
> > >
> > > **1-** Since our primary objective is to evaluate the predictive performance of each model relative to the competing algorithms, all models are trained and evaluated on the same splits to ensure that performance differences arise from model differences rather than variations in the data splits.
> > >
> > > **2-** Using fixed train/test splits allows for isolating the impact of random initializations from that of data partitioning. Introducing both multiple data splits and multiple random seeds simultaneously would make it difficult to disentangle their individual effects on performance variations, i.e., would blur **the evaluation of the model's robustness to random initializations**.
> > >
> > > **3-** While we agree that using **a single split** on **a single dataset** can possibly make the evaluation biased. However, we here employ a large number of medium to large-sized datasets (as detailed in Table 19). Therefore, a bias due to a specific train/test split is eliminated by the large number of datasets as well as the large number of data examples, and the results generalize beyond a specific split.
> > >
> > > **4-** We employ statistical significance tests (Friedman and Nemenyi) to confirm whether performance differences are not due to random chance but are meaningful comparisons, which directly counters the concern that performance variations might be due to dataset-specific splits rather than actual model differences.
> > >
> > > We hope that our answer sufficiently addresses the reviewer's concerns.

---

### Official Review · Reviewer_L8aK · 2025-03-14

**Overall Recommendation:** 3

**Summary:**

This paper proposes a method that simultaneously computes both the Shapley values and the predicted output, where the predicted output is equal to the sum of the Shapley values. To achieve this, the authors train the network to learn and approximate Shapley values by minimizing the weighted least squares in Eq. (6), while simultaneously optimizing the predicted output in Eq. (7). The authors employ MLP and KAN as the model architectures. Experimental results on several tabular datasets demonstrate the effectiveness of the proposed explanation method.

**Claims And Evidence:**

1.	My major concern is its positioning within the existing literature. If the proposed method is model-specific, prior works such as [1][2] have already demonstrated approaches to compute exact Shapley values in a single forward pass. If the method is intended to be model-agnostic, various approaches [3][4][5] already exist to compute Shapley values either in an unbiased or biased manner.

Since the proposed method does not apply to arbitrary black-box models, but rather trains an inherently interpretable model that also predicts outputs, the authors should sufficiently compare it with prior works (not limited to the listed interpretable models), and highlight the advantages of the proposed method.

2.	The main challenge of jointly training a model to predict Shapley values while performing task predictions is that, it may be difficult to achieve great model performance on complex tasks with relatively accurate Shapley values. For example, on complex datasets such as CIFAR-10, CIFAR-100, and Tiny ImageNet, the authors should report both the classification accuracy and the approximation error of the Shapley values (e.g., root mean square error). Besides, the choice of the trade-off parameter $\beta$ in Algorithm 1 should be reported in different tasks.



[1] Chen et al. HarsanyiNet: Computing Accurate Shapley Values in a Single Forward Propagation. ICML, 2023.

[2] Wang et al. Shapley explanation networks. ICLR, 2021.

[3] Castro et al. Polynomial calculation of the shapley value based on sampling. Computers & Operations Research, 36(5):1726–1730, 2009.

[4] Mitchell et al. Sampling permutations for shapley value estimation. Journal of Machine Learning Research, 23:1–46, 2022.

[5] Covert et al. Improving kernelshap: Practical shapley value estimation using linear regression. International Conference on Artificial Intelligence and Statistics, 2021.

**Essential References Not Discussed:**

The paper lacks a discussion of related works in closely related directions.

1.	Methods that compute exact Shapley values and predict model outputs in a single forward pass:

[1] Chen et al. HarsanyiNet: Computing Accurate Shapley Values in a Single Forward Propagation. ICML, 2023.

[2] Wang et al. Shapley explanation networks. ICLR, 2021.


2.	Approximation algorithms for Shapley values:

[3] Castro et al. Polynomial calculation of the shapley value based on sampling. Computers & Operations Research, 36(5):1726–1730, 2009.

[4] Mitchell et al. Sampling permutations for shapley value estimation. Journal of Machine Learning Research, 23:1–46, 2022.

[5] Covert et al. Improving kernelshap: Practical shapley value estimation using linear regression. International Conference on Artificial Intelligence and Statistics, 2021.

3. Other methods for attribution computation in a single forward pass

**Experimental Designs Or Analyses:**

1.	In Lines 278-283, using KernelSHAP as the ground-truth Shapley value may be inappropriate. One concern is that the number of samples required for KernelSHAP to produce reliable estimates depends on the complexity of the task [1]. A more rigorous approach would be to compute exact Shapley values using the definition for datasets with a small number of input variables. For larger image datasets, the authors should compare their method with unbiased sampling-based Shapley value estimation [3].

2.	Using cosine similarity as the metric for evaluating the difference between approximated and ground-truth Shapley values may be insufficient. The authors could consider incorporating additional metrics, such as root mean square error, to provide a more comprehensive assessment.


3.	In Section 4.3, when comparing interpretability methods, it is necessary to include a broader set of model-agnostic approaches, such as those proposed in [3], [4], and [5].

**Methods And Evaluation Criteria:**

Is the second categorical loss in Eq. (7) incorrectly written? Why is it $-j log (\hat{j})$, where $j$ is the category, $j\in\\{1, \cdots, d\\}$, and $\hat{j}$ is the probability of the $j$-th category.

In addition, it would be helpful to explicitly explain the physical meaning of Eq. (6) to improve clarity.

**Other Comments Or Suggestions:**

N/A

**Other Strengths And Weaknesses:**

Strengths:

1.	The paper is well written and easy to follow.

2.	The authors explored the possibility of training both predicted shapley values and predicted outcomes.

Weaknesses:

See “Claims And Evidence”,“Methods And Evaluation Criteria” and “Experimental Designs Or Analyses.”

**Questions For Authors:**

See “Claims And Evidence”,“Methods And Evaluation Criteria” and “Experimental Designs Or Analyses.”

**Relation To Broader Scientific Literature:**

See "Claims And Evidence"

**Theoretical Claims:**

Theorem 1 is not carefully checked.

---

> ### Author Rebuttal · Authors · 2025-03-28
>
> We appreciate the reviewer's feedback. We provide answers in the following part.
>
> > My major concern is its positioning...
>
> The proposed method does not fall strictly into the categories of model-specific or model-agnostic approaches. Instead, we introduce a framework for training by-design explainable models, where predictions and their Shapley value explanations are learned simultaneously. This distinguishes our approach from existing methods that compute Shapley values post-hoc or in a separate forward pass.
>
> > ... authors should sufficiently compare it with prior works...
>
> We appreciate the reviewer’s suggestion. Our primary objective, however, is to demonstrate that the proposed method is inherently explainable and that the generated explanations are accurate. To this end, we compared our approach to the unbiased KernelSHAP, as well as FastSHAP, which directly inspired our method and represents the most closely related work. Unfortunately, due to the constraints of the rebuttal phase, we are unable to add new results at this stage. However, we acknowledge the value of broader comparisons.
>
> > ...on complex datasets such as CIFAR-10...
>
> We have indeed reported the model’s predictive performance on complex datasets such as CIFAR-10, along with the approximation error of the explanation, and compared the results to FastSHAP. We kindly refer the reviewer to Appendix F, which provides detailed results for CIFAR-10.
> The evaluation of Shapley value estimates on images using the root mean square error requires access to ground truth Shapley values, which is computationally infeasible for image datasets. Instead, we assess explanation quality using inclusion and exclusion curves, which measure an explanation’s ability to identify informative image regions. This evaluation strategy is commonly used for image datasets, as seen in [1,2,3].
>
> > Is the second categorical loss in Eq. (7) incorrectly written?...
>
> We thank the reviewer for pointing out that the notations in Eq.7 are confusing. Indeed, j in Eq.7 represents the true probability to be estimated for each category. We will correct and clarify this in the revised paper.
>
> > ...explicitly explain the physical meaning of Eq. (6)...
>
> We appreciate the reviewer’s suggestion. We explained the intuition behind Eq.6 in lines [173,186] and illustrated the main idea in Figure 2 and Algorithm 1. However, as recommended by the reviewer, we will further refine our explanation to enhance clarity.
>
> > ...using KernelSHAP as the ground-truth...
>
> We agree with the reviewer that KernelSHAP is unreliable, particularly for high-dimensional data. Therefore, we employed an unbiased version of KernelSHAP [4], which aligns with the reviewer’s suggested approach [5], as mentioned in lines 358 and 359 " the ground truth Shapley values ($\phi$), computed by the unbiased KernelSHAP". The same evaluation setup, using unbiased KernelSHAP, has been employed in [3]. For the tabular datasets, we allowed the unbiased KernelSHAP to keep sampling until convergence. The bias of KernelSHAP shrinks to zero as the sample size grows [4,6]. It has been shown that the unbiased KernelSHAP converges to the true values if provided with a large enough number of samples [3,4], which we did for the tabular data in the experiments.
>
> > Using cosine similarity as the metric...
>
> We completely agree with the reviewer that cosine similarity alone is insufficient for comparing explanations. Therefore, we employed three metrics: Spearman rank correlation, $R^2$, and cosine similarity. The results are reported using the three metrics in Tables 7, 9, 12, 14, and 16.
>
> > ... include a broader set of model-agnostic approaches...
>
> We acknowledge that incorporating additional experiments can enhance the quality of the paper. We want to clarify that the method in reference [5] provided by the reviewer (Covert et al. Improving KernelSHAP...) is the unbiased KernelSHAP method, which we have already compared our proposed method with in our evaluation.
>
> > lacks a discussion of related works...
>
> We appreciate the reviewer’s suggestions for additional related work to discuss. We kindly clarify that the papers (Mitchell et al. Sampling Permutations for Shapley Value Estimation) and (Covert et al. Improving KernelSHAP) are already cited multiple times in our manuscript.
>
> [1] Hooker, S., et al. A benchmark for interpretability methods in deep neural networks.
>
> [2] Jethani, N., et al. Have we learned to explain?: How interpretability methods can learn to encode predictions in their interpretations. AISTATS 2021
>
> [3] Jethani, N., et al. FastSHAP: Real-time shapley value estimation. ICLR, 2022
>
> [4] Covert, I. et al. Improving kernelshap: Practical shapley value estimation using linear regression. AISTATS 2021
>
> [5] Castro, J., et al. Polynomial calculation of the shapley value based on sampling. Computers & Operations Research
>
> [6] Covert, I., et al. Stochastic Amortization: A Unified Approach to Accelerate Feature and Data Attribution.

---

> > ### Comment · Reviewer_L8aK · 2025-04-04
> >
> > Thank you for your detailed response. Your response has basically addressed my concerns, and I will raise my score accordingly. However, I strongly encourage the authors to incorporate the following revisions in the next version of the manuscript, which would further improve clarity and better highlight the contribution of the work,
> >
> > **1.	Clarification of the unbiased KernelSHAP**
> >
> > Please clearly state in Section 2.3 or in the Introduction that you use the **unbiased KernelSHAP** method as the baseline, rather than introducing this only in the experimental section. This clarification is particularly important as Reviewer 1ExE also raised concerns regarding estimation bias. Since one of the goals in Shapley value estimation is to approximate the ground-truth values accurately, both **the choice of ground truth values** and **the evaluation metrics** must be clearly stated.
> >
> > **2.	Comparisons with other unbiased methods**
> >
> > As representative unbiased model-agnostic methods, [3] and [4] should be included in experimental comparisons, both on tabular datasets and on image datasets. These methods, like unbiased KernelSHAP, serve as natural baselines. It would be helpful to show that the gap between the proposed method and these unbiased estimators narrows as the number of samples increases. Such results would strengthen the empirical justification of your method. I recommend presenting these results in the main paper, possibly in a format similar to Figure 1 in [6].
> >
> > **3.	Similarity Metrics**
> >
> > In addition to the similarity metric currently used, please consider including more standard metrics that are commonly used to evaluate Shapley value approximation accuracy, such as the root mean squared error (RMSE) used in [6] or mean squared error (MSE) used in [5]. Please also cite the respective works where these metrics were used to improve clarity.
> >
> > **4.	Distinguishing from Prior Work**
> >
> > Please explicitly distinguish the proposed method from that of [1,2]，especially in terms of application scenarios. These works train inherently explainable models that output both predictions and exact Shapley values in a single forward pass.
> >
> > Given the constraints of the rebuttal phase and the absence of above results in the current version, I cannot give a higher score. Nonetheless, I believe these additions would substantially enhance the quality of the manuscript and more clearly convey the significance of the contribution.
> >
> >
> > [6] Ancona et al. Explaining Deep Neural Networks with a Polynomial Time Algorithm for Shapley Values Approximation, ICML 2019.

---

> > > ### Author Response · Authors · 2025-04-04
> > >
> > > Thank you very much for your constructive feedback. We sincerely appreciate your positive assessment. We fully agree with your suggestions, and we acknowledge that incorporating these revisions will significantly improve both the clarity and strength of the manuscript.
> > >
> > > Currently, we are updating our manuscript accordingly and are committed to implementing your recommendations in the next revision of the manuscript.

---

### Official Review · Reviewer_TGJg · 2025-03-17

**Overall Recommendation:** 3

**Summary:**

The paper proposes a new method, ViaSHAP, which learns a function that computes Shapley values alongside the model’s prediction. This method works by training a machine learning model that minimizes a weighed least squares lost similar to that of KernelSHAP and FastSHAP. The authors present many experiments showing that the predictions of the model on tabular data are on-par with state-of-the-art tabular models (XGBoost, MLP, Kolmogorov-Arnold Networks) and that the explanations are more accurate than FastSHAP.

**Claims And Evidence:**

Strengths

The authors perform experiments across a large number of datasets, showing that the predictive power of ViaSHAP is on-par with other methods. Additionally, they show that ViaSHAP generates more accurate explanations than FastSHAP. The main selling point of ViaSHAP is to train a model that simultaneously provides accurate predictions and Shapley values.

Weaknesses

The authors argue that one major drawback of existing methods is that Shapley values are computed post-hoc, that is, after a model has already been trained. More specifically, “generating instance-base explanations or learning a pre-trained explainer always demands
further data, time, and resources” (page 1). The authors’ main selling point of ViaSHAP, then, is the ability to train a single model that generates both accurate predictions and Shapley values—training a model plus training a FastSHAP explainer model unnecessarily increases computational resources. However, the importance of computing Shapley values efficiently lies in explaining already existing machine learning models. Therefore, one might argue that the authors’ claim that generating Shapley explanations post-hoc is a drawback is a fundamentally biased view of existing Shapley literature.

The methodologies are highly similar and inspired by FashSHAP, even though FastSHAP is a posthoc method and ViaSHAP is not.

**Essential References Not Discussed:**

It would be good to discuss the recent papers on amortized SHAP, which directly related to the fast computation of SHAP, for example,

[1] Stochastic Amortization: A Unified Approach to Accelerate Feature and Data Attribution https://arxiv.org/pdf/2401.15866
[2] SHAP zero Explains Genomic Models with Near-zero Marginal Cost for Future Queried Sequences https://arxiv.org/abs/2410.19236

**Experimental Designs Or Analyses:**

Extensive empirical evaluations have been performed. Several of these are supplementary, perhaps due to space constraints. But it would be good to have some summary tables of these results in the main text.

**Methods And Evaluation Criteria:**

The method is explained clearly, and the evaluation criteria make sense.

**Other Comments Or Suggestions:**

Discussed before.

**Other Strengths And Weaknesses:**

Discussed before.

**Questions For Authors:**

1) In what problem settings should one decide to use ViaSHAP instead of training a regular model and then apply FastSHAP?
2) In total, how much computation time will be saved using ViaSHAP in the previous question?
3) Can ViaSHAP be used while training LLMs? What are the tradeoffs and fundamental computational limits?

**Relation To Broader Scientific Literature:**

There are already several works on computing Shapley's explanation faster. Again, even though this work is new from the angel learning to explain rather than postdoc explanation, still the methologies is not very distinct from FastSHAP.

**Theoretical Claims:**

No theoretical claims.

---

> ### Author Rebuttal · Authors · 2025-03-28
>
> We thank the reviewer for their time and valuable feedback on our manuscript. We provide answers and clarifications below.
>
> > The authors argue that one major drawback of existing methods is that Shapley values are computed post-hoc....
>
> We do not intend to suggest that computing Shapley values in a post-hoc setup is inherently a drawback. Rather, our work builds on existing post-hoc methods and aims to further reduce computational costs by eliminating the need for training a separate explainer model and running two models at inference time. In particular, it is true that in cases where a pre-trained model already exists or when users cannot access or modify the black-box model, post-hoc methods might be more suitable. Nonetheless, this is not always the case. ViaSHAP offers an alternative for users who prioritize developing an inherently explainable model, as well as for scenarios where computational efficiency and resources are important considerations.
>
> > There are already several works on computing Shapley's explanation faster. Again, even though this work is new from the angel learning to explain rather than postdoc explanation, still the methologies is not very distinct from FastSHAP.
>
> Our proposed method is indeed inspired by FastSHAP. However, a key difference is that we do not start with a pre-trained black-box model that requires post-hoc explanations. This difference is fundamental in the settings, i.e., building a by-design explainable model vs. explaining a pre-trained model. In the adapted setting, ViaSHAP thus reduces the computational cost needed from training two models to a single one. The use of a by-design explainable model also ensures that the model explains itself, thus providing explanations that align with the prediction. On the other hand, explaining a black-box model through post-hoc methods leaves the explanations open to the inherent randomness issues of statistical learning [1].
>
> > It would be good to discuss the recent papers on amortized SHAP...
>
> We thank the reviewer for pointing out the missing recent paper. We will indeed add it to the discussion.
>
> > In what problem settings should one decide to use ViaSHAP instead of training a regular model and then apply FastSHAP?
>
> ViaSHAP is particularly beneficial in environments with limited time or computational resources, as it eliminates the need to train and run separate models for prediction and explanation. Additionally, it is well-suited for scenarios where high-fidelity explanations are a central concern, as the model’s predictions are inherently tied to its explanations.
>
> On the other hand, post-hoc explanation methods remain necessary when users cannot access or modify the black-box model or when the model does not support optimization via gradients and backpropagation. We do not propose to replace or discard post-hoc methods, but rather to offer an alternative approach that promotes explainability while addressing computational efficiency.
>
> > how much computation time will be saved using ViaSHAP...
>
> When comparing ViaSHAP to KernelSHAP, ViaSHAP significantly reduces computational cost at inference time by avoiding the need to solve a separate optimization problem for each prediction. Therefore, computational cost is reduced by an order of magnitude, as demonstrated in Appendix L and Table 15.
>
> When compared to real-time explanation methods such as FastSHAP, ViaSHAP further reduces computation at training time (by avoiding the training of a separate explainer model) and at inference time (by running inference on a single model instead of two).
>
> More importantly, the computational cost of ViaSHAP at inference time remains the same as a similar model that does not compute Shapley values. Detailed computational cost analyses for ViaSHAP can be found in Appendix N.
>
> > Can ViaSHAP be used while training LLMs? What are the tradeoffs and fundamental computational limits?
>
> In principle, ViaSHAP could be adapted to train LLMs. However, there are significant challenges. For instance, LLM training is exceptionally computationally intensive, and incorporating Shapley-based explanations during training would add more computational burden at training time, as ViaSHAP requires sampling from the input and propagating gradients. From the previously evoked results (Appendix N), the inference time should not be harmed by this procedure.
>
> Another possible solution is to use ViaSHAP on pre-trained LLMs, where all the learned parameters of the model are fixed, and one or two output layers are modified and optimized to approximate the explanation.
>
> That being said, we cannot provide an accurate answer without a proper study similar to what we did with tabular data and images.
>
> [1] Rudin et al., Stop explaining black box machine learning models for high stakes decisions and use interpretable models instead, 2019

---

### Decision · Program_Chairs · 2025-05-01

**Decision:**

Accept (poster)

**Comment:**

This paper proposes a method dubbed ViaShap, which is an interesting variation of Shapley-based methods that jointly learns to predict outputs and compute Shapley value explanations in a single model. This sits in-between learning to explain and post-hoc methods; Unlike the latter, ViaSHAP trains a network (via MLPs and KANs) such that the output is the sum of its learned Shapley values. The method optimizes two losses: one for prediction accuracy and another to ensure Shapley consistency via a weighted least squares loss inspired by KernelSHAP and FastSHAP. Experimental results across 25 tabular datasets show that ViaSHAP achieves predictive performance comparable to state-of-the-art models (e.g., XGBoost, MLP) and produces more accurate Shapley explanations than FastSHAP, without significant performance degradation. While exact Shapley values are not used, the authors benchmark their estimates against approximations from KernelSHAP.

There is broad general support for this papers. All reviewers praised the extensive numerical validation across a large number of diverse datasets. Reviewers TGJg and L8aK had some concerns on similarities and differences with FastShap, Kernel Shap, the potential limited utility (not being applicable to off-the-shelf models), and concerns on their notions of "ground-truth" Shapley values. All/most concerns were addressed successfully.

Reviewer 86f8 had some concerns on the empirical evaluation. These were clarified by the authors, and I agree with the authors in that the score provided by 86f8 is disproportionately low compared to their written evaluation. Finally, reviewer 1ExE had some concerns on the empirical validation and methods of comparison, and suggested experiments based on tree-based models. The authors addressed these concerns explaining why they are not relevant here. While I agree that the experiment suggested by the reviewer is interesting (and the authors should consider adding it to the paper), it is not severe to prevent my recommendation for acceptance.